# Two-dimensional and high-order directional information modulations for secure communications based on programmable metasurface

Hui Xu[1,2,5], Jun Wei Wu [1,2,3,4,5], Zheng Xing Wang[1,2,5], Rui Wen Shao[1,2], Han Qing Yang[1,2] & Tie Jun Cui [1,2,3,4] ✉

Conventional wireless communication schemes indiscriminately transmit information into the whole space and pose inherent security risks. Recently, directional information modulation (DIM) has attracted enormous attention as a promising technology. DIM generates correct constellation symbols in the desired directions and distorts them in undesired directions, thus ensuring the security of the transmitted information. Although several DIM schemes have been reported, they suffer from defects of bulkiness, energy consumption, high cost, and inability to support two-dimensional (2D) and high-order modulations. Here, we propose a DIM scheme based on a 2-bit programmable metasurface (PM) that overcomes these defects. A fast and efficient discrete optimization algorithm is developed to optimize the digital coding sequences, and the correct constellation symbols can be generated and transmitted in multi-directional beams. As a proof-of-concept, three sets of constellation diagrams (8 phase shift keying (PSK), 16 quadrature amplitude modulation (QAM), and 64QAM) are realized in the multi-channel modes. This work provides an important route of employing DIM for ensuring physical-layer security and serves as a stepping stone toward endogenous secure communications.

The ever-increasing demands for higher data rates, lower latency, and fewer error rates are pushing the next-generation wireless communication systems toward higher carrier frequency and extremely large-scale antenna arrays[1–4]. However, a critical challenge is the security and resilience of communication networks against malicious eavesdropping[5,6]. The traditional cryptographic methods at the network layer increase the length of message code and transmission overhead, and need a key exchange, making it difficult to meet the requirements for high-bandwidth and ultralow-latency communication systems[7].

Over the past years, diverse physical-layer secure methods, including beamforming techniques with phased arrays and cooperative jamming with artificial noise, have been developed to mitigate the pressing issue[8–10]. The metasurface-coated device mitigates the critical problem of the tradeoff between ultralow-power transmissions and secrecy capacity for the first time, which is a great improvement compared with the traditional beamforming techniques[11,12]. These methods are motivated by Wyner's wiretap model, which measures security from an information-theoretic perspective[13]. The model formulates a secrecy metric as the difference in channel capacities

[1]Institute of Electromagnetic Space, Southeast University, Nanjing, China. [2]State Key Laboratory of Millimeter Waves, School of Information Science and Engineering, Southeast University, Nanjing, China. [3]Peng Cheng Laboratory, Shenzhen, Guangdong, China. [4]Pazhou Laboratory (Huangpu), Guangzhou, Guangdong, China. [5]These authors contributed equally: Hui Xu, Jun Wei Wu, Zheng Xing Wang. ✉e-mail: tjcui@seu.edu.cn

between the transmitter to a legitimate receiver (Alice to Bob) and the transmitter to an eavesdropper (Alice to Eve). Especially, $C_s = \log_2(1 + P_{Bob}/N_{Bob}) - \log_2(1 + P_{Eve}/N_{Eve})$, where $P_{Bob}$ and $P_{Eve}$ are the powers received by Bob and Eve, respectively, and $N_{Bob}$ and $N_{Eve}$ denote the noises. The above techniques increase the signal-to-noise ratio (SNR) disparity between the main lobe beam (Alice to Bob) and the side lobe beam (Alice to Eve). However, the security risk of the above strategies arises from the fact that a transmitter radiates undistorted signals to all directions indiscriminately, and an eavesdropper equipped with a sensitive receiver can intercept the information in theory. As a result, the demand for information security necessitates the idea of directional communications.

Directional information modulation (DIM), as one of the promising physical-layer security techniques, leverages the beamforming capability of multiple antennas to transmit the correct constellation symbols in the desired directions while distorting them into noise along other illegal directions[14–16]. The realization of the DIM system necessitates an appropriate transmitter architecture and an effective algorithm mapping the constellation symbols to the excitation vector. Serval primary implementations currently dominate the DIM design. The first involves phased arrays to realize a single-channel quadrature phase shift keying (QPSK) modulation empowered by the expensive transmitter/receiver components and heuristic algorithms[14]. The second category is time-modulated array (TMA), which focuses on DIM at harmonics through sensitive switching devices and pre-designed periodic waveform sequences[17–21]. Both classes of architectures can generate element responses of arbitrary magnitudes and phases, making them fairly powerful for DIM. However, these implementations are currently facing the limitations of expensive hardware and high energy consumption. Due to these defects, the current transmitters are mainly linear arrays with a small number of elements and only support one-dimensional transmissions. Furthermore, the parasitic harmonic signals produced by TMA will carry the information at the same time, and hence there are still secure risks.

More recently, programmable metasurface (PM) has been exploited for DIM due to its flexible and advanced capabilities for real-time manipulations of EM waves[22,23]. The PM can serve as a highly integrated communication system with simpler architecture, lower cost, and less energy consumption[24]. There have been several efforts aiming to realize directional communications based on PM, including three-channel near-field amplitude shift keying (ASK) modulation[25], dual-channel far-field quadrature phase shift keying (QPSK) modulation[26], and dual-channel far-field ASK modulation[27–29]. Moreover, to tackle the security risk in the condition that an eavesdropper is close to the desired user, a random-signal carrier was utilized to excite the metasurface[30]. Firstly, they only transmit signals. This is incompatible with the operation of traditional wireless systems, where the base station or terminal device can both transmit and receive the information. Secondly, the metasurface-based transmitter typically needs external RF sources and has a large profile, which is inconvenient for space-constrained applications such as wearable Internet of Things (IoT) devices, vehicles, and aircraft. The above-mentioned prototypes only use either the phase or magnitude features of EM waves, which suffer from a lack of high-order modulation and quadrature amplitude modulation (QAM) schemes that carry more information capacities. Moreover, the required energy per bit is high due to the low information capacity of a single transmission (see Supplementary Note 1 for more details).

Here, we propose and experimentally demonstrate a DIM scheme based on a two-dimensional (2D) PM[31]. The scheme integrates the controllable components and generates correct constellation symbols in the desired directions, forming a reconfigurable and low-profile modulator that provides simultaneous and independent communication links between the transmitter and multiple receivers. The scheme can also serve as a transmitter or receiver and is compatible with the conventional communication system. A fast and efficient algorithm in

the framework of the alternating direction method of multipliers (ADMM) is proposed to optimize the coding sequences given the discrete phase constraint of the element. Using the optimized non-periodic time-space coding sequences, our scheme is directional secure against harmonic (see Supplementary Note 2 for more details). As proof-of-concept examples, three sets of constellation diagrams, including 8PSK, 16QAM, and 64QAM schemes, are validated in the multi-channel modes. The measurements demonstrate that the received signals maintain the same structure as preassigned constellation diagrams, indicating that the system supports direct transmissions of digital information. Moreover, the transmission is directional, i.e., only the users in the desired directions can receive the correct symbols, while the users in other directions will receive the distorted symbols, which ensures the security of information.

## Results
### Principle
The PM-based DIM scheme is depicted in Fig. 1. It consists of four parts, namely, multi-channel bit streams, a microcontroller unit (MCU), a PM-based transmitter, and user equipment (UE) located in different directions. The scheme directly transmits the digital information, and the mechanism is presented as follows. Firstly, the phase coding library of the PM is determined by an efficient discrete optimization algorithm, and the aim is to generate the symbols of the specified modulation scheme in the desired directions. Then, the bit streams are translated into symbols, which are further mapped into the phase coding of PM by indexing the library. The phase coding is then imposed on the tunable components of PM via MCU. Finally, UEs extract the information bits by comparing the received signals with the corresponding constellation symbols. The desired UEs receive the regular constellation diagrams, which have a phase shift due to the EM wave propagation and can be compensated by adding an identical phase bias for each received signal. On the contrary, the undesired UEs will receive distorted constellation diagrams. Since the structure of the received signals is disturbed, the correct information acquisition is nearly impossible, regardless of the receiver sensitivity.

We briefly present the element of the PM utilized in this work. The geometric size is $13 \times 13 \times 2.5$ mm ($0.48 \times 0.48 \times 0.09\,\lambda$ at 11 GHz), exhibiting the low-profile feature which has not been reported in the previous DIM schemes based on the transmissive or reflective metasurfaces. According to the principle of the DIM scheme, the hardware can work in the transmitting or receiving mode (see Supplementary Note 3 for more details). The main difference between the transmitting and receiving

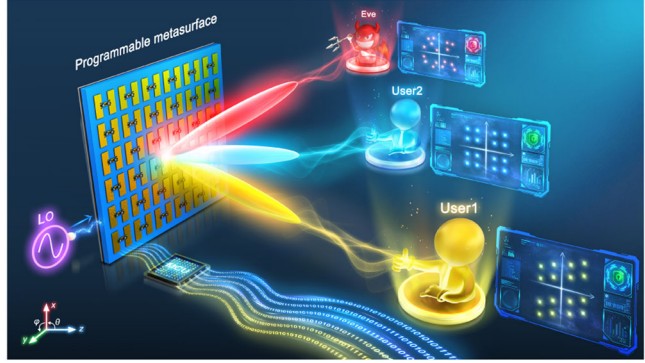

**Fig. 1 | Schematic diagram of the PM-based DIM scheme.** Multiple-bit streams are mapped into the optimized coding sequences that are then imposed on the tunable components of PM via MCU. The radiation fields in the desired directions contain the correct symbols, which are interpreted by the intended users. On the contrary, the fields in the undesired directions provide the unintended user (Eve) with distorted symbols, which cannot be interpreted, thereby ensuring DIM and physical security.

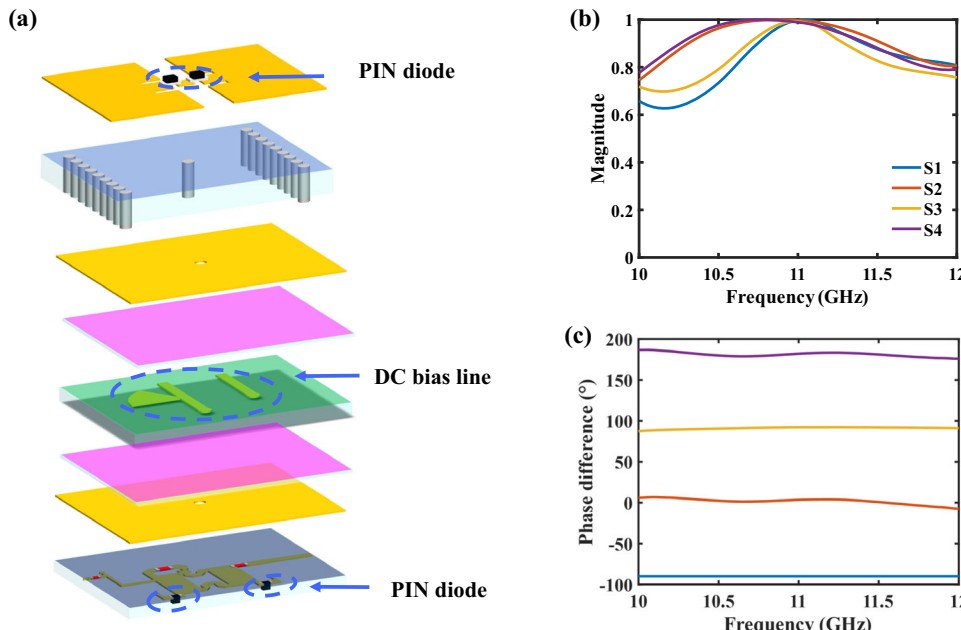

**Fig. 2 | Details of the meta-element used in the PM-based DIM scheme. a** Explosion view of the meta-element. **b** Simulations of the magnitude and **c** phase responses of the element.

modes is the coding pattern. The former is updated in real-time with the input constellation symbol, while the latter is fixed to ensure a high signal-to-noise ratio. It is worth noting that the receiving capability of the design in the current work was established by two important modifications to the PM. Firstly, we improved the feed network and used a Wilkinson power divider with high isolation. In the previous design, the feed network employed a T-type power divider, which made the received signals distorted due to serious interference between different channels. Secondly, we replaced the varactor with the PIN diodes at the feed network. This is necessary due to the limitations of the varactor, which requires high tunable voltages, complex control circuits, and large power supplies. Each element comprises three parts, including the radiation layer, the direct current bias layer, and the feed network layer, as depicted in Fig. 2a. The three parts are laminated together with two pieces of prepreg substrates. In the top radiation layer, dual PIN diodes bridge two rectangular metal patches and a small square patch. The direct current bias layer is located between the two grounding plates, creating an EM-shielded space and thus reducing the influence of the two bias lines. One bias line with the scalloped branch connects to the radiation layer, and the other links to the feed network layer. The feed network layer consists of a two-port reflection-type phase shifter that integrates two PIN diodes. The 2-bit phases (S1: − 90°, S2: 0°, S3: 90°, S4: 180°) are achieved by adjusting the voltages imposed on the PIN diodes. It is worth noting that the implementation of 2-bit phases does not need external phase shifters, thereby reducing the complexity and cost.

Figure 2b demonstrates the magnitude and phase responses of the meta-element. The magnitude responses of the four states are close to one in the wide frequency range of 10.7 - 11.7 GHz, and the phase difference of the four states remains almost 90° over the whole frequency band, exhibiting the broadband performance of the meta-element. More geometric parameters and operating principles of the element are detailed in our previous work[31].

## Algorithm

As the above-mentioned model, we consider the situation where a PM-based transmitter, located at the xoy plane and loaded with $N$ tunable elements, simultaneously serves $K$ single-antenna users by transmitting constellation symbols to each of them. The reference

constellation sets O, including 8PSK, 16QAM, and 64QAM, are encoded with Gray code to reduce the error bit rates (see Supplementary Note 4). The received signal at the $k$th user is

$$y_k = \mathbf{h}_k^H \mathbf{x} + n, k = 1, \ldots, K, \tag{1}$$

where $\mathbf{x} \in \mathbb{C}^N, n \sim \mathrm{CN}(0, \sigma^2)$, and $H$ are the coding sequences, the complex additive white Gaussian noise, and the conjugate transpose operator, respectively; and $\mathbf{h}_k^H \in \mathbb{C}^N$ is the channel vector

$$\mathbf{h_k} = e^{-j\beta d_k} \sqrt{G(\theta_k, \varphi_k)} \left[ e^{j\mathbf{v}_{1,1}^H \mathbf{u}}, \ldots, e^{j\mathbf{v}_{p,q}^H \mathbf{u}}, \ldots, e^{j\mathbf{v}_{N_x, N_y}^H \mathbf{u}} \right]^T, \tag{2}$$

where $d_k, \theta_k, \varphi_k$, and $G(\theta_k, \varphi_k)$ are the distance, the elevation angle, the azimuth angle, and the directional gain of the element with respect to the $k$th user, respectively. In addition, $\mathbf{v}_{p,q} = [pd_x, qd_y]^T$ and $\mathbf{u} = 2\pi/\lambda[\sin\theta_k \cos\varphi_k, \sin\theta_k \sin\varphi_k]^T$ are the auxiliary vectors to simplify Eq. (2). We clarify that although each meta-element can only provide finite possible discrete phases of $x_i$, their propagation distances to the user are different. As the scale of array increases, there will be many $h_i$ with different phases, and the role of $x_i$ is to combine those $h_i$ to provide the desired symbol at the $k$th user. We stress that this principle has been analyzed and demonstrated in ref. 32.

Our goal is to make the received signals $\mathbf{y} = [y_1, \ldots, y_K]^T$ as close to the reference constellation symbols $\mathbf{s} = [s_1, \ldots, s_K]^T \in O$ as possible. To this end, the coding sequences $\mathbf{x}$ of the PM are elaborately optimized to minimize signal variation. In this context, we formulate the optimization problem as

$$\begin{aligned} \min_{\mathbf{x}, \beta} \quad & \|\mathbf{s} - \beta \mathbf{H} \mathbf{x}\|_2^2 + K\beta^2\sigma^2 \\ \text{s. t.} \quad & x(i) \in X, i = 1, \ldots, N \end{aligned}, \tag{3}$$

where $\mathbf{H} = [\mathbf{h}_1^H, \mathbf{h}_1^H, \ldots, \mathbf{h}_K^H] \in \mathbb{C}^{K \times N}$ and $X = \{1/\sqrt{N}e^{jw_i} | w_i = 2\pi \cdot i/2^P, i = 1, 2, \ldots, 2^P\}$ are the channel matrix and the $P$-bit phase quantization set, respectively; and $\beta \in \mathbb{R}^+$ is the precoding factor to rescale the magnitude of the received signal $y_k$ to the same level as the constellation symbol $s_k$. In this work, we assume that the $\beta$ factor transmitted by the pilot technology is perfectly known to all users.

Problem (3) involves the widely recognized issue of finite resolution precoding in massive multi-input multi-output (MIMO) systems[33,34]. The problem is non-convex due to the discrete constraint. The problem can be firstly converted into a mixed integer linear program by using the special ordered set technique and then solved by using the branch-and-bound algorithm[35]. However, the computational

complexity is unaffordable for large-scale arrays. In this study, we employ the ADMM framework to address the problem. The framework adopts the idea of divide and conquer that effectively decomposes a non-convex problem into several convex problems and exhibits good performance in terms of fast convergence and stability[36,37]. By introducing an auxiliary variable, we rewrite the problem as

$$
\min_{\mathbf{x},\tilde{\mathbf{x}},\beta} \quad \|\mathbf{s} - \beta\mathbf{H}\mathbf{x}\|_2^2 + K\beta^2\sigma^2
$$
$$
\text{s.t.} \quad \tilde{\mathbf{x}} = \mathbf{x}, \tag{4}
$$
$$
\tilde{x}(i) \in X, i = 1, \ldots, N
$$

The augmented Lagrangian of the problem is

$$
L(\mathbf{x},\beta,\tilde{\mathbf{x}},\mathbf{u}) = \|\mathbf{s} - \beta\mathbf{H}\mathbf{x}\|_2^2 + K\beta^2\sigma^2 + \Re\{\mathbf{u}^H(\tilde{\mathbf{x}} - \mathbf{x})\} + \frac{1}{2}\rho\|\tilde{\mathbf{x}} - \mathbf{x}\|_2^2, \tag{5}
$$

where $\Re\{\cdot\}$, $\mathbf{u} \in \mathbb{C}^N$, and $\rho$ are the real-part operator, the dual vector, and the penalty parameter, respectively. By using the alternating optimization strategy, we decompose the problem as

$$
\tilde{\mathbf{x}}^{t+1} = \arg\min_{\tilde{\mathbf{x}}} L(\mathbf{x}^t,\beta^t,\tilde{\mathbf{x}},\mathbf{u}^t), \tag{6}
$$

**Table 1 | ADMM-Based coding sequences optimization**

| | |
|---|---|
| 1: | **Input:** $\mathbf{x}^{(0)}$, $\mathbf{u}^{(0)}$, $\sigma^2$, $\rho$, $T$ |
| 2: | Calculate the initial scaled factor $\beta^{(0)}$ using Eq. (12) |
| 3: | **For** $t = 0$ **To** $T$ |
| 4: | Update $\tilde{\mathbf{x}}^{t+1}$ using Eq. (10) |
| 5: | Update $\mathbf{x}^{t+1}$ using Eq. (11) |
| 6: | Update $\beta^{t+1}$ using Eq. (12) |
| 7: | Update $\mathbf{u}^t$ using Eq. (9) |
| 8: | $t = t + 1$ |
| 9: | **End** |
| 10: | **Output:** $\mathbf{x} = \mathbf{x}^T$ |

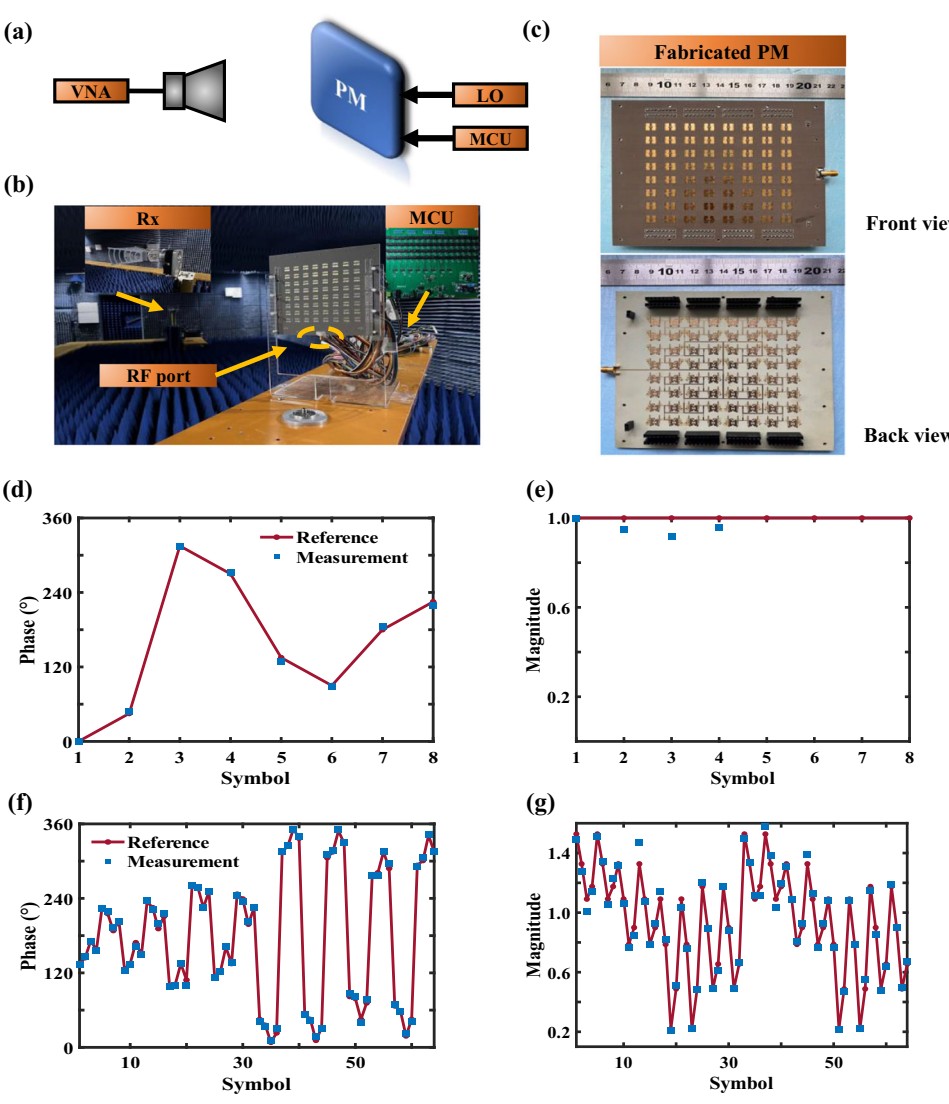

**Fig. 3 | Selected measurements of the single-channel mode. a–c** Experimental setup to verify the proposed DIM scheme. **a** The experiment configuration. **b** The measurement scene. **c** The photos of the fabricated PM. **d–g** The measured results of the single-channel DIM. **d** The phases and **e** magnitudes of the signals along $(\theta,\varphi) = (-19°, 0°)$ for the 8PSK modulation. **f** The phases and **g** magnitudes of the signals along $(\theta,\varphi) = (21.5°, 0°)$ for the 64QAM modulation.

$$\mathbf{x}^{t+1} = \arg\min_{\mathbf{x}} L(\mathbf{x}, \beta^t, \tilde{\mathbf{x}}^{t+1}, \mathbf{u}^t), \tag{7}$$

$$\beta^{t+1} = \arg\min_{\beta} L(\mathbf{x}^{t+1}, \beta, \tilde{\mathbf{x}}^{t+1}, \mathbf{u}^t), \tag{8}$$

$$\mathbf{u}^{t+1} = \mathbf{u}^t + \rho(\tilde{\mathbf{x}}^{t+1} - \mathbf{x}^{t+1}). \tag{9}$$

The problems (6), (7), and (8) have closed-form solutions, namely,

$$\tilde{\mathbf{x}}^{t+1} = \frac{1}{\sqrt{N}} \left[ e^{j\tilde{w}_1^{t+1}}, e^{j\tilde{w}_2^{t+1}}, \dots, e^{j\tilde{w}_N^{t+1}} \right]^T, \tag{10}$$

$$\mathbf{x}^{t+1} = \frac{1}{2} (\mathbf{R}_H^t)^{-1} \phi^{t+1}, \tag{11}$$

$$\beta = \left| \frac{\Re\{\mathbf{s}^H \mathbf{H} \mathbf{x}^{t+1}\}}{\|\mathbf{H}\mathbf{x}^{t+1}\|_2^2 + K\sigma^2} \right|, \tag{12}$$

where $\mathbf{R}_H^t = (\beta^t)^2 \mathbf{H}^H \mathbf{H} + \frac{\rho}{2} \mathbf{I}_N$, $\phi^{t+1} = 2\beta^t \mathbf{H}^H \mathbf{s} + \mathbf{u}^t + \rho\tilde{\mathbf{x}}^{t+1}$, and $|\cdot|$ are the absolute value operators. Moreover, $\tilde{w}_i^{t+1}$ is the optimized discrete phase of the $i$th element in the $t+1$ iteration. The detail is given in Supplementary Note 5. The optimization process is summarized in Table 1, and the detailed analysis of the computational complexity is also given in Supplementary Note 5.

It is worth noting that the discrete optimization algorithm remains available to larger-scale arrays owing to its excellent properties, including low complexity and fast convergence (see Supplementary Note 6 for more details). We also provide the compare-and-contrast study with other algorithms reported in the field of DIM, including the definition of the performance metrics and the discussion on the scalability to demonstrate the advances of our method, as shown in Supplementary Note 6.

## Experimental results

To demonstrate the PM-based DIM scheme, we conducted measurements on the fields radiated by a fabricated PM and compared the measured results with reference constellation diagrams. The experiments were conducted in a far-field anechoic chamber shown in Fig. 3a, b. The PM was excited by a sinusoidal carrier signal at the SMA port, and the coding sequences were imposed on PM via MCU (ST STM32F103C8T6). A horn antenna (HZ-90HA20NZJL) operating in 9.8 ~ 15.0 GHz was connected to the vector network analyzer (VNA, model Agilent N9010A) for signal detections. The fabricated PM is illustrated in Fig. 3c, which is composed of 8 × 8 elements with an overall size of 144 × 144 × 2.5 mm³. Moreover, it provides the far-field control capability, while keeping low cost and power consumption. All received signals were rescaled using the power normalization and the factor $\beta$, and a uniform phase was added to compensate for the propagation effect (see Supplementary Note 7 for more details).

To clearly demonstrate the working mechanism and measurements of our scheme, three representative experiments are presented in the main text, including the single-channel 8PSK, single-channel 64QAM, and dual-channel 16QAM modes. However, the capability of our scheme is not limited to dual channels, and the measurements for the four-channel DIM are demonstrated in Supplementary Note 8 (even more channels can be achieved if the scale of the metasurface increases). In the experiments, the PM was firstly configured to work in a single-channel mode, giving rise to 8PSK symbols to be transmitted in the desired direction $(\theta, \varphi) = (-19°, 0°)$. Figure 3d, e shows the phases and magnitudes of the measured signals, which are in excellent agreement with the reference symbols. Then, the PM was configured to give rise to 64QAM symbols to be transmitted in the desired direction $(\theta, \varphi) = (21.5°, 0°)$. Such a configuration is more difficult than the previous one since the modulation has a higher order, and the distances between adjacent symbols are closer. Figure 3f, g demonstrates the phases and

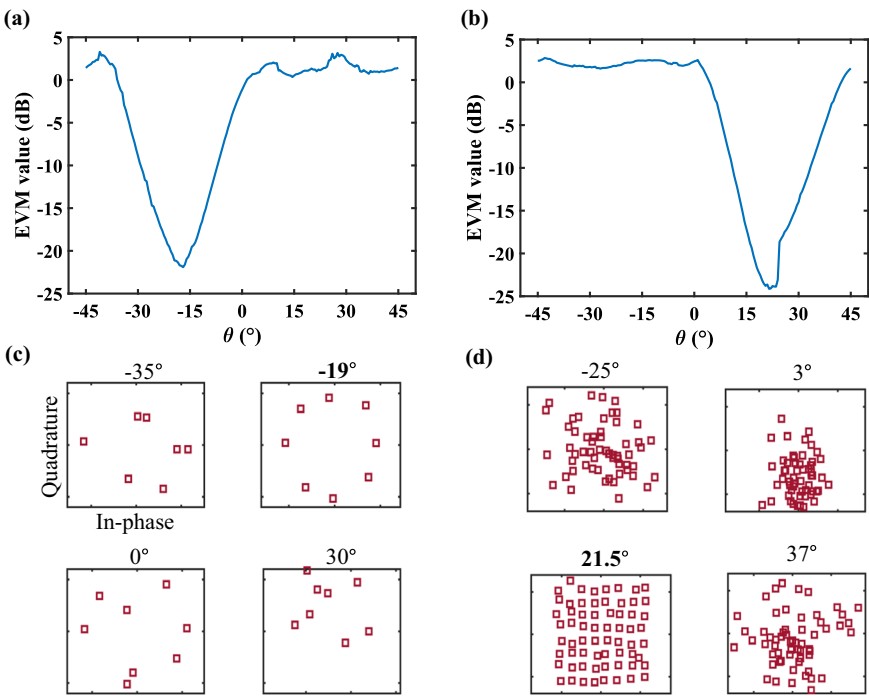

**Fig. 4 | The measured EVM values in the single-channel mode. a** The EVM distribution and **c** constellation diagrams for 8PSK modulation. **b** The EVM distribution and **d** constellation diagrams for 64QAM modulation.

magnitudes of the measured signals. The measured results exhibit good agreement with the reference symbols.

As shown in Fig. 4, we utilize the error vector magnitude (EVM) to quantify the performance of the directional security. The metric of EVM is more intuitive and commonly used in wireless communications to assess the quality of constellation diagrams. The EVM is defined at a given time as the vector difference between the measured and reference symbols: $EVM = \sqrt{\sum_{i=1}^{M}|\mathbf{s}_{ref}^{i} - \mathbf{s}_{mea}^{i}|^2 / \sum_{i=1}^{M}|\mathbf{s}_{ref}^{i}|^2}$, where $\mathbf{s}_{ref}$, $\mathbf{s}_{mea}$, and $M$ are the reference, measured signals, and the number of symbols, respectively. The lower the EVM value is, the closer the measured

symbols are to the reference. Figure 4a, c present the EVM distribution and signal structures of the 8PSK case, while Fig. 4b, d show the measured results of the 64QAM case. Clearly, the up-and-right subfigure of Fig. 4c and the left-and-down subfigure of Fig. 4d indicate that the measured signals in the desired directions are correct, while the other subfigures demonstrate that the signal structures in the unintended directions are wrong, which validates the directional security of the proposed method. We further propose the concept of a secure zone, defined by the value of EVM, to separate the desired users and eavesdroppers. The in-depth analyses are conducted on the minimum

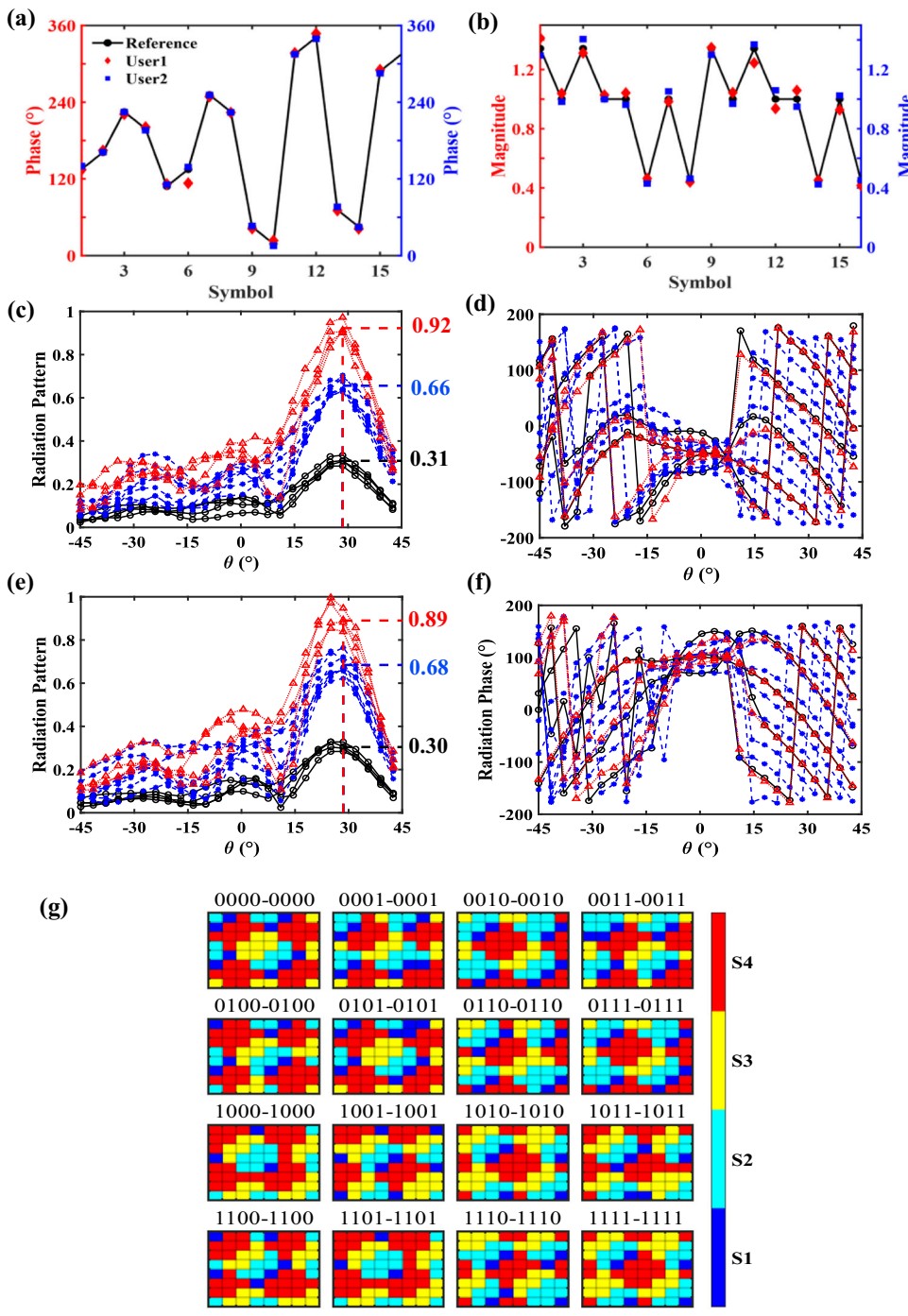

**Fig. 5 | Selected measurements in the dual-channel 16QAM scheme. a** The phases and **b** magnitudes of the signals in the target users. **c** The radiation patterns **d** and phases of fields in the $\varphi = 0°$ plane. **e** The radiation patterns **f** and phases of fields in the $\varphi = 90°$ plane. **g** The optimized coding sequences of the $8 \times 8$ PM that implements the dual-channel 16QAM. (S1: $-90°$, S2: $0°$, S3: $90°$, S4: $180°$).

angular width and the maximum number of supported channels (see Supplementary Note 13 for more details).

To further validate the 2D and dual-channel DIM capabilities of the proposed scheme, we then conducted experiments where the PM independently and simultaneously radiated the 16QAM symbols towards two users located respectively in the directions of $(\theta_1, \varphi_1) = (28.5°, 0°)$ and $(\theta_2, \varphi_2) = (28.5°, 90°)$. Figure 5a, b show the phases and magnitudes of the measured signals, which match well with the reference symbols. Figure 5c, d present the radiation powers and phases of the measured fields in the $\varphi = 0°$ plane associated with the 16QAM, respectively. The radiation powers form three clusters near the desired directions $\theta = 28.5°$, and the average powers of each cluster are presented on the right. From the above results, we clarify that the discrete optimization algorithm can effectively engineer the main-lobe beams to align with the desired directions. The main lobe powers have exhibited strong correlations with the magnitudes of the transmitted constellation symbols, addressing the effectiveness of energy allocation. The radiation phases of the measured fields are regularly arranged in the vicinity of the target direction. Figure 5e, f presents the results in the $\varphi = 90°$ plane, where we can observe similar features in the target direction. The above results demonstrate that the simultaneous dual-channel 16QAM is realized. Finally, the optimized digital coding sequences of the $8 \times 8$ PM that implements the dual-channel 16QAM are shown in Fig. 5g.

For a multi-user communication system, the evaluation of the crosstalk of different users is crucial because it determines whether the channels are independent. To obtain the complete cross-talk matrix (see "**Methods**" for the definition and calculation), we have conducted two experiments. Specifically, In the first group of experiments, the sixteen different symbols are sent to user 1, and the same symbols coded as "0000" are sent to user 2. In the second group of experiments, the sixteen different symbols are sent to user 2, and the same symbols coded as "0011" are sent to user 2. The magnitudes and phases of the measured fields are demonstrated in Supplementary Note 9, which is well consistent with the transmitted symbols. The measured constellation diagrams of each user are presented in Fig. 6a, b. Taking Fig. 6a as an illustrative example, the symbols received by user1 are in good agreement with the 16QAM, while the signals of user2 are clustered near the corresponding reference point. We then calculate the cross-talk matrix **C** according to Eq. (8), and the results are shown in Fig. 6c. The cross-talk values between different users are relatively low, and the maximum is 0.12 (−18.4 dB), implying acceptable performance under the dual-channel transmission. We remark that the transmission efficiency of different channels can be further improved by three methods: (1) increase the phase quantization level of the metasurface; (2) increase the scale of the metasurface to reduce the multi-user interference; and (3) employ an improved channel model to take the mutual coupling of the metasurface into account[38].

Figure 7a, c demonstrate the EVM distribution and signal structures in the $\varphi = 0°$ plane, while Fig. 7b, d show the measured results in the $\varphi = 90°$ plane. The EVM values in the vicinity of the target direction $\theta = 28.5°$ are relatively lower than in other directions. The left-and-down subfigures in Fig. 7c, d indicate that the measured signals in the desired directions are correct, while the other subfigures demonstrate that the signals in the undesired directions are distorted, which validates the directional security of the proposed method.

DIM has the advanced characteristics of providing a secure zone wherein the received signals exhibit a similar structure to the constellation diagrams, which improves the robustness of directional communication. To assess the secure zone, we measured the symbols in the vicinity of the desired directions. Figure 8a–d present the signal structures of the 8PSK case, the 64QAM case, the 16QAM case in the $\varphi = 0°$ plane, and the 16QAM case in the $\varphi = 90°$ plane, respectively. The received signals in different directions gather near the reference constellation symbols, which verifies the secure-zone feature of the proposed DIM scheme. Furthermore, we have also presented the distorted constellation diagrams in other directions as a reference to prove the directional security (see Supplementary Note 10).

To further verify the availability of the DIM at different frequencies, we also measured the symbols in the vicinity of the operating frequency (namely, 11 GHz). Figure 8e–h presents the signal structures of the 8PSK case, the 64QAM case, the 16QAM case in the $\varphi = 0°$ plane, and the 16QAM case in the $\varphi = 90°$ plane, respectively. The received signals are clustered near the reference constellation symbols in the broadband frequency range, indicating that the proposed DIM scheme supports next-generation high-throughput wireless communication.

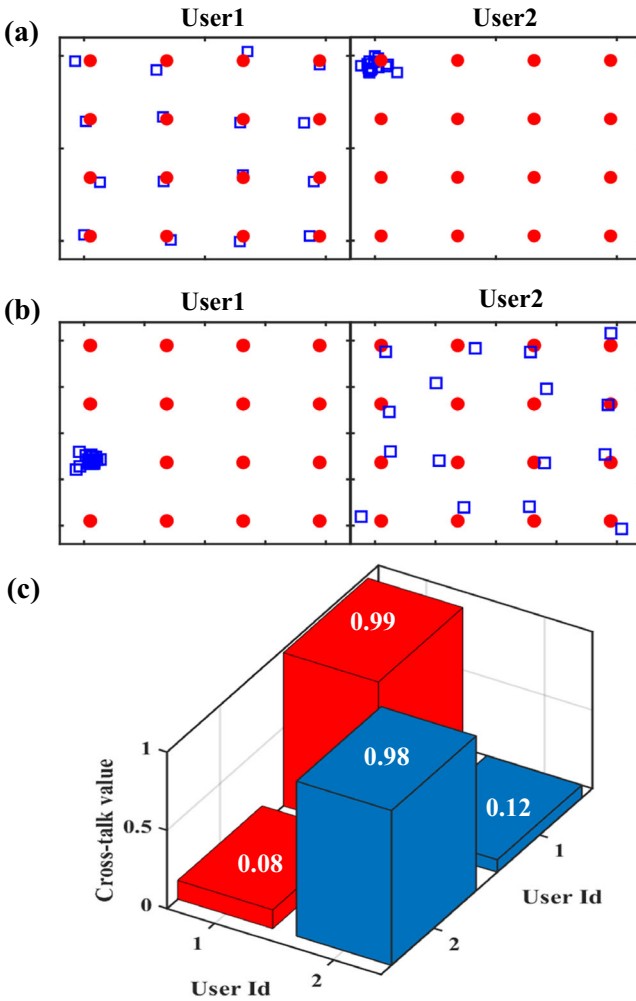

**Fig. 6 | Results for assessing the crosstalk in the dual-channel 16QAM. a, b** The measured constellation diagrams in the two groups of experiments, where the received and reference symbols are represented by the blue square and red circular markers, respectively. **a** The diagrams in the first group of experiments where sixteen different symbols are sent to user 1 and the same symbols coded as "0000" to user 2. **b** The diagrams in the second group of experiments where sixteen different symbols are sent to user 2 and the same symbols coded as "0011" to user 1. **c** The cross-talk values of dual users.

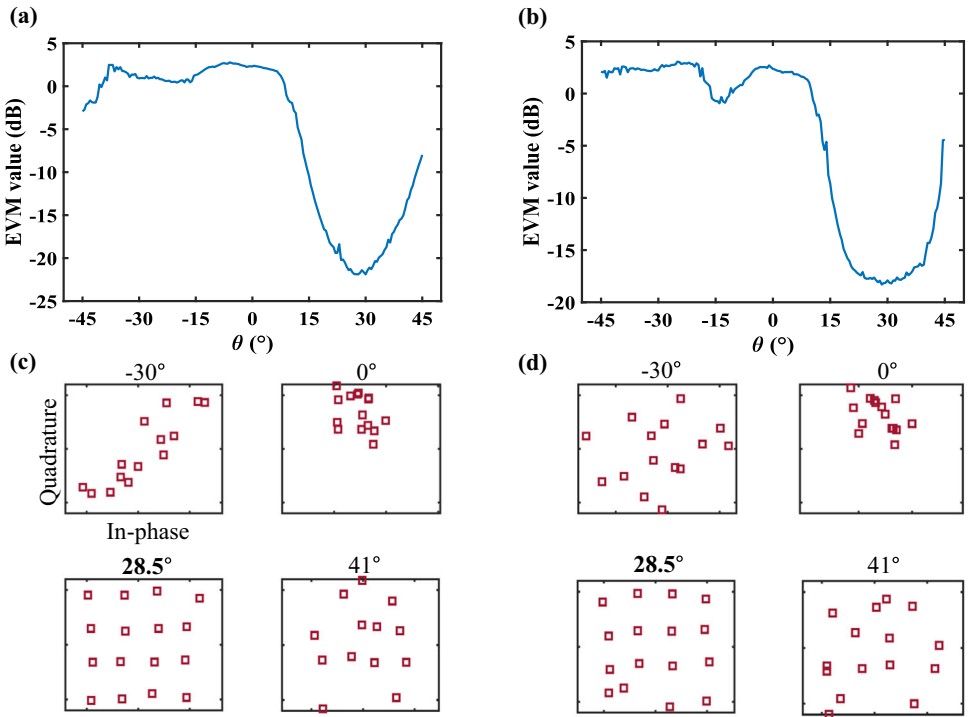

**Fig. 7 | The measured EVM values in the dual-channel 16QAM experiment. a** The EVM distribution and **c** constellation diagrams for user 1 in the $\varphi = 0°$ plane. **b** The EVM distribution and **d** diagrams for user 2 in the $\varphi = 90°$ plane.

## Discussion

We propose a novel scheme for 2D and high-order DIMs that supports multiple users. In the implementation, a 2-bit PM and a fast and efficient discrete optimization algorithm are used, which have attractive features of low complexity, serving as the transmitter or receiver and directional security against harmonics. The proposed scheme has been validated by measuring the 8PSK and 64QAM signals in the multi-channel modes. The measurement signals in the desired directions agree well with the reference symbols but are distorted in other directions, thereby demonstrating the secure communications of the proposed scheme. Moreover, the capability of DIM is available in a broadband frequency range, which will contribute to the next-generation high-throughput wireless communication systems. Further work will be focused on developing advanced systems for real-time communications and mitigating or overcoming the defects of DIM by exploiting positional modulations.

## Methods

### Calculation of multi-user crosstalk

To the best of the authors' knowledge, the definition and calculation method of multi-user crosstalk in the field of DIM is still absent. To fill up this gap, we try to quantify the crosstalk for multiple users in terms of a matrix **C** in which the matrix is composed of $K \times K$ elements $c_{ij}$ ($K$ is the number of users). We should first explain the motivations for calculating cross-talk values. In the field of electrical or optical communications, crosstalk is defined as the cross-power coupling between multiple channels[39]. When the input signal of one channel changes, receivers measure the variation of signals at the other channels. According to the difference values, the magnitudes of crosstalk are obtained. We here mimic the operation and perform the corresponding experiments. For calculating the element

$c_{ij}(i \neq j, j = 1, 2, …, i-1, i+1, …, K)$, we perform the experiment in which the transmitter simultaneously sends $M$ different symbols to user $i$ and the same symbols to the other users. The $c_{ij}(i \neq j)$ is

$$c_{ij} = \sqrt{\sum\nolimits_{m=1}^{M} |\boldsymbol{s}_{mea}^{j}(m) - \boldsymbol{s}_{ref}^{j}(m)|^2 / \sum\nolimits_{m=1}^{M} |\boldsymbol{s}_{ref}^{j}(m)|^2}, \quad (13)$$

where $\boldsymbol{s}_{mea}^{j}$ and $\boldsymbol{s}_{ref}^{j}$ are the measured and reference symbols of user $j$. Finally, the value of $c_{ii}$ is $\sqrt{1 - \sum_{j \neq i} c_{ij}^2}$.

### Detection of the direction of the user

Before directly transmitting digital information, the PM first operates in receiving mode and measures its relative position with respect to the desired user, which radiates incoming waves. The metasurface scans the whole space and generates sum, azimuth difference, and elevation difference beams, respectively, in three consecutive time slots by switching the coding sequences using MCU. The power of the sum beam $P_{\Sigma}(\theta, \varphi)$ reaches the maximal in the direction of the incoming wave. The power of the difference beam $P_{\Delta}(\theta, \varphi)$ reaches the minimum in the direction of the incoming wave. We define the magnitude ratio of the power of the sum- and difference- beams as

$$r(\theta, \varphi) = \frac{P_{\Sigma}(\theta, \varphi)}{P_{\Delta}(\theta, \varphi)}. \quad (14)$$

According to the physical meaning of $r(\theta, \varphi)$, the radio reaches the maximal in the direction of the incoming wave. Therefore, we can obtain the curve of $r(\theta, \varphi)$ and the estimated direction $(\theta, \varphi)$ of the incoming wave (see Supplementary Note 11 for more details).

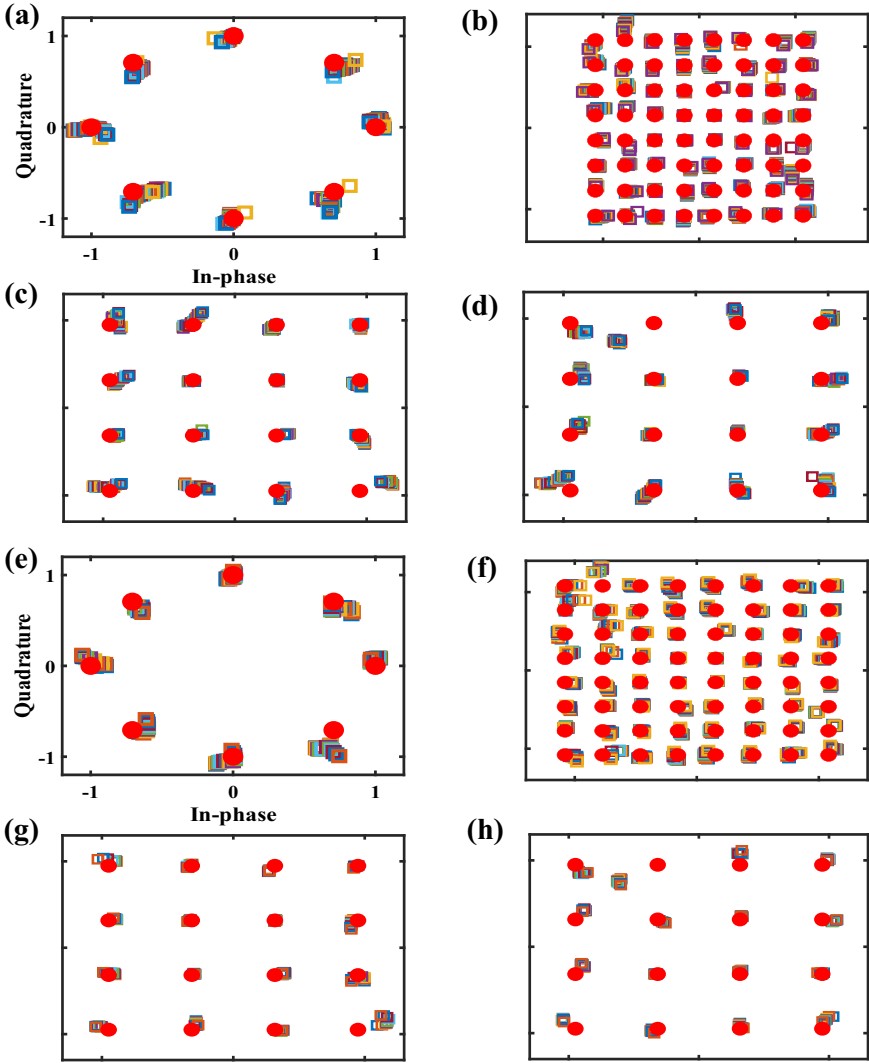

**Fig. 8 | Measured signal structures for validating the secure-zone characteristics and broadband performance of the proposed DIM scheme where the red circular marker represents the reference constellation symbols. a–d** Measured symbols that validate the feature of a secure zone. **a** The measured symbols of the single-channel 8PSK in the directions $(\theta, \varphi) = (-24.5° \sim -13.5°, 0°)$. **b** The measured symbols of the single-channel 64QAM in the directions $(\theta, \varphi) = (19° \sim 24.5°, 0°)$. **c** The measured symbols of 16QAM in the directions of $(\theta, \varphi) = (23.5° \sim 34°, 0°)$.

**d** The measured symbols of 16QAM in the directions of $(\theta, \varphi) = (23.5° \sim 34°, 90°)$. **e–h** Measured symbols that validate the broadband performance of the proposed DIM scheme. **e** The measured symbols of 8PSK with the frequency range 10.3 - 11.8 GHz. **f** The measured symbols of 64QAM with the frequency range 10.7 - 11.6 GHz. **g** The measured symbols of 16QAM in the direction $(\theta, \varphi) = (28.5°, 0°)$ with the frequency range 10.6 - 11.4 GHz. **h** The measured symbols of 16QAM in the direction $(\theta, \varphi) = (28.5°, 90°)$ with the frequency range 10.6 - 11.4 GHz.

## Data availability

The data that support the findings of this study are presented in the paper and Supplementary information file. We have improved the feed network and replaced the varactor with PIN diodes compared with our previous hardware design in ref. 31. The improvements enable the transmission and reception of digital information only in an identical device and decrease the complexity and energy consumption.

## Code availability

The codes that support the multi-channel DIM scheme are available from the corresponding author upon request.

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

## Acknowledgements

This work was supported by the National Key Research and Development Program of China (2021YFA1401002, J.W.W.), the National Natural Science Foundation of China (62288101, T. J.C.; 62171124, J.W.W; and 62225108, H.Q.Y), the Major Project of the Natural Science Foundation of Jiangsu Province (BK20212002, T.J.C.), the 111 Project (111-2-05, T.J.C.), the Fundamental Research Funds for the Central Universities (2242023k5002, J.W.W.), and the Jiangsu Provincial Innovation and Entrepreneurship Doctor Program (J.W.W.).

## Author contributions

T.J.C. and J.W.W. conceived the idea and revised the manuscript. H.X. developed the system model and algorithm and conducted the proof-of-principle experiments. Z.X.W. built the programmable metasurface prototype and conducted the experiments. R.W.S. and H.Q.Y. helped to process the measured data and revised the manuscript. All authors participated in the data analysis and read this paper.

## Competing interests

The authors declare no competing interests.
