## [Peer Review File · Nature Communications]

Two-dimensional and high-order directional information modulations for secure communications based on programmable metasurfaceREVIEWER COMMENTS

Reviewer #1 (Remarks to the Author):

In this paper, the authors propose a DIM scheme based on a 2D programmable metasurface (PM). A fast and efficient discrete optimization algorithm is developed to optimize the digital coding sequences, and the correct constellation symbols of different high-order modulations can be generated and transmitted in multi-directional beams. Although the results are interesting and timely, there are several major issues that the authors should take into account seriously before I can consider its suitability of publication. Firstly, the proposed DIM scheme is similar to the so-called backscatter communications which are already well known for decades (e.g., see [R1], [R2]), therefore as a novelty consideration the authors should explain whether there is any intrinsic difference between them. Secondly, in traditional wireless communication, either the base stations (or microcell) or the terminal devices need to transmit as well receive information simultaneously, while in the reported DIM scheme it can only work for transmitting signals. Therefore, different RF systems are still required for signal receiving which makes it more complicated and more difficult to maintain in one wireless communication system. Finally, the metasurface-assisted high-order modulations and multi-direction transmission are already realized by previous researches using time-domain digital coding metasurfaces [M. Z. Chen, W. Tang, J. Y. Dai, et al. Natl. Sci. Rev., 9(1), 2022; J. Y. Dai, W. Tang, L. X. Yang, IEEE Trans. Antennas Propag., 68(3), 2020; T. J. Cui, S. Liu, G. D. Bai, et al. Research, 2019, 2019]. Compared with these previous works, what are the unique features or noticeable performance improvements in this work?

Besides, I also found the following technical issues that the authors may carefully address before making a definite recommendation:

1. It seems unconvincing that the authors only apply dual-channel cases in the experiment to prove the multi-direction property of the proposed scheme. More cases about multi-direction transmission (eg. four-channel cases) measurements should be given in the article to prove the multi-channel property as described in Figure 4.
2. The authors apply an efficient discrete optimization algorithm to calculate the coding arrays of the metasurface. However, the authors only display the amplitude distributions of different far-field patterns in measurement while the simulated far-field patterns of different coding arrays are not mentioned in the article. The calculated results of the algorithm are not convincing enough since the phase properties of different coding arrays are unknown. The authors should add the amplitude and phase responses of different coding arrays in the article and at least give the simulated far-field patterns to prove the validity of the coding arrays.
3. In the article, the authors use Figure 9 to validate the secure-zone characteristic of the proposed DIM scheme. However, the depicted constellation diagrams are all measured in the coverage area of the generated beams. Distorted constellation diagrams in other directions should be added as a reference to prove the directional security.
4. The authors choose 2-bit quantization level to realize different high-order modulation schemes, such as 16 QAM and 64 QAM. Generally, coding arrays with 2-bit quantization are difficult to realize desired phase and amplitude distributions of high-order modulations since the 2-bit discrete phase distributions has accuracy limitations. However, the measured constellation diagrams of 64 QAM and 16 QAM are nearly the same with the ideal constellation symbols. More detailed analysis of the optimization process with a specific example (eg. 16QAM) is needed.
5. From the explosion view of the element in figure 2(a), the DC bias line is attached with the ground layer and the PIN diodes in feed network are sandwiched between ground layer and dielectric layer, which are unreasonable in the practical situation. The authors should check it carefully.
6. In Figure 1(c), the phase responses of the element keep the same in the frequency band from 10 GHz to 12 GHz, which seems to violate the dispersion effect. The authors should explain it in detail.
7. According to the authors, the direction (28.5° , 90°) is also the transmission channel. The far-field patterns should form three clusters at $\theta = 28.5^\circ$ as Figure 7(a). However, the far-field patterns in Figure 7(b) associated with 16QAM symbols in the $\phi = 90^\circ$ plane show the same power at $\theta = 28.5^\circ$. The author should explain this carefully.

8. What's the transmission rate of the DIM scheme, and what is the upper modulation frequency limit by using the PIN diodes?

9. There are some spelling mistakes in the article, such as "Figure 6d to 6d" at line 312, page 12 and "in the vicinity of" at line 367, page 16. The authors should carefully check the article for grammatical errors.

[R1] "Lev Termen's Great Seal bug analyzed," IEEE Trans. Aerosp. Electron. Syst., vol. 28, no. 11, Art. no. 11, 2013, doi: 10.1109/MAES.2013.6678486.

[R2] "Ambient Backscatter: Wireless Communication Out of Thin Air," ACM SIGCOMM Comput. Commun. Rev., vol. 43, no. 4, Art. no. 4, 2013, doi: doi: 10.1145/2534169.2486015.

Reviewer #2 (Remarks to the Author):

Thanks for submitting your manuscript to Nature Communications. I enjoyed reading your paper as the paper is easy to follow and well-written. However, I have several concerns regarding the novelty and significance of the proposed technique.

First, the idea of directional information modulation is not new. As acknowledged by the authors, DIM has been investigated heavily in the past in theory and in practice. There are many proposed architectures that support the transmission of DIM. The use of metasurfaces for such transmission is not new either. Indeed, this has been published in the past by the authors themselves.

Hence, both the concept and implementation of the idea are known in the literature. The delta in this work seems to be the optimization framework and algorithm to find the correct phase profile on the metasurface for multi-user support. However, the reviewer does not see that as a major and significant contribution for this prestigious journal.

The optimization framework and the algorithm are both straightforward. Further, the implementation is done for up to two simultaneous users. It is not clear how the performance scales with more users. Further, what are other baseline algorithms that can do the job? what are the relevant performance metrics and how does the proposed algorithm compare against the baseline (currently undefined)? The answers to these questions are currently missing.

Further, the authors should show the performance with many more users and with a larger metasurface to justify the scalability of their algorithm. Can each user see a different constellation? how much is the cross-talk?

Finally, the authors should investigate the harmonic transmission and show that their scheme is secure against harmonics.

Unfortunately, given the limited novelty of this work, the lack of extensive experiments, and the rich literature in this domain, I recommend rejecting the paper.

Reviewer #3 (Remarks to the Author):

A directional information modulation scheme is proposed with use of a programmable metasurface assisted by algorithms that optimize the digital coding sequences with information symbols being transmitted on multi-directional beams. Several types of modulations are tried and the BER is suppressed along certain directions. A prototype has been fabricated and measurements indicate that there are certain angles at which the transmissivity gets maximized for all the adopted symbols. The obtained constellation diagrams also validate the correct operation of the device and the proper distribution of power across the space of elevation angle.

The paper is well-written and deals with a very important topic. However, it cannot be published at such a prestigious publication venue as Nature Communications unless extensive improving modifications are performed. In particular:

(A) How the metasurface knows and the position of eavesdropper? What detection process is hidden behind? This information is critical for the efficient operation of the device since the directions of distorted constellation diagrams should be determined and taken into account by the programmable metasurface. The authors should clarify this issue further in a revised version of their manuscript.

(B) How the performance of the device can be enhanced in the presence of screens that increase the directivity along specific angles [1,2]? Can such components be used to further boost the operation of the described metasurface?

(C) The authors should provide a compare-and-contrast study with competing structures achieving security in communications with metasurfaces programmable [3] or not [4]. A comparison regarding the balance between the simplicity of the respective design and the effectiveness of the overall communication device.

[1] Lan et al., Real-time programmable metasurface for terahertz multifunctional wave front engineering, *Light*, 2023.

[2] Valagiannopoulos et al., Metasurface-enabled interference mitigation in visible light communication architectures, *Journal of Optics*, 2019.

[3] Saifullah et al., Dual-band multi-bit programmable reflective metasurface unit cell: design and experiment, *Optics Express*, 2021.

[4] Tsiftsis et al., Metasurface-Coated Devices: A New Paradigm for Energy-Efficient and Secure 6G Communications, *IEEE Vehicular Technology Magazine*, 2022.

Response Letter to Reviewers

We sincerely appreciate the constructive comments and suggestions from all reviewers, which help us improve the quality of the manuscript significantly. Below are our responses (in regular font) to each comment (*in italics and blue font*). We have also revised the manuscript and supplementary materials accordingly and **highlighted** all changes in the revised documents. We hope that the response letter has clarified the aspects concerned by the reviewers. We would be glad to respond to any further questions and comments. Thanks again for your possible consideration.

General comments from Reviewer #1

In this paper, the authors propose a DIM scheme based on a 2D programmable metasurface (PM). A fast and efficient discrete optimization algorithm is developed to optimize the digital coding sequences, and the correct constellation symbols of different high-order modulations can be generated and transmitted in multi-directional beams. Although the results are interesting and timely, there are several major issues that the authors should take into account seriously before I can consider its suitability of publication.

Response:

We thank the reviewer for the positive comments. The insightful comments are very constructive for further improvement of our work. We hope that the following responses are sufficient for highlighting the value of the work and dispelling your doubts.

Specific comments from Reviewer #1

Reviewer #1 – Comments 1:

Firstly, the proposed DIM scheme is similar to the so-called backscatter communications which are already well known for decades (e.g., see [R1], [R2]), therefore as a novelty consideration the authors should explain whether there is any intrinsic difference between them.

[R1] "Lev Termen's Great Seal bug analyzed," *IEEE Trans. Aerosp. Electron. Syst.*, vol. 28, no. 11, Art. no. 11, 2013, doi: 10.1109/MAES.2013.6678486.

[R2] "Ambient Backscatter: Wireless Communication Out of Thin Air," *ACM SIGCOMM Comput. Commun. Rev.*, vol. 43, no. 4, Art. no. 4, 2013, doi: doi: 10.1145/2534169.2486015.

Response:

We thank the reviewer for the important comments and bringing the interesting works to our attention. Backscatter communication has been a well-established domain for many years, dating back at least to the famous Great Seal Bug during World War II^[R1], and flourishes in the field of radio frequency identification (RFID). Backscatter communication modulates the dedicated emitted or ambient signals (e.g., ubiquitous commodity WIFI, TV, 5G signals, etc.) to transmit information^[R2-R5]. We note that the similarity between the backscatter communication and our scheme is that they both encode information on the radiation fields by adjusting the aperture impedance or coding sequences in real time. However, there are several important differences between the backscatter communication and our DIM scheme.

Firstly, our DIM scheme actively radiates the electromagnetic signals and has advanced wave-information manipulations. The conventional backscatter communication scheme is passive to the external environment, which exploits one or a few antennas to capture ambient waves and re-emit the modulated signals by adjusting the impedance. From the perspective of MIMO wireless communications, our DIM scheme provides many independently controllable antenna elements, and thus increases the signal-to-noise ratio (SNR) and supports massive spatial communication channels. The information capacity can be drastically improved compared to that of the traditional backscatter communication systems.

Secondly, it is our DIM scheme that enables the metasurfaces to directionally communicate with the desired users. The conventional backscatter communication scheme codes the information in the captured ambient waves using the time dimension and re-emits the signals to the whole space indiscriminately. The signals in different directions only vary on the power but their constellation structures are identical, thus posing inherent security risks when the eavesdropper is equipped

with a sensitive receiver. However, in our scheme, the information encoded by the metasurfaces can only be received within the main lobe beams, where the desired users are located. The constellation structures outside the main lobe beams are distorted, thus the eavesdropper cannot extract any information.

Thirdly, our DIM scheme can serve as a transmitter or receiver. The hardware design integrates the advantages of electronically programmable metasurfaces and the low profile of the microstrip antenna arrays. Thus, our device is reciprocal. The corresponding experimental demonstration is shown in the response to **Comment 2**. Compared with the backscatter communication scheme, our proposal does not need the extra receiver and decreases the complexity.

[R1] G. Brooker, J. Gomez, *IEEE Aerospace and Electronic Systems Magazine*. 2013, 28, 4.

[R2] V. Liu, A. Parks, V. Talla, S. Gollakota, D. Wetherall, J. R. Smith, *SIGCOMM Computer Communication Review*. 2013, 43, 39.

[R3] J. P. Niu, G. Y. Li, *Journal of Communications and Information Networks*. 2019, 4, 1.

[R4] J. Zhao, W. Gong, J. Liu, in *Proceedings of the 16th Annual International Conference on Mobile Systems, Applications, and Services, Association for Computing Machinery, Munich, Germany* 2018, 191.

[R5] C. Xu, L. Yang, P. Zhang, *IEEE Signal Processing Magazine*. 2018, 35, 16.

Reviewer #1 – Comments 2:

Secondly, in traditional wireless communication, either the base stations (or microcell) or the terminal devices need to transmit as well receive information simultaneously, while in the reported DIM scheme it can only work for transmitting signals. Therefore, different RF systems are still required for signal receiving which makes it more complicated and more difficult to maintain in one wireless communication system.

Response:

We thank the reviewer for the valuable comments. We should clarify that our DIM scheme can serve as a transmitter or receiver. Our hardware design takes the advantages of programmable metasurfaces and phased array antennas together. Therefore, our device is reciprocal.

To demonstrate the unique property, we made experiments with simultaneously transmitting and receiving signals using the same programmable metasurface. Figure R1.1 illustrates the experimental scene, in which the transmitter and receiver are equipped with the identical device. In the experiment, the transmitter radiates 8PSK constellation symbols to the desired direction $(\theta, \varphi) = (-17^\circ, 0^\circ)$. Figure R1.2a presents the magnitudes of measured fields for the single-channel 8PSK mode, in which the main lobe beams are located near the target direction $\theta = -17^\circ$, and the normalized radiation powers are larger than 0.8. As illustrated in Figure R1.2b, the phases of measured fields have an interval of nearly 45° in the target direction. The above results demonstrate that the single-channel 8PSK modulation is realized. We here should admit that the performance of the measurements is relatively poor compared to the case of utilizing a high-gain horn antenna as a transmitter. However, the problem can easily be mitigated by increasing the array scale.

As shown in Figure R1.3, we utilize the error vector magnitude (EVM) to quantify the performance of the directional secure transmission. The metric of EVM is more intuitive and commonly used in wireless communications to assess the quality of constellation diagrams. ^[R6] The EVM is defined at a given time as the vector difference between the measured and reference symbols: $EVM = \sqrt{\sum_{i=1}^M |\mathbf{s}_{ref}^i - \mathbf{s}_{mea}^i|^2 / \sum_{i=1}^M |\mathbf{s}_{ref}^i|^2}$, where \mathbf{s}_{ref} , \mathbf{s}_{mea} , and M are the reference, measured signals, and the number of symbols, respectively. The lower the EVM value is, the closer the measured symbols are to the reference.

We observe that the EVM values in the vicinity of the target directions are relatively lower. At the bottom of Figure R1.3, the constellation diagrams in different directions are presented. The diagram in the target direction matches well with the reference, while the others are distorted, which validates the security property of the proposed method. The above measurements demonstrate that DIM can be implemented by simultaneously exploiting the programmable metasurfaces as the transmitter and receiver.

Changes:

We have added the following sentences to clarify the unique feature that our design can serve as a transmitter or receiver.

- Lines 85~87, page 3: Firstly, they only transmit signals. This is incompatible with the operation of traditional wireless systems, where the base station or terminal device can both transmit and receive information.
- Lines 138~140, page 5: Furthermore, the hardware can also serve as a transmitter or receiver and is compatible with the conventional communication system (see Supplementary Note 3 for more details).

Furthermore, we have added the measurements for validation of the reciprocity of our DIM scheme to Supplementary Note 3.

Figure R1.1. Experimental scene for validating that our scheme can serve as a transmitter or receiver.

Figure R1.2. The measured fields for the 8PSK scheme where the transmitter and receiver are implemented using the same programmable metasurface. (a) The magnitudes of measured fields. (b) The phases of measured fields.

Figure R1.3. The EVM for validating the security of DIM for the 8PSK scheme where the transmitter and receiver are implemented using the same programmable metasurface. The bottom presents three constellation diagrams at different elevation angles.

[R6] S. Forestier, P. Bouysse, R. Quere, A. Mallet, J. M. Nebus, L. Lapierre, *IEEE Transactions on Microwave Theory and Techniques*. 2004, 52, 1132.

Reviewer #1 – Comments 3:

Finally, the metasurface-assisted high-order modulations and multi-direction transmission are already realized by previous researches using time-domain digital coding metasurfaces [M. Z. Chen, W. Tang, J. Y. Dai, et al. *Natl. Sci. Rev.*, 9(1), 2022; J. Y. Dai, W. Tang, L. X. Yang, *IEEE Trans. Antennas Propag.*, 68(3), 2020; T. J. Cui, S. Liu, G. D. Bai, et al. *Research*, 2019, 2019]. Compared with these previous works, what are the unique features or noticeable performance improvements in this work?

Response:

We thank the reviewer for the valuable questions about the previous papers. Our implementations are different and have two distinguished advantages compared with the mentioned works. On the one hand, our hardware can serve as a transmitter or receiver with a low profile and thus it is compatible with conventional communication systems. On the other hand, our hardware is directional secure against harmonic using the non-periodic optimized time-space coding sequences. Below we focus on the comparison of the methods to realize high-order modulation.

For the time-domain digital coding metasurfaces, the realization of high-order modulations is to optimize the periodic waveform, including the duty cycle and delay time of the pulse, to change the magnitude and phase of the harmonic, respectively. The method results in spectra pollution and is unable to ensure directional secure transmission due to the lack of space coding.

However, our proposed DIM scheme is based on time-space-domain digital coding. The coding sequences are optimized using the discrete optimization algorithm, which supports the manipulation of multiple spatial beams and temporal information with directional security. More importantly, the optimized coding sequences are not periodic, which means that the system does not generate the harmonic. The received signals in the harmonic frequency are noise, which ensures security against the harmonic.

Proof: The received signal at moment t is expressed as,

$$E(\theta_k, \varphi_k; t) = e^{j2\pi f_0 t} \sum_{p=1}^P \sum_{q=1}^Q \sqrt{G(\theta_k, \varphi_k)} x_{p,q}(t) e^{j\mathbf{v}_{p,q}^H \mathbf{u}} + n_k, \tag{R1}$$

where f_0 , θ_k , φ_k , $G(\theta_k, \varphi_k)$, $x_{p,q}$, and n_k are the central frequency, the elevation angle, the azimuth angle, the directional gain, the optimized phase response of the element, and the noise with respect to the k th user, respectively. In addition, $\mathbf{v}_{p,q} = [pd_x, qd_y]^T$ and $\mathbf{u} = 2\pi/\lambda[\sin\theta_k \cos\varphi_k, \sin\theta_k \sin\varphi_k]^T$ are the auxiliary vectors to simplify Equation (R1).

For the K -bit phase quantization, the response $x_{p,q}(t)$ at moment t is expressed as,

$$x_{p,q}(t) = \sum_{i=0}^{\infty} a_{p,q}^i \Gamma(t - iT), \quad (\text{R2})$$

where $a_{p,q}^i \in \{1/\sqrt{N} e^{jw_m} \mid w_i = 2\pi \cdot m/2^K, m = 1, 2, \dots, 2^K\}$ is the optimized discrete response of the (p, q) element for generating the desired field in the target direction. The optimized coding sequences are first stored in the FPGA and switched every passing T gap. The rectangular pulse signal $\Gamma(t)$ is

$$\Gamma(t) = \begin{cases} 1, & 0 \leq t \leq T \\ 0, & \text{otherwise} \end{cases}. \quad (\text{R3})$$

We perform the Fourier transform of Equation (R1) and obtain the spectral response of the received signal, namely,

$$E(\theta_k, \varphi_k; f) = \sum_{p=1}^P \sum_{q=1}^Q \sqrt{G(\theta_k, \varphi_k)} x_{p,q}(f - f_0) e^{j\mathbf{v}_{p,q}^H \mathbf{u}} + n_k(f). \quad (\text{R4})$$

In the above equation, the spectral response of the response $x_{p,q}(t)$ is

$$x_{p,q}(f) = \sum_{i=0}^{\infty} a_{p,q}^i \Gamma(f) e^{-j2\pi f iT}, \quad (\text{R5})$$

where the spectral response of the pulse signal $\Gamma(t)$ is

$$\Gamma(f) = \int_{-\infty}^{\infty} \Gamma(t) e^{-j2\pi ft} dt = \frac{\sin(\pi f T)}{\pi f}. \quad (\text{R6})$$

It is noticed that the spectral response of the received signal at the harmonic frequency $f_0 + nf_T$ ($f_T = 1/T$) is

$$\begin{aligned}
E(\theta_k, \varphi_k; f_0 + nf_T) &= \sum_{p=1}^P \sum_{q=1}^Q \sqrt{G(\theta_k, \varphi_k)} x_{p,q}(nf_T) e^{jv_{p,q}^H} + n_k(f + nf_T) \\
&= \sum_{p=1}^P \sum_{q=1}^Q \sqrt{G(\theta_k, \varphi_k)} e^{jv_{p,q}^H} \sum_{i=0}^{\infty} a_{p,q}^i \Gamma(nf_T) e^{-j2\pi nif_T T} \\
&\quad + n_k(f + nf_T) \quad , \quad n \neq 0. \quad (\text{R7}) \\
&= \sum_{p=1}^P \sum_{q=1}^Q \sqrt{G(\theta_k, \varphi_k)} e^{jv_{p,q}^H} \sum_{i=0}^{\infty} a_{p,q}^i \frac{\sin(n\pi f_T T)}{n\pi f_T} e^{-j2\pi nif_T T} \\
&\quad + n_k(f + nf_T) \\
&= n_k(f + nf_T)
\end{aligned}$$

It is worth emphasizing that the value of the harmonic frequency is determined according to the definition of the time-modulated array or time-coding metasurface. ^[R7] Equation (R7) indicates that the strength of the received signal in the target direction at the harmonic frequency $f_0 + nf_T$ is noise, which demonstrates that our proposal is secure transmission against harmonic.

To demonstrate the unique secure feature against harmonic, we perform simulations of the dual-channel 8PSK configuration. The two users are at $(\theta, \varphi) = (40^\circ, 0^\circ)$ and $(\theta, \varphi) = (25^\circ, 90^\circ)$, respectively. The switching time of FPGA is 10 us (i.e., the frequency offset is $f_T = 100$ KHz in according with the above definition) and the signal length is 5 ms. The received signals are calculated using Equation (R4) and normalized, and the SNR is 15 dB.

As shown in Figure R1.4, the received signals of user 1 at the central frequency f_0 have the characteristic of the 8PSK symbols, in which the magnitudes fluctuate around 1 and the phases are distributed in the vicinity of the eight discrete values (i.e., $-180^\circ, -135^\circ, -90^\circ, -45^\circ, 0^\circ, 45^\circ, 90^\circ$, and 135°). The signals received by user 2 are demonstrated in Figure R1.5, in which the magnitude values are relatively lower than those of user 1 and fluctuate between 0.6~1. However, the phase values have a form similar to the 8PSK symbols, which implies that the fluctuations of magnitude do not affect the signal recovery. Furthermore, we present the received signals at harmonic $f_0 + f_T$ and $f_0 + 2f_T$ of user 1 in Figures R1.6a and R1.6b, in which the magnitude values are low and the phase distributions are disorganized. The above results demonstrate that

the received signals at harmonic are mainly the noise and the information is unable to be transmitted.

Figure R1.4. The signals received by user 1 with about 5 ms. The user 1 is located at $(\theta, \varphi) = (40^\circ, 0^\circ)$ and the frequency is f_0 . The signals within 0.85-2.15 ms are enlarged on the right.

Figure R1.5. The signals received by user 2 with about 5 ms. The user 2 is located at $(\theta, \varphi) = (25^\circ, 90^\circ)$ and the frequency is f_0 . The signals within 0.85-2.15 ms are enlarged on the right.

Figure R1.6. (a) The signals received by user 1 within 5 ms and the signal frequency is $f_0 + f_T$. Further, the signals within 0.85-2.15 ms are enlarged on the right. (b) The signals at harmonic $f_0 + 2f_T$.

Changes:

We have added the following sentences to clarify the unique feature of directional security against harmonic.

- Lines 101~103, pages 3~4: **Using the optimized non-periodic time-space coding sequences, our scheme is directional secure against harmonic (see Supplementary Note 2 for more details).**

Furthermore, we have added the proof and simulations about the security against harmonic Supplementary Note 2.

[R7] J. Zhao, X. Yang, J. Y. Dai, Q. Cheng, X. Li, N. H. Qi, J. C. Ke, G. D. Bai, S. Liu, S. Jin, A.

Reviewer #1 – Comments 4:

It seems unconvincing that the authors only apply dual-channel cases in the experiment to prove the multi-direction property of the proposed scheme. More cases about multi-direction transmission (eg. four-channel cases) measurements should be given in the article to prove the multi-channel property as described in Figure 4.

Response:

We also believe that the experiment validation to support a greater number of channel direction transmission is crucial to improve the quality and advance of our work. We are greatly encouraged by the reviewer's suggestion.

As an illustrative example, we conducted new experiments with up to four simultaneous users, whose locations are $(\theta, \varphi) = (12^\circ, 0^\circ)$, $(\theta, \varphi) = (-30^\circ, 0^\circ)$, $(\theta, \varphi) = (42^\circ, 90^\circ)$, and $(\theta, \varphi) = (-34^\circ, 90^\circ)$, respectively. The performance of DIM for more users is verified using the 8PSK, 16QAM, and 64QAM schemes, respectively. We here briefly summarize the data and figures for each scheme. For the 8PSK scheme, we present the magnitudes and phases of measured fields, the received constellation diagrams of each user, the optimized coding sequences of the 8×8 programmable metasurface, and the EVM distribution over the elevation angles for validating the directional security. For the 16QAM scheme, apart from the above data, we also perform the cross-talk measurements and draw the corresponding matrix values. For the 64QAM scheme, we only present the received constellation diagrams of each user and the EVM distribution for the consideration of simplicity and clarity.

1. The results of four-channel 8PSK scheme

Figures R1.7a and R1.7b show the magnitudes and phases of the measured fields for the 8PSK scheme along the $\varphi = 0^\circ$ plane. In this case, the main lobe beams are located near the target directions $\theta = 12^\circ$ and $\theta = -30^\circ$, respectively, and the radiation powers are almost equal for

all symbols. Another interesting observation is that the phases of measured fields have an interval of nearly 45° in the vicinity of the target directions. Figures R1.7c and R1.7d show the results along the $\varphi = 90^\circ$ plane, where we observe the similar features in the target directions $\theta = 42^\circ$ and $\theta = -34^\circ$. The above results demonstrate that the simultaneous four-channel 8PSK modulation is realized with the help of our DIM system. More intuitionistic evidence is shown in Figure R1.8, in which the measured symbols are in excellent agreement with the references. The optimized digital coding distributions of the 8×8 PM that implements the four-channel 8PSK modulation are demonstrated in Figure R1.9.

The EVM distribution as the functions of the elevation angles for the 8PSK scheme is demonstrated in Figure R1.10. The EVM values in the vicinity of the target directions are relatively lower, which validates the security property of the proposed method. We should explain the observation that the EVM values are low near the direction $\theta = 0^\circ$ along the $\varphi = 90^\circ$ plane shown in Figure R1.10b. The reason behind this is that the measured fields in this part are also within the secure zone of user 1 (i.e., in the vicinity of the direction $(\theta, \varphi) = (12^\circ, 0^\circ)$). We also present the measured constellation in the unintended directions shown in the bottom subfigures, in which the signal structures are distorted.

Figure R1.7. The measured fields in the four-channel 8PSK experiment. (a)-(b) The results in the $\varphi = 0^\circ$ plane. (c)-(d) The results in the $\varphi = 90^\circ$ plane.

Figure R1.8. The measured constellation diagrams of the four-channel 8PSK experiment. The measured and

the reference symbols are represented by the blue square and the red circular markers, respectively. (a) The diagrams of user 1 located at $(\theta, \varphi) = (12^\circ, 0^\circ)$. (b) The diagrams of user 2 located at $(\theta, \varphi) = (-30^\circ, 0^\circ)$. (c) The diagrams of user 3 located at $(\theta, \varphi) = (42^\circ, 90^\circ)$. (d) The diagrams of user 4 located at $(\theta, \varphi) = (-34^\circ, 90^\circ)$.

Figure R1.9. The optimized coding distributions of the 8×8 programmable metasurface that implement four-channel 8PSK. (S1: -90° , S2: 0° , S3: 90° , S4: 180°)

Figure R1.10. The measured EVM values in the four-channel 8PSK experiment. (a) The EVM distributions in the $\varphi = 0^\circ$ plane. The bottom subfigures present two constellation diagrams that deviate from the desired users. (b) The EVM distributions along the $\varphi = 90^\circ$ plane. The bottom subfigures present two constellation diagrams that deviate from the desired users.

2. The results of four-channel 16QAM scheme

Figures R1.11a and R1.11b present the magnitudes and phases of the measured fields along the $\varphi = 0^\circ$ plane associated with the 16QAM, respectively. As illustrated in Figure R1.11a, the radiation patterns form three clusters near the desired directions $\theta = 12^\circ$ and $\theta = -30^\circ$, and the average powers of each cluster are presented. The radiation phases of measured fields are regularly arranged in the vicinity of the target directions. Figures R1.11c and R1.11d show the results along the $\varphi = 90^\circ$ plane, where we observe the similar features in the target directions $\theta = 42^\circ$ and $\theta = -34^\circ$. The above results demonstrate that the simultaneous four-channel 16QAM is realized. More intuitionistic evidence is shown in Figure R1.12, in which the measured symbols are in good agreement with the reference.

The EVM distribution as the functions of the elevation angles is demonstrated in Figure R1.13. The EVM values in the vicinity of the target directions are relatively lower than in other directions, which validates the security property of the proposed method.

For a multi-user communication system, the evaluation of cross-talk value is crucial because it determines whether the channels are independent. However, to the best of authors' knowledge, the definition and calculation method of multi-user crosstalk in the field of DIM are still absent. For filling up this gap, we try to quantify the crosstalk for multiple users in terms of a matrix \mathbf{C} in which the matrix is composed of $K \times K$ elements c_{ij} (K is the number of users). We should first explain the motivations for calculating cross-talk values. In the field of electrical or optical communications, the crosstalk is defined as the cross-power coupling between multiple channels. [R8] When the input signal of one channel changes, receivers measure the variation of signals at the other channels. According to the difference values, the magnitudes of crosstalk are obtained. We here mimic the operation and perform the corresponding the experiments. For calculating the element c_{ij} ($i \neq j, j = 1, 2, \dots, i-1, i+1, \dots, K$), we perform the experiment that the transmitter simultaneously sends M different symbols to user i and the M same symbols to the other users. The c_{ij} ($i \neq j$) is

$$c_{ij} = \sqrt{\sum_{m=1}^M |s_{mea}^j(m) - s_{ref}^j(m)|^2 / \sum_{m=1}^M |s_{ref}^j(m)|^2}, \quad (R8)$$

where s_{mea}^j and s_{ref}^j are the measured and reference symbols of user j . Finally, the value of

$$c_{ii} \text{ is } \sqrt{1 - \sum_{j \neq i} c_{ij}^2}.$$

To measure the crosstalk values of four users using the proposed method, we conducted four independent experiments. In the first group of experiments, the sixteen different symbols are sent to user 1, and the same symbols coded as “0011” are sent to user 2, and the same symbols coded as “1001” are sent to user 3, and the same symbols coded as “1100” are sent to user 4. In the second group of experiments, the sixteen different symbols are sent to user 2, and the same symbols coded as “1011” are sent to user 1, and the same symbols coded as “0110” are sent to user 3, and the same symbols coded as “1110” are sent to user 4. In the third group of experiments, the sixteen different symbols are sent to user 3, and the same symbols coded as “1100” are sent to user 1, and the same symbols coded as “0011” are sent to user 2, and the same symbols coded as “1110” are sent to user 4. In the fourth group of experiments, the sixteen different symbols are sent to user 4, and the same symbols coded as “1011” are sent to user 1, and the same symbols coded as “0001” are sent to user 2, and the same symbols coded as “0011” are sent to user 3.

The measurements are presented in Figures R1.14a – R1.14d. Taking Figure R1.14a as an illustrative example, we observe that the symbols received by user 1 are in good agreement with the 16QAM, while the signals of user 2, 3, and 4 are clustered near the corresponding reference point. The results demonstrate that our system has good crosstalk resistance. We then calculate the whole cross-talk matrix \mathbf{C} according to the above definition of c_{ij} . As show in Figure R1.15, the cross-talk values between difference users are relatively low and the maximum is 0.24 (-12.4 dB), implying the acceptable performance under the four-channel transmission.

$$\mathbf{C} = \begin{bmatrix} c_{11} & c_{12} & c_{13} & c_{14} \\ c_{21} & c_{22} & c_{23} & c_{24} \\ c_{31} & c_{32} & c_{33} & c_{34} \\ c_{41} & c_{42} & c_{43} & c_{44} \end{bmatrix} = \begin{bmatrix} 0.95 & 0.13 & 0.23 & 0.13 \\ 0.15 & 0.97 & 0.13 & 0.13 \\ 0.23 & 0.18 & 0.95 & 0.13 \\ 0.24 & 0.17 & 0.19 & 0.94 \end{bmatrix}$$

Figure R1.11. The measured fields in the four-channel 16QAM scheme. (a)-(b) The results in the $\varphi = 0^\circ$ plane. (c)-(d) The results in the $\varphi = 90^\circ$ plane.

Figure R1.12. The measured constellation diagrams of the four-channel 16QAM experiment. The measured and the reference symbols are represented by the blue square and the red circular markers, respectively. (a) The diagrams of user 1 located at $(\theta, \varphi) = (12^\circ, 0^\circ)$. (b) The diagrams of user 2 located at $(\theta, \varphi) = (-30^\circ, 0^\circ)$. (c) The diagrams of user 3 located at $(\theta, \varphi) = (42^\circ, 90^\circ)$. (d) The diagrams of user 4 located at $(\theta, \varphi) = (-34^\circ, 90^\circ)$.

Figure R1.13. The measured EVM values in the four-channel 16QAM experiment. (a) The EVM distribution in the $\varphi = 0^\circ$ plane. The bottom subfigures present two constellation diagrams that deviate from the desired users. (b) The EVM distribution in the $\varphi = 90^\circ$ plane. The bottom subfigures present two constellation diagrams that deviate from the desired users.

Figure R1.14. The measured constellation diagrams of each user in the experiments that measure the cross-talk values of multiple users. The measured and the reference symbols are represented by the blue square and the red circular markers, respectively. (a) The diagrams of the first group of experiments. (b) The diagrams of the second group of experiments. (c) The diagrams of the third group of experiments. (d) The diagrams of the fourth group of experiments.

Figure R1.15. The cross-talk values of four users.

3. The results of four-channel 64QAM scheme

We present the measurements of four-channel 64QAM scheme. Such configuration is more difficult than the previous experiments since the modulation has higher order, and the distances between adjacent symbols are closer. Figure R1.16 shows the measured symbols of each user, in which the results of user 3 and 4 are relatively better than user 1 and 2. The EVM distribution as the functions of the elevation angles is given in Figure R1.17. The EVM values in the vicinity of the target directions are relatively lower than in other directions, which validates the security property of the proposed method.

Figure R1.16. The received constellation diagrams in the four-channel 64QAM experiment. The measured and the reference symbols are represented by the blue square and the red circular markers, respectively. (a) The diagrams of user 1 located at $(\theta, \varphi) = (12^\circ, 0^\circ)$. (b) The diagrams of user 2 located at $(\theta, \varphi) = (-30^\circ, 0^\circ)$. (c) The diagrams of user 3 located at $(\theta, \varphi) = (42^\circ, 90^\circ)$. (d) The diagrams of user 4 located at $(\theta, \varphi) = (-34^\circ, 90^\circ)$.

Figure R1.17. The EVM values as the functions of the elevation angles in the four-channel 64QAM scheme.

(a) The EVM distribution in the $\varphi = 0^\circ$ plane. The bottom subfigures present two constellation diagrams that deviate from the desired users. (b) The EVM distribution in the $\varphi = 90^\circ$ plane. The bottom subfigures present two constellation diagrams that deviate from the desired users.

Changes:

We have added the following sentences to clarify that our scheme can support four-channel transmissions.

- Lines 241~244, page 9: **However, the capability of our scheme is not limited to dual channels, and the measurements for the four-channel DIM are demonstrated in Supplementary Note 8 (even more channels can be achieved if the scale of metasurface increases).**

Furthermore, we have added the measurements for validation of the reciprocity of our DIM scheme to Supplementary Note 8.

[R8] S. SeyedinNavadeh, M. Milanizadeh, F. Zanetto, G. Ferrari, M. Sampietro, M. Sorel, D. A. B. Miller, A. Melloni, F. Morichetti, *Nature Photonics*. **2023**.

Reviewer #1 – Comments 5:

The authors apply an efficient discrete optimization algorithm to calculate the coding arrays of the metasurface. However, the authors only display the amplitude distributions of different far-field patterns in measurement while the simulated far-field patterns of different coding arrays are not mentioned in the article. The calculated results of the algorithm are not convincing enough since the phase properties of different coding arrays are unknown. The authors should add the amplitude and phase responses of different coding arrays in the article and at least give the simulated far-field patterns to prove the validity of the coding arrays.

Response:

We thank the reviewer for the valuable suggestions. We have added the radiation phase distributions of different coding arrays for each modulation scheme. Furthermore, the simulated fields of the single-channel 8PSK scheme have also been added for comparison with the

measurements.

Figures R1.18a and R1.18b demonstrate the magnitudes and phases of the measured fields for the single-channel 8PSK scheme, respectively, in which the main lobe beams are along the target direction $\theta = -19^\circ$ and the radiation powers are larger than 0.8 for all symbols. Another interesting observation is that the phases of the measured fields are distributed with nearly 45° interval in the vicinity of the target direction. Figures R1.18c and R1.18d show the magnitudes and phases of simulated fields, respectively, which demonstrate similar features to the measurements. The magnitudes and phases of measured fields for the single-channel 64QAM scheme are shown in Figures R1.19a and R1.19b. The curves are quite dense, but we can still observe that the main lobe beams with different magnitudes are generated and radiation phases are regularly arranged in the vicinity of the target direction $\theta = 21.5^\circ$, which matches with the characteristics of 64QAM constellation symbols.

Figures R1.20a and R1.20b present the magnitudes and phases of measured fields in the $\varphi = 0^\circ$ plane for the dual-channel 16QAM scheme, respectively. As illustrated in Figure R1.20a, the radiation patterns form three clusters near the target direction $\theta = 28.5^\circ$. The radiation phases of measured fields are regularly arranged in the vicinity of the target direction. Figures R1.20c and R1.20d show the results in the $\varphi = 90^\circ$ plane, where we can observe the similar features in the target direction $\theta = 28.5^\circ$.

Figure R1.18. The fields in the single-channel 8PSK experiment. The measured (a) magnitudes and (b) phases of fields. The simulated (c) magnitudes and (d) phases of fields.

Figure R1.19. The fields in the single-channel 64QAM experiment. The measured (a) magnitudes and (b) phases of fields.

Figure R1.20. The fields in the dual-channel 16QAM experiment. (a)-(b) The measurements in the $\varphi = 0^\circ$ plane. (c)-(d) The measurements in the $\varphi = 90^\circ$ plane.

Changes:

The information has been added to Supplementary Note 9.

Reviewer #1 – Comments 6:

In the article, the authors use Figure 9 to validate the secure-zone characteristic of the proposed DIM scheme. However, the depicted constellation diagrams are all measured in the coverage area of the generated beams. Distorted constellation diagrams in other directions should be added as a reference to prove the directional security.

Response:

We thank the reviewer for valuable suggestion. We have added the distorted constellation diagrams in other directions for each modulation scheme, as shown in Figure R1.21. Furthermore, we have also presented the EVM distribution for each modulation scheme to assess the quality

of received symbols, as demonstrated in Figure R1.22. The EVM values in the vicinity of the target direction are relatively lower than in other directions, which validates the security property of the proposed method.

Figure R1.21. The distorted constellation diagrams in other directions for demonstrating the directional security. (a) The diagrams for the single-channel 8PSK experiment in the directions $(-45^\circ \sim 29^\circ)$ and $(-5^\circ \sim 20^\circ)$ in the $\varphi = 0^\circ$ plane. (b) The diagrams for the single-channel 64QAM experiment in the directions $(-45^\circ \sim 12^\circ)$ and $(28^\circ \sim 40^\circ)$ in the $\varphi = 0^\circ$ plane. (c) The diagrams for the double-channel 16QAM experiment in the directions $(-45^\circ \sim 18^\circ)$ and $(39^\circ \sim 45^\circ)$ in the $\varphi = 0^\circ$ plane. (d) The diagrams for the double-channel 16QAM experiment in the directions $(-45^\circ \sim 18^\circ)$ and $(39^\circ \sim 45^\circ)$ in the $\varphi = 90^\circ$ plane.

Figure R1.22. The EVM values as the functions of the elevation angles for demonstrating the directional security. (a) The EVM distribution for the single-channel 8PSK experiment in the $\varphi = 0^\circ$ plane. (b) The EVM distribution for the single-channel 64QAM experiment in the $\varphi = 0^\circ$ plane. (c) The EVM distribution for the double-channel 16QAM experiment in the $\varphi = 0^\circ$ plane. (d) The EVM distribution for the double-channel 16QAM experiment in the $\varphi = 90^\circ$ plane.

Changes:

In the revised manuscript, we have replaced Figures 4a and 4b, and Figures 7a and 7b with the corresponding figures of EVM to intuitively demonstrate the directional security.

We have added the distorted diagrams in other directions to Supplementary Note 10 to prove the directional security.

Reviewer #1 – Comments 7:

The authors choose 2-bit quantization level to realize different high-order modulation schemes, such as 16 QAM and 64 QAM. Generally, coding arrays with 2-bit quantization are difficult to realize desired phase and amplitude distributions of high-order modulations since the 2-bit discrete phase distributions has accuracy limitations. However, the measured constellation diagrams of 64 QAM and 16 QAM are nearly the same with the ideal constellation symbols. More detailed analysis of the optimization process with a specific example (eg. 16QAM) is needed.

Response:

We thank the reviewer for the valuable suggestion. We fully understand the reviewer's doubt about utilizing 2-bit phase quantization to realize multi-user and high-order modulation.

Firstly, we should explain that the degree of freedom of the programmable metasurface can support the DIM task. There are 4^{64} coding distributions for an 8×8 metasurface with 2-bit quantization level. We found that although the quantization level is low, the large number of different fields can be generated by using a large-scale metasurface. The field y_k received by the k th user is

$$y_k = \sum_{i=1}^N h_i x_i + n_k, \quad (\text{R9})$$

where x_i , h_i , and n_k are the EM response, channel factor of the i th element, and the noise, respectively. We present a conceptual illustration of the optimization process with a 16QAM symbol, as shown in Figure R1.23. Although each element can only provide four possible discrete phases of x_i , their propagation distances to the user are different. As the scale of array increases, there are many h_i with different phases, and the role of x_i is to combine those h_i to provide the desired symbol at the k th user. The process is shown in Figures R1.23c and R1.23d. We stress that this principle has been analyzed and demonstrated in [R9]. Therefore, our discrete optimization algorithm can find a good feasible coding distribution from the massive candidates.

The results show that although the state of each element is small, the metasurface can still achieve advanced functions using the large-scale arrays and efficient discrete optimization algorithms.

Figure R1.23. The conceptual diagrams for demonstration of the optimization process with an 16QAM symbol. (a) The contributions of each element. (b) The possible contributions of each element under 2-bit quantization level. (c) The overall response without phase optimization. (d) The overall response with phase optimization.

[R9] Y. Shuang, H. Zhao, M. Wei, Q. Cheng, S. Jin, T. Cui, P. Del Hougne, L. Li, *Science China Information Sciences*. **2022**, 65, 172301.

Changes:

We have cited the previous work [R9] as Reference [32], and clarified the optimization process in the revised manuscript.

- Lines 174-178, page 7: We clarify that although each meta-element can only provide finite possible discrete phases of x_i , their propagation distances to the user are different. As the scale of array increases, there will be many h_i with different phases, and the role of x_i is

to combine those h_i to provide the desired symbol at the k th user. We stress that this principle has been analyzed and demonstrated in Ref. [32].

[32] Y. Shuang, H. Zhao, M. Wei, Q. Cheng, S. Jin, T. Cui, P. Del Hougne, L. Li, *Science China Information Sciences*. **2022**, 65, 172301.

Reviewer #1 – Comments 8:

From the explosion view of the element in figure 2(a), the DC bias line is attached with the ground layer and the PIN diodes in feed network are sandwiched between ground layer and dielectric layer, which are unreasonable in the practical situation. The authors should check it carefully.

Response:

We thank the reviewer for comment and apologize for the mistake. We here repaint the explosion view of the element show in Figure R1.24. The DC bias layer is laminated using two pieces of prepreg substrates marked by the pink color. Furthermore, the DC bias line and feed network are located on the bottom of the corresponding substrates.

Figure R1.24. Explosion view of the element.

Changes:

We modified the explosion view and description about the structure around lines xx, page xx in the revised manuscript.

- Lines 140-142, page 5: Each element comprises three parts, including the radiation layer, the direct current bias layer, and the feed network layer, as depicted in Figure 2a. The three parts are laminated together with two pieces of prepreg substrates.

Reviewer #1 – Comments 9:

In Figure 1(c), the phase responses of the element keep the same in the frequency band from 10 GHz to 12 GHz, which seems to violate the dispersion effect. The authors should explain it in detail.

Response:

We thank the reviewer for pointing out the mistake about the phase responses of the element. We wanted to present the phase difference curves in which State 1 (S1) is set as the reference. However, we wrote the Y-axis label wrong and are very sorry for the confusion caused to the reviewer. We have changed the Y-axis label of the phase responses figure by the phase difference, as shown in Figure R1.25b. The phase difference of the four states remains almost 90° over the whole frequency band, which presents the broadband performance of the element.

Figure R1.25. Simulated response of the element. (a) The magnitude. (b) The phase difference.

Changes:

We modified the figure of the phase responses of the element and updated the description.

- Lines 151~154, page 6: The magnitude responses of the four states are close to one in the wide frequency range of 10.7 ~ 11.7 GHz, and the phase difference of the four states remains almost 90° over the whole frequency band, exhibiting the broadband performance of the meta-element.

Reviewer #1 – Comments 10:

According to the authors, the direction (28.5°, 90°) is also the transmission channel. The far-field patterns should form three clusters at $\theta = 28.5^\circ$ as Figure 7(a). However, the far-field patterns in Figure 7(b) associated with 16QAM symbols in the $\varphi = 90^\circ$ plane show the same power at $\theta = 28.5^\circ$. The author should explain this carefully.

Response:

We thank the reviewer for the valuable question. For the results mentioned by the reviewer, we should clarify that the metasurface sends sixteen different symbols to user1 located at the direction (28.5°, 0°) and the same symbols to user 2 at (28.5°, 90°). The measured radiation patterns are consistent with the expectation. We wanted to utilize the above experiment to calculate the cross-talk values between dual users and demonstrate the independent information transmission of the two channels.

In order to obtain the complete crosstalk matrix, we have conducted two independent experiments. Specifically, In the first group of experiments, sixteen different symbols are sent to user 1, and same symbols coded as “0000” are sent to user 2. In the second group of experiments, the sixteen different symbols are sent to user 2, and same symbols coded as “0011” are sent to user 2. The results are shown in Figures R1.26 and R1.27, which matches well with the transmitted symbols.

The measured constellation diagrams of each user are presented in Figures R1.28a and R1.28b.

Taking Figure R1.28a as an illustrative example, we observe that the received symbols of user1 are in good agreement with the 16QAM, while the signals of user2 are clustered near the corresponding reference point. We then calculate the cross-talk matrix C according to Equation (R8), and the results are shown in Figure R1.28c. The cross-talk values between difference users are relatively low, and the maximum is 0.12 (-18.4 dB), implying the acceptable performance under the dual-channel transmission.

Figure R1.26. Results of the first group of tests for the dual-channel 16QAM. (a) The magnitudes and (b) phases of the signals in the desired directions. (c) The magnitudes and (d) phases of fields in the $\varphi = 0^\circ$ plane. (e) The magnitudes and (f) phases of fields in the $\varphi = 90^\circ$ plane.

Figure R1.27. Results of the second group of tests for the dual-channel 16QAM. (a) The magnitudes and (b) phases of the signals in the desired directions. (c) The magnitudes and (d) phases of fields in the $\varphi = 0^\circ$ plane. (e) The magnitudes and (f) phases of fields in the $\varphi = 90^\circ$ plane.

Figure R1.28. Results for assessing the crosstalk in the dual-channel 16QAM. (a-b) The constellation diagrams of each user, where the received and reference symbols are represented by the blue square and red circular markers, respectively. (a) The diagrams in the first group of experiments where sixteen different symbols are sent to user 1 and the same symbols coded as “0000” to user 2. (b) The diagrams in the second group of experiments where sixteen different symbols are sent to user 2 and the same symbols coded as “0011” to user 1. (c) The cross-talk values of dual users.

Changes:

To illustrate the dual-channel directional secure information transmission clearly, we have presented the results of two users receiving 16QAM symbols in the revised manuscript.

- Lines 279~292, pages 12: Figures 5a and 5b show the phases and magnitudes of the measured signals, which match well with the reference symbols. Figures 5c and 5d present the radiation powers and phases of the measured fields in the $\varphi = 0^\circ$ plane associated with the 16QAM, respectively. The radiation powers form three clusters near the desired directions $\theta = 28.5^\circ$, and the average powers of each cluster are presented on the right. From the above results, we clarify that the discrete optimization algorithm can effectively engineer the main-lobe beams to align with the desired directions. The main-lobe powers have exhibited strong correlations with the magnitudes of the transmitted constellation symbols, addressing the effectiveness of energy allocation. The radiation phases of the measured fields are regularly arranged in the vicinity of the target direction. Figures 5e and 5f present the results in the $\varphi = 90^\circ$ plane where we can observe the similar features in the target direction. The above results demonstrate that the simultaneous dual-channel 16QAM is realized. Finally, the optimized digital coding sequences of the 8×8 PM that implements the dual-channel 16QAM are shown in Figure 5g.

The measurements about the crosstalk between dual users have also been added to revised manuscript.

- Lines 293~307, page 12: For a multi-user communication system, the evaluation of crosstalk of different users is crucial because it determines whether the channels are independent. In order to obtain the complete cross-talk matrix (see **Methods** for the definition and calculation), we have conducted two experiments. Specifically, In the first group of experiments, sixteen different symbols are sent to user 1, and same symbols coded as “0000” are sent to user 2. In the second group of experiments, the sixteen different symbols are sent to user 2, and same symbols coded as “0011” are sent to user 1. The magnitudes and phases of the measured fields is demonstrated in Supplementary Note 9, which is well consistent with the transmitted symbols. The measured constellation diagrams of each user are presented in Figures 6a and 6b. Taking Figure 6a as an illustrative example, the symbols received by user1 are in good agreement with the 16QAM, while the signals of user2 are clustered near the corresponding reference point. We then calculate the cross-talk matrix \mathbf{C}

according to Equation (8), and the results are shown in Figure 6c. The cross-talk values between different users are relatively low, and the maximum is 0.12 (-18.4 dB), implying the acceptable performance under the dual-channel transmission.

We have added the measured fields of the two group of experiments to Supplementary Note 9.

Reviewer #1 – Comments 11:

What's the transmission rate of the DIM scheme, and what is the upper modulation frequency limit by using the PIN diodes?

Response:

We thank the reviewer for the questions. The transmission rate limitation of the DIM scheme is mainly determined by the ON-OFF switch speed of meta-atoms and the modulation order. The switch speed of meta-atoms depends on the performances of adopted PIN diodes and the connector between the FPGA board and programmable metasurface. On the one hand, the adopted PIN diode (MACOMMADP-000907-14020x) can support the switching speed up to 2-3 ns. However, the switch speed of meta-atoms cannot reach the limit due to a relatively low-speed connector. Figure R1.29 demonstrates the measured control waveforms with different switching frequency using an oscilloscope. We can observe that the signal waveform with 50 MHz is relatively distorted due to the influence of the parasitic inductance of the connector. Therefore, the modulation frequency shift of the current DIM scheme can reach around 10 MHz.

Furthermore, we should emphasize that the current DIM scheme is a proof-of-concept DIM scheme demonstrating its flexibility for two-dimensional and multi-user directionally secure transmission using high-order modulation. The current transmission rate is 2 Mbps using 16QAM modulation. The limitation of the modulation speed of the DIM scheme can be largely mitigated by, e.g., (1) exploiting a high-speed connector (such as FMC, PCI Express); (2) packaging the FPGA and programmable metasurface on one PCB board; (3) optimizing the circuit layout after professional signal integrity analysis.

Figure R1.29. Measured control signal waveforms with different frequencies. (a) 10 MHz. (b) 50 MHz.

Changes:

We have added the description and measurements about the upper modulation frequency limit to Supplementary Note 12.

Reviewer #1 – Comments 12:

There are some spelling mistakes in the article, such as “Figure 6d to 6d” at line 312, page 12 and “in the vicinity of” at line 367, page 16. The authors should carefully check the article for grammatical errors.

Response:

We are sorry for the mistakes and thank the reviewer for pointing out the grammatical errors.

Changes:

We have corrected these mistakes and carefully examined the revised manuscript.

- Equation (4):

$$\begin{array}{ccc}
 \min_x & \|s - \beta Hx\|_2^2 + K\beta^2\sigma^2 & \longrightarrow & \min_{x, \tilde{x}, \beta} & \|s - \beta Hx\|_2^2 + K\beta^2\sigma^2 \\
 \text{s. t.} & \tilde{x} = x, & & \text{s. t.} & \tilde{x} = x, \\
 & \tilde{x}(i) \in \mathcal{X}, i = 1, \dots, N & & & \tilde{x}(i) \in \mathcal{X}, i = 1, \dots, N
 \end{array}$$

General comments from Reviewer #2

Thanks for submitting your manuscript to Nature Communications. I enjoyed reading your paper as the paper is easy to follow and well-written. However, I have several concerns regarding the novelty and significance of the proposed technique.

Response:

We thank the reviewer for the comments about our manuscript, which encouraged us a lot. Following the reviewer's comments, we have provided a compare-and-contrast study with the existing directional information modulation systems and summarized the novelty and advantages of our work. Furthermore, we have carefully responded to the comments point-to-point and revised the manuscript. We hope that the following responses are sufficient for highlighting the value of the work and dispelling your doubts.

Specific comments from Reviewer #2

Reviewer #2 – Comments 1:

First, the idea of directional information modulation is not new. As acknowledged by the authors, DIM has been investigated heavily in the past in theory and in practice. There are many proposed architectures that support the transmission of DIM. The use of metasurfaces for such transmission is not new either. Indeed, this has been published in the past by the authors themselves. Hence, both the concept and implementation of the idea are known in the literature.

Response:

We thank the reviewer for the comment. First, we fully agree that the idea of directional information modulation is not new. There are several typical schemes achieving the DIM transmission, including the phased arrays, time-modulated arrays, time-domain coding metasurfaces and time-space-domain coding metasurfaces. Second, to clarify the novelty and improvement of our work, we here demonstrate a compare-and-contrast study with the above schemes.

The comparison of our work in contrast to other existing categories achieving physical-layer

security (PLS) are listed in terms of directional security, system complexity, performance, realization strategy, advantages, and disadvantages, as summarized in Table R2.1. Specifically, we do not give the corresponding performance comparison since the beamforming-based techniques and metasurface-coated devices are unable to achieve directional security. Compared with other existing schemes, our proposal has the following advantages:

- Our proposal can achieve two-dimensional and multi-channel (up to four channel and even more with larger-scale metasurfaces) DIM.
- Our proposal can achieve different high order modulations, such 8PSK, 16QAM and 64QAM schemes.
- Our proposal is secure against harmonic compared with time-coding digital metasurfaces.
- Our proposal can serve as a low-profile transmitter or receiver, which greatly reduce system complexity.

1. Beamforming-based techniques

Beamforming (BF) based techniques have been widely employed for the PLS of wireless communications. The techniques are designed either to enhance the power at the receiver or to degrade the wiretap channel used by the eavesdropper. Additionally, BF often adopts other PLS techniques involving the hybridization of four methods ^[R1-R3] (1) secure BF, (2) artificial noise BF, (3) cooperative jamming-aided BF, and (4) joint channel coding with BF.

In summary, these methods increase the SNR difference between the receiver and eavesdropper and they are motivated by Wyner's wiretap model. However, the information is still encoded in the time domain. The broadcast nature of wireless communication channels allows the signals in all directions to have an identical structure, only with different power levels. Therefore, an eavesdropper equipped with a sensitive receiver can intercept the information in theory. As a result, the demand for information security necessitates the idea of directional communications.

2. Metasurface-coated devices

The passive metasurface-coated devices can concentrate the power of ambient signals on the

transmitter or receiver, which hopefully realizes energy efficiency, energy harvesting, and ultralow-power transmissions. ^[R4, R5] Furthermore, the PLS is ensured by increasing the power disparity between the receiver with the metasurface-coated device and scattered eavesdroppers. The device mitigates the critical problem of the tradeoff between ultralow-power transmissions and secrecy capacity for the first time, which is a great improvement compared with the traditional beamforming techniques. However, although the power received by eavesdroppers is very low, the signal structure is still undistorted. Therefore, the security risk of the above method exists when an eavesdropper equipped with a sensitive receiver can intercept the information in theory. Furthermore, the metasurface-coated device is passive, thus it is unable to realize real-time spatial wave manipulations and directional information modulation.

3. Phased arrays

The phased arrays usually require expensive transmitter and receiver (T/R) components for each radio-frequency channel, which results in a bulky and energy-hungry system. In the first realization of DIM based on phased arrays, the single-channel mode was realized using the QPSK modulation. ^[R6] The phase shifts of each element are optimized based on a genetic algorithm to minimize the bit error rate (BER) of the desired direction while maximizing the BER elsewhere. However, the system complexity is high, and it is hard to extend larger scales due to the expensive T/R components.

4. Time-domain coding metasurfaces

The time-domain coding metasurface (TDCM) is constructed by integrating the electronically controllable components, such as the varactor or positive-intrinsic-negative (PIN) diodes, with the low-cost metal patches. The realization of high-order modulations is to optimize the periodic sequences to change the magnitude and phase of the harmonic wave, respectively. Therefore, the TDCM-based DIM scheme is low-cost and power-efficient, and the complexity mainly depends on the optimization of the periodic sequences. However, the TDCM employs the harmonic of the electromagnetic (EM) waves, which will result in spectra pollution and is unable to ensure

directional secure transmission due to the lack of space coding.

We here present two representative works about the TDCM-based DIM scheme. A single-channel 16QAM mode is implemented based on a varactor-loaded metasurface with 8×16 elements.^[R7] The phases and magnitudes of the harmonic signals are changed by the introduction of delay time and the relative phase difference of the pulse wave. The second example realizes the single-channel 256QAM transmission based on a 1-bit metasurface with 56×20 elements.^[R8] The phases and magnitudes of the harmonic signals are changed by the introduction of delay time and duty ratio of the pulse wave. From the above examples, we find that the harmonic signal of arbitrary modulation can be realized by elaborately changing the form of the pulse wave in theory. However, the corresponding changes also occur at other-order harmonic at the same time, which means there exists the risk of information leakage.

5. Time-space-domain coding metasurfaces

Time-space-domain coding metasurfaces (TSDCM) have a similar structure to TDCM, but the coding strategy is different. First, the TSDCM-based DIM scheme does not utilize periodic sequences, which means that the system does not generate the harmonic. We also prove that the received signals in the harmonic are noise. Second, the TSDCM-based DIM scheme supports the manipulation of multiple spatial beams and temporal information with directional security.

We here present two representative examples that utilize the TSDCM-based DIM scheme. The hardware is a 1-bit programmable metasurface with different scales and the method is a modified Gerchberg-Saxton (GS) algorithm.^[R9, R10] They can realize the two-dimensional and multi-channel transmission. However, these implementations are facing several limitations. First, they only transmit signals. This is incompatible with the operation of traditional wireless systems, where the base station or terminal device can both transmit and receive information. Second, the metasurface-based transmitter typically needs external RF sources and has a large profile, which is inconvenient for space-constrained applications such as wearable Internet of Things (IoT) devices, vehicles, and aircraft. Moreover, the modulation are only BPSK or QPSK, which suffer

from a lack of high-order modulation and QAM schemes that carry more information capacities.

6. The proposed DIM scheme

From the perspective of the coding strategy, our proposed scheme can also be classified as TSDCM. However, one significant difference is that our system is reciprocal, which means that it can serve as a low-profile transmitter and receiver simultaneously. The scheme supports the identical metasurface device to complete the signal radiating and receiving, thus making the system simpler and easier to maintain.

We note that the main limitation of the TSDCM scheme is the finite phase-shift states of each meta-atom. Therefore, an efficient optimization algorithm for spatial coding sequences is needed. We have developed the ADMM-based discrete optimization algorithm and experimentally validated the four-channel high-order modulation. We have also compared it with other methods reported in the field of DIM, such as the GA, modified GS, and squared-infinity norm Douglas-Rachford splitting. The simulations demonstrate that our method has faster convergence, better performance, and scalability. Although we conducted the experimental validation on a small-scale metasurface (8×8), our design can be easily extended to a larger scale, and the proposed algorithm is more competitive in this scenario.

Table R2.1. Comparison with reported schemes achieving physical-layer security

	Directional security	System Complexity	Performance	Realization strategy	Advantages	Disadvantages
Beamforming-based techniques	No	--	--	--	--	--
Metasurface-coated devices	No	--	--	--	--	--
Phased arrays	Yes	High: the utilization of expensive T/R components	Single-channel QPSK modulation ^[R6]	A four-element linear array; GA algorithm	The continuous phase shifts	Poor scalability due to expensive cost
Time-domain coding	No: Information	Low: optimization of	Single-channel 16QAM ^[R7]	A varactor-loaded metasurface with	The continuous phase shifts and	Unsecure on harmonic;

metasurfaces	leakage on harmonics	periodic sequences		8×16 elements; Optimization of periodic waves	magnitudes;	Single-channel mode;
			Single-channel 256QAM ^[R8]	A 1-bit metasurface with 56×20 elements; Optimization of periodic waves		
Time-space-domain coding metasurfaces	Yes	Low: optimization of coding sequences	Dual-channel QPSK modulation ^[R9]	A 1-bit metasurface with 20×20 elements; Modified GS algorithm	Secure against harmonic; Two-dimensional and multiple-channel mode.	Finite phase shifts; Low-order modulation;
			Three-channel BPSK modulation ^[R10]	A 1-bit metasurface with 24×32 elements; Modified GS algorithm		
Our work	Yes	Low: optimization of coding sequences	Four-channel 8PSK, 16QAM and 64QAM	A 2-bit metasurface with 8×8 elements; ADMM-based algorithm	Secure against harmonic; Two-dimensional and multiple-channel mode; Serve as transmitter or receiver with low profile; High-order modulation;	Finite phase shifts; Small-scale array

Changes:

We have modified the introduction to clarify the value and novelty of our work.

We have added the above reply to Supplementary Note 1.

[R1] Y. Liu, J. Li, A. P. Petropulu, *IEEE Transactions on Information Forensics and Security*. **2013**, 8, 682.

- [R2] Q. Wang, F. Zhou, R. Q. Hu, Y. Qian, *IEEE Transactions on Wireless Communications*. **2021**, 20, 2592.
- [R3] R. M. Yamada, A. O. Steinhardt, L. Mili, *IEEE Transactions on Wireless Communications*. **2017**, 16, 8026.
- [R4] T. A. Tsiftsis, C. Valagiannopoulos, H. Liu, A. A. A. Boulogeorgos, N. I. Miridakis, *IEEE Vehicular Technology Magazine*. **2022**, 17, 27.
- [R5] D. Tulegenov, C. Valagiannopoulos, *Journal of Applied Physics*. **2022**, 131.
- [R6] M. P. Daly, J. T. Bernhard, *IEEE Transactions on Antennas and Propagation*. **2009**, 57, 2633.
- [R7] J. Y. Dai, W. Tang, L. X. Yang, X. Li, M. Z. Chen, J. C. Ke, Q. Cheng, S. Jin, T. J. Cui, *IEEE Transactions on Antennas and Propagation*. **2020**, 68, 1618.
- [R8] M. Z. Chen, W. Tang, J. Y. Dai, J. C. Ke, L. Zhang, C. Zhang, J. Yang, L. Li, Q. Cheng, S. Jin, T. J. Cui, *National Science Review*. **2022**, 9, nwab134.
- [R9] H. Zhao, Y. Shuang, M. Wei, T. J. Cui, P. D. Hougne, L. Li, *Nature Communications*. **2020**, 11, 3926.
- [R10] X. Wan, C. Xiao, H. Huang, Q. Xiao, W. Xu, Y. Li, J. Eisenbeis, J. Wang, Z. Huang, Q. Cheng, S. Jin, T. Zwick, T. Cui, *Engineering*. **2022**, 8, 86.

Reviewer #2 – Comments 2:

The delta in this work seems to be the optimization framework and algorithm to find the correct phase profile on the metasurface for multi-user support. However, the reviewer does not see that as a major and significant contribution for this prestigious journal.

Response:

We thank the reviewer for the valuable comment. We admit that one of our contributions is the proposed ADMM-based discrete optimization algorithm. However, the focus of the present work is to demonstrate the application of two-dimensional and multi-user DIM. It is worth noting that our hardware can serve as a transmitter or receiver with a low profile and our scheme is secure against harmonics.

Furthermore, our system supports high-order modulation schemes, such as 8PSK, 16QAM, and 64QAM. The optimization algorithm is an efficient tool that we leverage to obtain the feasible coding sequences to implement DIM. This is also the trend that intelligent and efficient algorithms are required to achieve complex and advanced functions based on the metasurface platform apart from the hardware realization with excellent performance.

Reviewer #2 – Comments 3:

The optimization framework and the algorithm are both straightforward. Further, the implementation is done for up to two simultaneous users. It is not clear how the performance scales with more users.

Response:

We also believe that the experiment validation to support a greater number of channel direction transmission is crucial to improve the quality and advance of our work. We are greatly encouraged by the reviewer's suggestion.

As an illustrative example, we conducted new experiments with up to four simultaneous users, whose locations are $(\theta, \varphi) = (12^\circ, 0^\circ)$, $(\theta, \varphi) = (-30^\circ, 0^\circ)$, $(\theta, \varphi) = (42^\circ, 90^\circ)$, and $(\theta, \varphi) = (-34^\circ, 90^\circ)$, respectively. The performance of DIM for more users is verified using the 8PSK, 16QAM, and 64QAM schemes, respectively. We here briefly summarize the data and figures for each scheme. For the 8PSK scheme, we present the magnitudes and phases of measured fields, the received constellation diagrams of each user, the optimized coding sequences of the 8×8 programmable metasurface, and the EVM distribution over the elevation angles for validating the directional security. For the 16QAM scheme, apart from the above data, we also perform the cross-talk measurements and draw the corresponding matrix values. For the 64QAM scheme, we only present the received constellation diagrams of each user and the EVM distribution for the consideration of simplicity and clarity.

1. The results of four-channel 8PSK scheme

Figures R2.1a and R2.1b show the magnitudes and phases of the measured fields for the 8PSK scheme along the $\varphi = 0^\circ$ plane. In this case, the main lobe beams are located near the target directions $\theta = 12^\circ$ and $\theta = -30^\circ$, respectively, and the radiation powers are almost equal for all symbols. Another interesting observation is that the phases of measured fields have an interval of nearly 45° in the vicinity of the target directions. Figures R2.1c and R2.1d show the results along the $\varphi = 90^\circ$ plane, where we observe the similar features in the target directions $\theta = 42^\circ$ and $\theta = -34^\circ$. The above results demonstrate that the simultaneous four-channel 8PSK scheme is realized with the help of our DIM system. More intuitionistic evidence is shown in Figure R2.2, in which the measured symbols are in excellent agreement with the references. The optimized digital coding distributions of the 8×8 PM that implements the four-channel 8PSK modulation are demonstrated in Figure R2.3.

As shown in Figure R2.4, we utilize the error vector magnitude (EVM) to quantify the performance of directional secure transmission. The metric of EVM is more intuitive and commonly used in wireless communications to assess the quality of constellation diagrams. ^[R11] The EVM is defined at a given time as the vector difference between the measured and reference symbols: $EVM = \sqrt{\sum_{i=1}^M |s_{ref}^i - s_{mea}^i|^2 / \sum_{i=1}^M |s_{ref}^i|^2}$, where s_{ref} , s_{mea} , and M are the reference, measured signals, and the number of symbols, respectively. The lower the EVM value is, the closer the measured symbols are to the reference. The EVM distribution as the functions of the elevation angles for the 8PSK scheme is demonstrated in Figure R2.4. The EVM values in the vicinity of the target directions are relatively lower, which validates the security property of the proposed method. We should explain the observation that the EVM values are low near the direction $\theta = 0^\circ$ along the $\varphi = 90^\circ$ plane shown in Figure R2.4b. The reason behind this is that the measured fields in this part are also within the secure zone of user 1 (i.e., in the vicinity of the direction $(\theta, \varphi) = (12^\circ, 0^\circ)$). We also present the measured constellation in the unintended directions shown in the bottom subfigures, in which the signal structures are distorted.

2. The results of four-channel 16QAM scheme

Figures R2.5a and R2.b present the magnitudes and phases of the measured fields along the $\varphi = 0^\circ$ plane associated with the 16QAM, respectively. As illustrated in Figure R2.5a, the radiation patterns form three clusters near the desired directions $\theta = 12^\circ$ and $\theta = -30^\circ$, and the average powers of each cluster are presented. The radiation phases of measured fields are regularly arranged in the vicinity of the target directions. Figures R2.5c and R2.5d show the results along the $\varphi = 90^\circ$ plane, where we observe the similar features in the target directions $\theta = 42^\circ$ and $\theta = -34^\circ$. The above results demonstrate that the simultaneous four-channel 16QAM is realized. More intuitionistic evidence is shown in Figure R2.6, in which the measured symbols are in good agreement with the reference.

Figure R2.1. The measured fields in the four-channel 8PSK experiment. (a)-(b) The results in the $\varphi = 0^\circ$ plane. (c)-(d) The results in the $\varphi = 90^\circ$ plane.

Figure R2.2. The measured constellation diagrams of the four-channel 8PSK experiment. The measured and the reference symbols are represented by the blue square and the red circular markers, respectively. (a) The diagrams of user 1 located at $(\theta, \varphi) = (12^\circ, 0^\circ)$. (b) The diagrams of user 2 located at $(\theta, \varphi) = (-30^\circ, 0^\circ)$. (c) The diagrams of user 3 located at $(\theta, \varphi) = (42^\circ, 90^\circ)$. (d) The diagrams of user 4 located at $(\theta, \varphi) = (-34^\circ, 90^\circ)$.

Figure R2.3. The optimized coding distributions of the 8×8 programmable metasurface that implement four-channel 8PSK. (S1: -90° , S2: 0° , S3: 90° , S4: 180°)

Figure R2.4. The measured EVM values in the four-channel 8PSK experiment. **(a)** The EVM distributions in the $\varphi = 0^\circ$ plane. The bottom subfigures present two constellation diagrams that deviate from the desired users. **(b)** The EVM distributions along the $\varphi = 90^\circ$ plane. The bottom subfigures present two constellation diagrams that deviate from the desired users.

The EVM distribution as the functions of the elevation angles is demonstrated in Figure R2.7. The EVM values in the vicinity of the target directions are relatively lower than in other directions, which validates the security property of the proposed method.

For a multi-user communication system, the evaluation of cross-talk value is crucial because it determines whether the channels are independent. However, to the best of authors' knowledge, the definition and calculation method of multi-user crosstalk in the field of DIM are still absent. For filling up this gap, we try to quantify the crosstalk for multiple users in terms of a matrix \mathbf{C} in which the matrix is composed of $K \times K$ elements c_{ij} (K is the number of users). We should first explain the motivations for calculating cross-talk values. In the field of electrical or optical communications, the crosstalk is defined as the cross-power coupling between multiple channels [R12]. When the input signal of one channel changes, receivers measure the variation of signals at the other channels. According to the difference values, the magnitudes of crosstalk are obtained. We here mimic the operation and perform the corresponding the experiments. For calculating

the element c_{ij} ($i \neq j, j = 1, 2, \dots, i-1, i+1, \dots, K$), we perform the experiment that the transmitter simultaneously sends M different symbols to user i and the M same symbols to the other users. The c_{ij} ($i \neq j$) is

$$c_{ij} = \sqrt{\sum_{m=1}^M |s_{mea}^j(m) - s_{ref}^j(m)|^2 / \sum_{m=1}^M |s_{ref}^j(m)|^2}, \quad (R1)$$

where s_{mea}^j and s_{ref}^j are the measured and reference symbols of user j . Finally, the value of

$$c_{ii} \text{ is } \sqrt{1 - \sum_{j \neq i} c_{ij}^2}.$$

To measure the crosstalk values of four users using the proposed method, we conducted four independent experiments. In the first group of experiments, the sixteen different symbols are sent to user 1, and same symbols coded as “0011” are sent to user 2, and same symbols coded as “1001” are sent to user 3, and same symbols coded as “1100” are sent to user 4. In the second group of experiments, the sixteen different symbols are sent to user 2, and same symbols coded as “1011” are sent to user 1, and same symbols coded as “0110” are sent to user 3, and same symbols coded as “1110” are sent to user 4. In the third group of experiments, the sixteen different symbols are sent to user 3, and same symbols coded as “1100” are sent to user 1, and same symbols coded as “0011” are sent to user 2, and same symbols coded as “1110” are sent to user 4. In the fourth group of experiments, the sixteen different symbols are sent to user 4, and same symbols coded as “1011” are sent to user 1, and same symbols coded as “0001” are sent to user 2, and same symbols coded as “0011” are sent to user 3.

The measurements are presented in Figures R2.8a – R2.8d. Taking Figure R2.8a as an illustrative example, we observe that the symbols received by user 1 are in good agreement with the 16QAM, while the signals of user 2, 3, and 4 are clustered near the corresponding reference point. The results demonstrate that our system has good crosstalk resistance. We then calculate the whole cross-talk matrix \mathbf{C} according to the above definition of c_{ij} . As show in Figure R2.9, the cross-talk values between difference users are relatively low and the maximum is 0.24 (-12.4 dB), implying the acceptable performance under the four-channel transmission.

Figure R2.5. The measured fields in the four-channel 16QAM scheme. (a)-(b) The results in the $\varphi = 0^\circ$ plane. (c)-(d) The results in the $\varphi = 90^\circ$ plane.

Figure R2.6. The measured constellation diagrams of the four-channel 16QAM experiment. The measured and the reference symbols are represented by the blue square and the red circular markers, respectively. (a) The diagrams of user 1 located at $(\theta, \varphi) = (12^\circ, 0^\circ)$. (b) The diagrams of user 2 located at $(\theta, \varphi) = (-30^\circ, 0^\circ)$. (c) The diagrams of user 3 located at $(\theta, \varphi) = (42^\circ, 90^\circ)$. (d) The diagrams of user 4 located at $(\theta, \varphi) = (-34^\circ, 90^\circ)$.

Figure R2.7. The measured EVM values in the four-channel 16QAM experiment. (a) The EVM distribution in the $\varphi = 0^\circ$ plane. The bottom subfigures present two constellation diagrams that deviate from the desired users. (b) The EVM distribution in the $\varphi = 90^\circ$ plane. The bottom subfigures present two constellation diagrams that deviate from the desired users.

Figure R2.8. The measured constellation diagrams of each user in the experiments that measure the cross-talk

values of multiple users. The measured and the reference symbols are represented by the blue square and the red circular markers, respectively. (a) The diagrams of the first group of experiments. (b) The diagrams of the second group of experiments. (c) The diagrams of the third group of experiments. (d) The diagrams of the fourth group of experiments.

$$\mathbf{C} = \begin{bmatrix} c_{11} & c_{12} & c_{13} & c_{14} \\ c_{21} & c_{22} & c_{23} & c_{24} \\ c_{31} & c_{32} & c_{33} & c_{34} \\ c_{41} & c_{42} & c_{43} & c_{44} \end{bmatrix} = \begin{bmatrix} 0.95 & 0.13 & 0.23 & 0.13 \\ 0.15 & 0.97 & 0.13 & 0.13 \\ 0.23 & 0.18 & 0.95 & 0.13 \\ 0.24 & 0.17 & 0.19 & 0.94 \end{bmatrix}$$

Figure R2.9. The cross-talk values of four users.

3. The results of four-channel 64QAM scheme

We present the measurements of the four-channel 64QAM scheme. Such configuration is more difficult than the previous experiments since the modulation has higher order, and the distances between adjacent symbols are closer. Figure R2.10 shows the measured symbols of each user, in which the results of user 3 and 4 are relatively better than user 1 and 2. The EVM distribution as the functions of the elevation angles is given in Figure R2.11. The EVM values in the vicinity of the target directions are relatively lower than in other directions, which validates the security property of the proposed method.

Figure R2.10. The received constellation diagrams in the four-channel 64QAM experiment. The measured and the reference symbols are represented by the blue square and the red circular markers, respectively. (a) The diagrams of user 1 located at $(\theta, \varphi) = (12^\circ, 0^\circ)$. (b) The diagrams of user 2 located at $(\theta, \varphi) = (-30^\circ, 0^\circ)$. (c) The diagrams of user 3 located at $(\theta, \varphi) = (42^\circ, 90^\circ)$. (d) The diagrams of user 4 located at $(\theta, \varphi) = (-34^\circ, 90^\circ)$.

Figure R2.11. The EVM values as the functions of the elevation angles in the four-channel 64QAM scheme. (a) The EVM distribution in the $\varphi = 0^\circ$ plane. The bottom subfigures present two constellation diagrams that deviate from the desired users. (b) The EVM distribution in the $\varphi = 90^\circ$ plane. The bottom subfigures present two constellation diagrams that deviate from the desired users.

Changes:

We have added the following sentences to clarify that our scheme can support four-channel transmissions.

- Lines 241~244, page 9: **However, the capability of our scheme is not limited to dual channels, and the measurements for the four-channel DIM are demonstrated in Supplementary Note 8 (even more channels can be achieved if the scale of metasurface increases).**

Furthermore, we have added the measurements for validation of the reciprocity of our DIM scheme to Supplementary Note 8.

[R11] S. Forestier, P. Bouysse, R. Quere, A. Mallet, J. M. Nebus, L. Lapierre, *IEEE Transactions on Microwave Theory and Techniques*. 2004, 52, 1132.

[R12] S. SeyedinNavadeh, M. Milanizadeh, F. Zanetto, G. Ferrari, M. Sampietro, M. Sorel, D.

A. B. Miller, A. Melloni, F. Morichetti, *Nature Photonics*. **2023**.

Reviewer #2 – Comments 4:

Further, what are other baseline algorithms that can do the job? what are the relevant performance metrics and how does the proposed algorithm compare against the baseline (currently undefined)? The answers to these questions are currently missing.

Further, the authors should show the performance with many more users and with a larger metasurface to justify the scalability of their algorithm. Can each user see a different constellation? how much is the cross-talk?

Response:

We thank the reviewer for the valuable comments and suggestions, and apologize for the lack of the compare-and-contrast study with other algorithms, the definition of the performance metrics, and the discussion about the scalability. They are crucial for the large-scale applications in communication systems.

To the first question: as we mentioned in the introduction section of the main text, we notice that GA has been utilized to realize DIM in previous work, however, the optimization variables are continuous rather discrete. For the metasurface-based DIM system, the modified Gerchberg-Saxton (MGS) algorithm has been exploited to optimize the discrete coding sequences, ^[R13, R14] however, they only realize the PSK modulation scheme. We also find that the squared-infinity norm Douglas-Rachford splitting (SQUID), one of the baseline algorithms in the field of low-resolution Massive MIMO, can be used to realize the job. ^[R15] We modify the GA and MGS algorithms of the reference papers to realize the transmission of QAM constellation symbols by introducing the β factor (defined in the main text). Therefore, we compare the proposed ADMM-based method with the GA, MGS and SQUID algorithms. It is worth noting that in order to make a fair comparison, the parameters in the GA, MGS and SQUID algorithms are configured according to the recommended values of the corresponding reference papers.

To the second question: In the field of DIM, the bit error rate (BER) in the desired directions, EVM values that quantify the quality of received constellation diagrams, and the secrecy rate have been used the performance metrics. ^[R16] However, the secrecy rate is a system performance that depends on the channel type, environment noise and modulation scheme, thus it is not suitable for being the metric of algorithm comparison. The BER and EVM are both used to assess the received quality, ^[R17] thus the EVM is adopted to assess the performance of DIM physical-layer secure transmission systems. Furthermore, the convergence curve of the object value (i.e., $\|\mathbf{s} - \beta\mathbf{H}\mathbf{x}\|_2^2 + K\beta^2\sigma^2$ defined in the main text) with the number of iterations is also employed to evaluate the feasibility and scalability of the algorithm in optimization problems. In summary, we utilize the EVM values and the convergence performance of different algorithms as the metrics.

To the suggestion: we perform two representative simulation cases, as summarized in Table R2.2, to demonstrate the scalability of the algorithms for multiple users and different scale metasurfaces. Specially, we utilize the simulation of Case I to demonstrate the performance of the algorithms with different scale metasurface. We utilize the simulation of Case II to demonstrate the performance of the algorithms with different scale users. In Case I, there are four desired directions, in which the first two are along $\theta = 20^\circ$ and -30° , respectively, on the $\varphi = 0^\circ$ plane; while the last two are along $\theta = 34^\circ$ and -41° , respectively, on the $\varphi = 45^\circ$ plane. The simulations are conducted within three types of metasurface scale, including 4×4 , 8×8 , and 16×16 . In Case II, the metasurface scale is 16×16 . The simulations are conducted within four and six desired directions, respectively. For each of the simulations, the modulation is 16QAM, and the signal-to-noise (SNR) is 15 dB, and the symbols in different desired directions are selected randomly, and 2000 times of realization are performed in the Monte Carlo simulations. Each simulation is implemented using the ADMM, GA, MGS, and SQUID, respectively.

Table R2.2. Simulation configurations for validating the scalability of the algorithms

Scenario	Desired directions	Scale	Common settings
Case I	$(\theta_1 = 20^\circ, \theta_2 = -30^\circ, \varphi_{1,2} = 0^\circ)$	4×4	Modulation: 16QAM; SNR: 15 dB; Simulations: 2000 Algorithms: ADMM, GA, MGS, SQUID;
	$(\theta_3 = 34^\circ, \theta_4 = -41^\circ, \varphi_{3,4} = 45^\circ)$	8×8	
Case II	$(\theta_1 = 20^\circ, \theta_2 = -30^\circ, \varphi_{1,2} = 0^\circ)$	16×16	
	$(\theta_3 = 34^\circ, \theta_4 = -41^\circ, \varphi_{3,4} = 45^\circ)$		
	$(\theta_1 = 18^\circ, \theta_2 = 50^\circ, \theta_3 = -30^\circ, \varphi_{1,2,3} = 0^\circ)$		
	$(\theta_4 = 33^\circ, \theta_5 = -11^\circ, \theta_6 = -51^\circ, \varphi_{4,5,6} = 45^\circ)$		

The results of Case I

Figures R2.12a, R2.12b, and R2.12c demonstrate the variations of the object functions using different algorithms for the metasurface scales 4×4 , 8×8 , and 16×16 , respectively, and there are four desired directions. As shown in Figure R2.12a, the GA algorithm has the best convergence accuracy in the small-scale metasurface optimization problems, while the proposed ADMM algorithm has the worst performance. We should admit that the proposed algorithm is not competitive when dealing with small-scale problems. However, when the metasurface scale increases to 8×8 , the ADMM algorithm has comparable performance the benchmark SQUID, as shown in Figure R2.12b. Furthermore, we notice that the GA and MGS algorithms quickly fall into the local minimum value. Figure R2.12c demonstrates the case of higher scale 16×16 , where our algorithm has the best convergence rate and accuracy, while the SQUID is unable to converge.

Below each corresponding convergence picture, we plot the EVM distribution as the functions of the elevation angle in the $\varphi = 0^\circ$ plane and in the $\varphi = 45^\circ$ plane, respectively. As shown in Figure R2.12d, the lowest EVM values are located right at the desired directions, while the values in other directions are relatively larger, thus validating the directional secure property of the system. Comparing the results shown in Figures R2.12d-R2.12f, we observe that with the

increase of metasurface scale, the EVM values optimized by the ADMM algorithm decreases continuously in the desired directions, indicating that our proposed method can handle the large-scale problems and has better performance.

Figure R2.12. Simulations that validate the performance of the algorithms with different metasurface scales. There are four desired directions, including $(\theta_1, \varphi_1) = (20^\circ, 0^\circ)$, $(\theta_2, \varphi_2) = (-30^\circ, 0^\circ)$, $(\theta_3, \varphi_3) = (34^\circ, 45^\circ)$, and $(\theta_4, \varphi_4) = (-41^\circ, 45^\circ)$ (a-c) Variations of the object functions for three metasurface scales, 4×4 , 8×8 , and 16×16 , respectively. (d), (g) The EVM distributions as the function of the elevation angles in the $\varphi = 0^\circ$ plane and $\varphi = 45^\circ$ plane, respectively, and the metasurface scale is 4×4 . (e), (h) The EVM distributions as the function of the elevation angles in the $\varphi = 0^\circ$ plane and $\varphi = 45^\circ$ plane, respectively, and the metasurface scale is 8×8 . (f), (i) The EVM distributions as the function of the elevation angles in the $\varphi = 0^\circ$ plane and $\varphi = 45^\circ$ plane, respectively, and the metasurface scale is 16×16 .

The results of Case II

Figures R2.13a and R2.13b demonstrate the variations of the object functions using different

algorithms for four and six desired directions, respectively, in which the metasurface scale is 16×16 . The results about the four-channel case have been analyzed in the previous section. Our attention turns to the results of six desired directions. As shown in Figure R2.13b, the proposed ADMM method has the best convergence performance compared with other algorithm, and the GA, MGS, SQUID quickly fall into the local minimum values. The results demonstrate that our method can solve larger-scale problems. Furthermore, the lowest EVM shown in Figures R2.13c and R2.13d are in the desired directions. The EVM values optimized by the ADMM are lowest, which matches well with the convergence results.

Figure R2.13. Simulations that validate the performance of the algorithms with different number of users. The

metasurface scale is 16×16 . (a), (b) Variations of the object functions when the number of users are four and six desired directions, respectively. (c), (d) The EVM distributions as the function of the elevation angles in the $\varphi = 0^\circ$ and $\varphi = 45^\circ$ planes, respectively, and there are four users. (e), (f) The EVM distributions as the function of the elevation angles in the $\varphi = 0^\circ$ plane and $\varphi = 45^\circ$ plane, respectively, and there are six users.

To your third and fourth questions: as an illustrative example, we simulate the case of serving four users with the metasurface scale of 8×8 elements combined with the proposed ADMM method. For the multi-user communication system, the evaluation of cross-talk value is crucial because it determines whether the channels are orthogonal. We employ the definition and calculation of cross-talk value (the response to the **Comment 3**) to assess the multi-users interference. The constellations of each user are also demonstrated in the following results.

To obtain the crosstalk values of four users, we conducted four independent experiments. In the first group of experiments, the different symbols are sent to user 1, and same symbols coded as “0011” are sent to user 2, and same symbols coded as “1001” are sent to user 3, and same symbols coded as “1100” are sent to user 4. In the second group of experiments, the different symbols are sent to user 2, and same symbols coded as “1011” are sent to user 1, and same symbols coded as “0110” are sent to user 3, and same symbols coded as “1110” are sent to user 4. In the third group of experiments, the different symbols are sent to user 3, and same symbols coded as “1100” are sent to user 1, and same symbols coded as “0011” are sent to user 2, and same symbols coded as “1110” are sent to user 4. In the fourth group of experiments, the different symbols are sent to user 4, and same symbols coded as “1011” are sent to user 1, and same symbols coded as “0001” are sent to user 2, and same symbols coded as “0011” are sent to user 3.

The received constellation diagrams are presented in Figures R2.14a-R2.14d. Take Figure R2.14a as an example, we observe that the received symbols of user 1 are in good agreement with the 16QAM, while the signals of user 2, 3, and 4 are clustered near the corresponding reference point. The results indicate that our system has good crosstalk resistance and the users can observe different constellation diagrams. We then calculate the cross-talk matrix \mathbf{C} according

to Equation (R1). As show in Figure R2.15, the cross-talk values between difference users are relatively low, and the maximum is 0.11 (-19.2 dB).

Figure R2.14. The simulated constellation diagrams of each user that measure the cross-talk values of four users. The simulated and the reference symbols are represented by the blue and the red circular markers, respectively. (a) The diagrams of the first group of experiments. (b) The diagrams of the second group of experiments. (c) The diagrams of the third group of experiments. (d) The diagrams of the fourth group of experiments.

$$\mathbf{C} = \begin{bmatrix} c_{11} & c_{12} & c_{13} & c_{14} \\ c_{21} & c_{22} & c_{23} & c_{24} \\ c_{31} & c_{32} & c_{33} & c_{34} \\ c_{41} & c_{42} & c_{43} & c_{44} \end{bmatrix} = \begin{bmatrix} 0.98 & 0.10 & 0.10 & 0.10 \\ 0.11 & 0.98 & 0.11 & 0.10 \\ 0.10 & 0.10 & 0.98 & 0.10 \\ 0.11 & 0.11 & 0.11 & 0.98 \end{bmatrix}$$

Figure R2.15. The cross-talk values of four users.

Changes:

We have added the following sentences to clarify the performance of our algorithm compared with other methods.

- Lines 218~223, page 8: **It is worth noting that the discrete optimization algorithm remains available to larger-scale arrays owing to its excellent properties, including low complexity and fast convergence (see Supplementary Note 6 for more details). We also provide the compare-and-contrast study with other algorithms reported in the field of DIM, the definition of the performance metrics, and the discussion on the scalability to demonstrate the advances of our method, as shown in Supplementary Note 6.**

Furthermore, we have added the above reply to Supplementary Note 6.

[R13] M. Wei, H. Zhao, Y. Chen, Z. Wang, T. J. Cui, L. Li, *Applied Physics Letters*. 2023.

[R14] M. Wei, H. Zhao, V. Galdi, L. Li, T. J. Cui, *Nature Electronics*. 2023, 6, 610.

[R15] S. Jacobsson, G. Durisi, M. Coldrey, T. Goldstein, C. Studer, *IEEE Transactions on Communications*. 2017, 65, 4670122.

[R16] Y. Ding, V. F. Fusco, *IEEE Transactions on Antennas and Propagation*. 2014, 62, 2745.

[R17] R. A. Shafik, M. S. Rahman, A. R. Islam, presented at *2006 International Conference on Electrical and Computer Engineering*, December 2006.

Reviewer #2 – Comments 5:

Finally, the authors should investigate the harmonic transmission and show that their scheme is secure against harmonic.

Response:

We thank the reviewer for the suggestive comment, which inspires us to study deeply the secure properties of our proposed scheme against harmonic. We will theoretically demonstrate that our DIM scheme does not generate harmonic signals to target users.

The DIM scheme based on the time-modulated array requires a carefully designed periodic sequence and thereby generates harmonics. On the one hand, the extra harmonic contaminates the valuable spectrum resources. On the other hand, the harmonic carry information as well, which raises risks of information leakage. Although these defects have been mitigated by the recent sideband-free metasurface antennas, they need complex leaky wave antenna and coding designs. ^[R18] In contrast, our DIM scheme utilizes a low-profile programmable metasurface and changes the coding sequence in real time, which corresponds to a control sequence with infinitely long random periods. According to the Fourier theory, the frequency offset of harmonic is zero, thus ensuring that our scheme is secure against harmonic.

Proof: The received signal at moment t is expressed as,

$$E(\theta_k, \varphi_k; t) = e^{j2\pi f_0 t} \sum_{p=1}^P \sum_{q=1}^Q \sqrt{G(\theta_k, \varphi_k)} x_{p,q}(t) e^{j\mathbf{v}_{p,q}^H \mathbf{u}} + n_k, \quad (\text{R2})$$

where f_0 , θ_k , φ_k , $G(\theta_k, \varphi_k)$, $x_{p,q}$, and n_k are the central frequency, the elevation angle, the azimuth angle, the directional gain, the optimized phase response of the element, and the noise with respect to the k th user, respectively. In addition, $\mathbf{v}_{p,q} = [pd_x, qd_y]^T$ and $\mathbf{u} = 2\pi/\lambda [\sin \theta_k \cos \varphi_k, \sin \theta_k \sin \varphi_k]^T$ are the auxiliary vectors to simplify Equation (R2).

For the K -bit phase quantization, the response $x_{p,q}(t)$ at moment t is expressed as,

$$x_{p,q}(t) = \sum_{i=0}^{\infty} a_{p,q}^i \Gamma(t - iT), \quad (\text{R3})$$

where $a_{p,q}^i \in \{1/\sqrt{N} e^{jw_m} \mid w_i = 2\pi \cdot m/2^K, m = 1, 2, \dots, 2^K\}$ is the optimized discrete response of the (p, q) element for generating the desired field in the target direction. The optimized coding sequences are first stored in the FPGA and switched every passing T gap. The rectangular pulse signal $\Gamma(t)$ is

$$\Gamma(t) = \begin{cases} 1, & 0 \leq t \leq T \\ 0, & \text{otherwise} \end{cases}. \quad (\text{R4})$$

We perform the Fourier transform of Equation (R2) and obtain the spectral response of the received signal, namely,

$$E(\theta_k, \varphi_k; f) = \sum_{p=1}^P \sum_{q=1}^Q \sqrt{G(\theta_k, \varphi_k)} x_{p,q}(f - f_0) e^{jv_{p,q}^H} + n_k(f). \quad (\text{R5})$$

In the above equation, the spectral response of the response $x_{p,q}(t)$ is

$$x_{p,q}(f) = \sum_{i=0}^{\infty} a_{p,q}^i \Gamma(f) e^{-j2\pi f iT}, \quad (\text{R6})$$

where the spectral response of the pulse signal $\Gamma(t)$ is

$$\Gamma(f) = \int_{-\infty}^{\infty} \Gamma(t) e^{-j2\pi f t} dt = \frac{\sin(\pi f T)}{\pi f}. \quad (\text{R7})$$

It is noticed that the spectral response of the received signal at the harmonic frequency $f_0 + nf_T$ ($f_T = 1/T$) is

$$\begin{aligned} E(\theta_k, \varphi_k; f_0 + nf_T) &= \sum_{p=1}^P \sum_{q=1}^Q \sqrt{G(\theta_k, \varphi_k)} x_{p,q}(nf_T) e^{jv_{p,q}^H} + n_k(f + nf_T) \\ &= \sum_{p=1}^P \sum_{q=1}^Q \sqrt{G(\theta_k, \varphi_k)} e^{jv_{p,q}^H} \sum_{i=0}^{\infty} a_{p,q}^i \Gamma(nf_T) e^{-j2\pi n f_T i T} \\ &\quad + n_k(f + nf_T), \quad n \neq 0. \quad (\text{R8}) \\ &= \sum_{p=1}^P \sum_{q=1}^Q \sqrt{G(\theta_k, \varphi_k)} e^{jv_{p,q}^H} \sum_{i=0}^{\infty} a_{p,q}^i \frac{\sin(n\pi f_T T)}{n\pi f_T} e^{-j2\pi n f_T i T} \\ &\quad + n_k(f + nf_T) \\ &= n_k(f + nf_T) \end{aligned}$$

It is worth emphasizing that the value of the harmonic frequency is determined according to the definition of the time-modulated array or time-coding metasurface ^[R19]. Equation (R8) indicates that the strength of the received signal in the target direction at the harmonic frequency $f_0 + nf_T$ is the noise, which demonstrates that our proposal is secure transmission against harmonic.

To demonstrate the unique secure feature against harmonic, we perform simulations of the dual-channel 8PSK configuration. The two users are at $(\theta, \varphi) = (40^\circ, 0^\circ)$ and $(\theta, \varphi) = (25^\circ, 90^\circ)$, respectively. The switching time of FPGA is 10 us (i.e., the frequency offset is $f_T = 100$ KHz in according with the above definition) and the signal length is 5 ms. The received signals are calculated using Equation (R4) and normalized, and the SNR is 15 dB.

As shown in Figure R2.16, the received signals of user 1 at the central frequency f_0 have the characteristic of the 8PSK symbols, in which the magnitudes fluctuate around 1 and the phases are distributed in the vicinity of the eight discrete values (i.e., $-180^\circ, -135^\circ, -90^\circ, -45^\circ, 0^\circ, 45^\circ, 90^\circ$, and 135°). The signals received by user 2 are demonstrated in Figure R2.17, in which the magnitude values are relatively lower than those of user 1 and fluctuate between 0.6~1. However, the phase values have a form similar to the 8PSK symbols, which implies that the fluctuations of magnitude do not affect the signal recovery. Furthermore, we present the signals received by user 1 at harmonic $f_0 + f_T$ and $f_0 + 2f_T$ in Figures R2.18a and R2.18b, respectively, where the magnitude values are low and the phase distributions are disorganized. The above results demonstrate that the received signals at harmonic are mainly the noise and the information is unable to be transmitted.

Figure R2.16. The signals received by user 1 with about 5 ms. The user 1 is located at $(\theta, \varphi) = (40^\circ, 0^\circ)$ and the frequency is f_0 . The signals within 0.85-2.15 ms are enlarged on the right.

Figure R2.17. The signals received by user 2 with about 5 ms. The user 2 is located at $(\theta, \varphi) = (25^\circ, 90^\circ)$ and the frequency is f_0 . The signals within 0.85-2.15 ms are enlarged on the right.

Figure R2.18. (a) The signals received by user 1 within 5 ms and the signal frequency is $f_0 + f_T$. Further, the signals within 0.85-2.15 ms are enlarged on the right. (b) The signals at harmonic $f_0 + 2f_T$.

Changes:

We have added the following sentences to clarify the unique feature of directional security against harmonic.

- Lines 101~103, pages 3~4: **Using the optimized non-periodic time-space coding sequences, our scheme is directional secure against harmonic (see Supplementary Note 2 for more details).**

Furthermore, we have added the proof and simulations about the security against harmonic Supplementary Note 2.

[R18] G.-B. Wu, J. Y. Dai, Q. Cheng, T. J. Cui, C. H. Chan, *Nature Electronics*. **2022**, 5, 808.

[R19] J. Zhao, X. Yang, J. Y. Dai, Q. Cheng, X. Li, N. H. Qi, J. C. Ke, G. D. Bai, S. Liu, S. Jin, A. Alu, T. J. Cui, *National Science Review*. **2019**, 6, 231.

Reviewer #2 – Comments 6:

Unfortunately, given the limited novelty of this work, the lack of extensive experiments, and the rich literature in this domain, I recommend rejecting the paper.

Authors Response:

We are especially grateful for the suggestive questions of the reviewer, which definitely and considerably help us enrich the content of our work and improve the quality of the manuscript. We have compared our work with the existing literatures and highlighted our differences and advantages. We hope that our replies and revisions have stated the novelty of our work and enriched the content of experiments.

General comments from Reviewer #3

A directional information modulation scheme is proposed with use of a programmable metasurface assisted by algorithms that optimize the digital coding sequences with information symbols being transmitted on multi-directional beams. Several types of modulations are tried and the BER is suppressed along certain directions. A prototype has been fabricated and measurements indicate that there are certain angles at which the transmissivity gets maximized for all the adopted symbols. The obtained constellation diagrams also validate the correct operation of the device and the proper distribution of power across the space of elevation angle.

The paper is well-written and deals with a very important topic. However, it cannot be published at such a prestigious publication venue as Nature Communications unless extensive improving modifications are performed.

Response:

We thank the reviewer for the positive comments. The insightful comments are very constructive for further improvement of our work. We hope that the following responses are sufficient for highlighting the value of our work.

Specific comments from Reviewer #3

Reviewer #3 – Comments 1:

How the metasurface knows and the position of eavesdropper? What detection process is hidden behind? This information is critical for the efficient operation of the device since the directions of distorted constellation diagrams should be determined and taken into account by the programmable metasurface. The authors should clarify this issue further in a revised version of their manuscript.

Response:

We appreciate the reviewer for the enlightening comments. We should state that our scheme provides directional information security for the desired users, and yields distorted constellation diagrams in other directions. The distorted diagrams received by eavesdroppers are determined by the random noise. Therefore, the transmitter should know the precise location of users while not the position of eavesdroppers. In the previous manuscript, we assume that the transmitter knows the direction of users. In the revised version, we realize two-dimensional sum- and difference- beams by the metasurface and utilize the amplitude ratio method to estimate the direction of arrival (DOA) according to the wave radiated by the user.

The principle of sum- and difference- beam angle measurement is shown in Figure R3.1. The metasurface operates in receiving mode when the target user radiates incoming wave. Meanwhile, the metasurface scans in the whole space and generates sum, azimuth difference, and elevation difference beams respectively in three consecutive time slots by switching the coding sequences with a microcontroller unit (MCU). The power of sum beam $P_{\Sigma}(\theta, \varphi)$ reaches the maximal in the direction of the incoming wave. The power of difference beam $P_{\Delta}(\theta, \varphi)$ reaches the

minimal in the direction of the incoming wave. We define the magnitude ratio of the power of the sum- and difference- beams as

$$r(\theta, \varphi) = \frac{P_{\Sigma}(\theta, \varphi)}{P_{\Delta}(\theta, \varphi)}. \quad (\text{R1})$$

According to the physical meaning of $r(\theta, \varphi)$, the ratio reaches the maximal in the direction of the incoming wave. Therefore, we can plot the curve of $r(\theta, \varphi)$ and get the estimated direction (θ, φ) of the incoming wave.

To demonstrate the ability of measuring DOA of the incoming wave, we first conducted experiments to show that our programmable metasurface can generate the sum, azimuth difference, and elevation difference beams with good performance. We then calculated the curve of $r(\theta, \varphi)$ using Equation (R1) and estimated the DOA of the incoming wave by finding the maximal $r(\theta, \varphi)$.

Figure R3.1. The conceptual diagram of sum and difference beam angle measurement.

1. The measured sum- and difference- beams

Figures R3.2a-3.2c demonstrate the measured sum, azimuth, and elevation difference beams, respectively. The main-lobe angle of the sum beam is -1° . The null angles of the azimuth and elevation difference beams are 0° and -1° , respectively. The results show good agreement with the excitation. As shown in Figure R3.3, the results of the sum beam and the azimuth difference beam is only given for the case of the direction of 30° due to the restriction of measurement conditions. However, the elevation difference beam can also be generated using the programmable metasurface.

Figure R3.2. Measurements for the direction of 0° . (a) The sum beam. (b) The azimuth difference beam. (c) The elevation difference beam.

Figure R3.3. Measurements for the direction of 30° . (a) The sum beam; (b) The azimuth difference beam.

2. Measuring DOA of the incoming wave

The experimental scene for measuring DOA of the incoming wave is shown in Figure R3.4. The user equipped with the proposed metasurface can flexibly generate the incoming waves with different angles. The metasurface in the base station operates in receiving mode and scans in the whole space. During the scan, the metasurface generates sum, azimuth difference, and elevation difference beams respectively in three consecutive time slots by switching the coding sequences

using MCU. It is worth noting that the beam scanning rate of the metasurface (about 1° per second) is much lower than the switching rate of MCU, thus the metasurface approximately obtains different data of magnitudes from the three beams simultaneously.

Figure R3.4. The photo of the experimental scene for measuring DOA of the incoming wave.

Figures R3.5a and R3.5b demonstrate the measured magnitude ratio in the azimuth and elevation plane, respectively. The directions of the maximal ratio in the azimuth and elevation plane are 0° and -1° , respectively. The estimated angle $(\theta, \varphi) = (0^\circ, -1^\circ)$ is in good agreement with the true direction of the incoming wave, which demonstrates the feasibility of our scheme. Figure R3.6 demonstrate the results for the incoming wave of 30° . The estimated angle is $\theta = 31^\circ$, which matches well with the ground truth.

Figure R3.5. Magnitude ratio for the incoming wave of 0° . (a) The magnitude ratio of the sum and azimuth

difference beam. (b) The magnitude ratio of the sum and elevation difference beam.

Figure R3.6. The magnitude ratio of the sum and azimuth difference beam for the incoming wave of 30°.

Changes:

We have added the following sentences to clarify the detection of the desired users in the revised manuscript.

Lines 396-407, page 18: Before directly transmitting digital information, the PM first operate in receiving mode and measures its relative position with respect to the desired user, which radiates incoming wave. The metasurface scans in the whole space and generates sum, azimuth difference, and elevation difference beams respectively in three consecutive time slots by switching the coding sequences using MCU. The power of sum beam $P_{\Sigma}(\theta, \varphi)$ reaches the maximal in the direction of the incoming wave. The power of difference beam $P_{\Delta}(\theta, \varphi)$ reaches the minimal in the direction of the incoming wave. We define the magnitude ratio of the power of the sum- and difference- beams as

$$r(\theta, \varphi) = \frac{P_{\Sigma}(\theta, \varphi)}{P_{\Delta}(\theta, \varphi)}. \quad (9)$$

According to the physical meaning of $r(\theta, \varphi)$, the radio reaches the maximal in the direction of the incoming wave. Therefore, we can plot the curve of $r(\theta, \varphi)$ and get the estimated direction (θ, φ) of the incoming wave (see Supplementary Note 11 for more details).

Furthermore, we have added the measurements and analysis of the detection of desired users to Supplementary Note 11.

Reviewer #3 – Comments 2:

*How the performance of the device can be enhanced in the presence of screens that **increase the directivity along specific angles** [1,2]? Can such components be used to further boost the operation of the described metasurface?*

[1] Lan et al., Real-time programmable metasurface for terahertz multifunctional wave front engineering, Light, 2023.

[2] Valagiannopoulos et al., Metasurface-enabled interference mitigation in visible light communication architectures, Journal of Optics, 2019.

Authors Response:

Thank you very much for your valuable suggestions and the interesting papers. The placement of a well-designed meta-lens in front of the metasurface can effectively improve the directivity along specific directions and mitigate the interference of ambient signals in multi-link environments. The metasurface-enabled devices with excellent performance can play a greater role in the terahertz and visible-light band, due to the high path loss.^[1,2]

Following your prompt, we have designed a meta-lens as a screen that increases the directivity of the metasurface along specific angles. However, the meta-lens is unable to boost the performance of our metasurface-based DIM system. The reason behind this is that the passive meta-lens is carefully optimized according to a certain symbol, which may lead to bad performance for other symbols due to the magnitude and phase mismatch in the meta-lens plane. Below we will give more details and present simulation results to prove our point, which includes the design of the meta-lens, the performance contrast between adding and not lens, and the influence on the DIM system.

The classical method to design a meta-lens is to compensate for the phase on the lens plane and

make the radiation field a plane wave ^[R1]. The starting point is to extract the distribution of fields on the meta-lens plane, and then carefully arrange the meta-atoms to compensate for the phase. The sketch of the element in the meta-lens is illustrated in Figure R3.5a, which consists of one substrate and two identical metal layers connected by vias. ^[R2] The transmission response of the meta-atom mainly depends on the parameters l and d . The parameter sweeping simulations are conducted and the results are demonstrated in Figure R3.5b. The phases cover $0\sim 320^\circ$ and the magnitudes are larger than -1.8 dB. Therefore, the meta-atom provides large phase coverage and high transmission efficiency.

The configuration in the joint simulation of the metasurface and meta-lens is illustrated Figure R3.6a, where the meta-lens compensates for the phase of fields emitted by the metasurface and increases the directivity along the specified angles. The meta-lens consists of 15×15 meta-atoms and is 52 mm (about 4λ) in front of the metasurface. The meta-lens is 104×104 mm (about $8\lambda\times 8\lambda$) being four times that of the metasurface. First, the metasurface is configured to work in the single-channel 8PSK scheme and yield one symbol coded as “000” along the direction of $\theta = 30^\circ$. After obtaining the magnitude and phase distributions of the field on the meta-lens plane that are shown on the left and middle of Figure R3.6b, respectively, the meta-atoms are arranged according to the principle of phase compensation. The comparison of the gain of the metasurface without and with meta-lens is demonstrated on the right of Figure R3.6b. The gain increases 1.5 dBi along $\theta = 30^\circ$. The influence of multiple reflections between the metasurface and the meta-lens results in a low gain improvement.

However, the previously optimized meta-lens may deteriorate the performance due to phase mismatches when the metasurface transmits another symbol coded as “001” along the direction of $\theta = 30^\circ$. In this case, the magnitude and phase distributions of the field shown on the left and middle of Figure R3.6c are distinct from the previous scene. The gain with the meta-lens is 1.2 dBi lower than the case of no lens, which shows the deteriorative performance.

We also demonstrate the performance comparison of the DIM, as shown in Figure R3.7. We

quantify the performance of the DIM in terms of the error vector magnitude (EVM) that is commonly used in wireless communications to evaluate the quantity of the received symbols. The definition and the physical meaning of EVM have been stated in the revised manuscript. We observe that with the meta-lens, the EVM is increased near the direction of $\theta = 30^\circ$, which indicates that the overall performance deteriorates in the single-channel 8PSK scheme.

Figure R3.5. The elements in the meta-lens. (a) Sketch of the meta-atom, in which the substrate and the metal are represented by the blue and yellow colors, respectively. The substrate is RO4350B ($\epsilon_r = 3.66$ and $\tan \delta = 0.004$), and the metal layers are rectangle with 0.018 mm thick. The diameters of the vias are 0.54 mm. The other parameters are $p_x = 135$ mm, $p_y = 13.5$ mm, $w = 1.8$ mm, $h = 3.86$ mm, and $d = 0.28l$. (b) Magnitudes and phases of the transmission response of the elements as functions of patch length l .

(a)

(b)

(c)

Figure R3.6. The joint simulation of the metasurface and the meta-lens. (a) Configuration for the joint simulation. (b) The metasurface sends one of the 8PSK symbol coded “000” to the user that located at $\theta = 30^\circ$. The left and middle figure: magnitude and phase distribution of fields on the meta-lens plane. The right figure: far-field pattern of the metasurface with and without lens. (c) The metasurface sends another symbol coded “001” to the user. The configuration is the same as before.

Figure R3.7. Performance comparison of the metasurface without and with meta-lens.

[R1] M. K. Chen, Y. Wu, L. Feng, Q. Fan, M. Lu, T. Xu, D. P. Tsai, *Advanced Optical Materials*. 2021, 9, 2001414.

[R2] J. W. Wu, Z. X. Wang, R. Y. Wu, H. Xu, Q. Cheng, T. J. Cui, *IEEE Transactions on Antennas and Propagation*. 2023, 71, 6652.

Reviewer #3 – Comments 3:

The authors should provide a compare-and-contrast study with competing structures achieving security in communications with metasurfaces programmable [3] or not [4]. A comparison regarding the balance between the simplicity of the respective design and the effectiveness of the overall communication device.

[3] Saifullah et al., Dual-band multi-bit programmable reflective metasurface unit cell: design and experiment, Optics Express, 2021.

[4] Tsiftsis et al., Metasurface-Coated Devices: A New Paradigm for Energy-Efficient and Secure 6G Communications, IEEE Vehicular Technology Magazine, 2022.

Response:

We thank the reviewer for recommending the interesting references.

The cited work in [3] reported a 2-bit and dual-band metasurface unit with great performance. However, we did not find its utilization for ensuring the security of wireless communication over the open literatures.

The cited work in [4] proposed a passive metasurface-coated device that can concentrate the power of ambient signals on the transmitter or receiver, which hopefully realizes energy efficiency, energy harvesting, and ultralow-power transmissions, after carefully reading. Furthermore, the physical layer security (PLS) is ensured by increasing the power disparity between the receiver with the metasurface-coated device and scattered eavesdroppers. The device mitigates the critical problem of the tradeoff between ultralow-power transmissions and secrecy capacity for the first time, which is a great improvement compared with the traditional beamforming techniques. Although the power received by eavesdroppers is very low, the signal structure is still undistorted. Therefore, the security risk of the above method exists when an eavesdropper equipped with a sensitive receiver can intercept the information in theory. Furthermore, the metasurface-coated device is passive, thus it is unable to realize real-time spatial wave manipulations and directional information modulation.

Following the reviewer's suggestion about the compare-and-contrast study, we list several approaches that realize wireless communication security, including the conventional method using beamforming-based techniques, passive metasurface-coated devices, phased arrays, time-domain coding metasurfaces, time-space-domain coding metasurfaces, and the proposed scheme. To address the novelty and significance, the comparison of our work in contrast to other approaches that achieving PLS are listed in terms of directional security, system complexity, performance, realization strategy, advantages, and disadvantages, as summarized in Table R3.1. In particular, we do not give the corresponding performance comparison since the beamforming-based techniques and the metasurface-coated devices are unable to achieve directional security.

Compared with other existing approaches, our proposal has the following advantages:

- Our proposal can achieve two-dimensional and multi-channel (four-channel modulation has been realized and even more channels can be achieved if the scale is increased) DIM.
- Our proposal can achieve different high order modulations, such as 8PSK, 16QAM, and 64QAM schemes.
- Our proposal is secure against harmonic compared with time-coding digital metasurfaces.
- Our proposal can serve as a low-profile transmitter or receiver, which greatly reduces system complexity.

1. Beamforming-based techniques

Beamforming (BF) based techniques have been widely employed for the PLS of wireless communications. The techniques are designed either to enhance the power at the receiver or to degrade the wiretap channel used by the eavesdropper. Additionally, BF often adopts other PLS techniques involving the hybridization of four methods ^[R3-R5] (1) secure BF, (2) artificial noise BF, (3) cooperative jamming-aided BF, and (4) joint channel coding with BF.

In summary, these methods increase the SNR difference between the receiver and eavesdropper and they are motivated by Wyner's wiretap model. However, the information is still encoded in the time domain. The broadcast nature of wireless communication channels allows the signals in all directions to have an identical structure, only with different power levels. Therefore, an eavesdropper equipped with a sensitive receiver can intercept the information in theory. As a result, the demand for information security necessitates the idea of directional communications.

2. Metasurface-coated devices

The passive metasurface-coated devices can concentrate the power of ambient signals on the transmitter or receiver, which hopefully realizes energy efficiency, energy harvesting, and ultralow-power transmissions. ^[R6, R7] Furthermore, the PLS is ensured by increasing the power disparity between the receiver with the metasurface-coated device and scattered eavesdroppers. The device mitigates the critical problem of the tradeoff between ultralow-power transmissions and secrecy capacity for the first time, which is a great improvement compared with the

traditional beamforming techniques. However, although the power received by eavesdroppers is very low, the signal structure is still undistorted. Therefore, the security risk of the above method exists when an eavesdropper equipped with a sensitive receiver can intercept the information in theory. Furthermore, the metasurface-coated device is passive, thus it is unable to realize real-time spatial wave manipulations and directional information modulation.

3. Phased arrays

The phased arrays usually require expensive transmitter and receiver (T/R) components for each radio-frequency channel, which results in a bulky and energy-hungry system. In the first realization of DIM based on phased arrays, the single-channel mode was realized using the QPSK modulation. ^[R8] The phase shifts of each element are optimized based on a genetic algorithm to minimize the bit error rate (BER) of the desired direction while maximizing the BER elsewhere. However, the system complexity is high, and it is hard to extend larger scales due to the expensive T/R components.

4. Time-domain coding metasurfaces

The time-domain coding metasurface (TDCM) is constructed by integrating the electronically controllable components, such as the varactor or positive-intrinsic-negative (PIN) diodes, with the low-cost metal patches. The realization of high-order modulations is to optimize the periodic sequences to change the magnitude and phase of the harmonic wave, respectively. Therefore, the TDCM-based DIM scheme is low-cost and power-efficient, and the complexity mainly depends on the optimization of the periodic sequences. However, the TDCM employs the harmonic of the electromagnetic (EM) waves, which will result in spectra pollution and is unable to ensure directional secure transmission due to the lack of space coding.

We here present two representative works about the TDCM-based DIM scheme. A single-channel 16QAM mode is implemented based on a varactor-loaded metasurface with 8×16 elements. ^[R9] The phases and magnitudes of the harmonic signals are changed by the introduction

of delay time and the relative phase difference of the pulse wave. The second example realizes the single-channel 256QAM transmission based on a 1-bit metasurface with 56×20 elements. ^[R10] The phases and magnitudes of the harmonic signals are changed by the introduction of delay time and duty ratio of the pulse wave. From the above examples, we find that the harmonic signal of arbitrary modulation can be realized by elaborately changing the form of the pulse wave in theory. However, the corresponding changes also occur at other-order harmonic at the same time, which means there exists the risk of information leakage.

5. Time-space-domain coding metasurfaces

Time-space-domain coding metasurfaces (TSDCM) have a similar structure to TDCM, but the coding strategy is different. First, the TSDCM-based DIM scheme does not utilize periodic sequences, which means that the system does not generate the harmonic. We also prove that the received signals in the harmonic are noise. Second, the TSDCM-based DIM scheme supports the manipulation of multiple spatial beams and temporal information with directional security.

We here present two representative examples that utilize the TSDCM-based DIM scheme. The hardware is a 1-bit programmable metasurface with different scales and the method is a modified Gerchberg-Saxton (GS) algorithm. ^[R11, R12] They can realize the two-dimensional and multi-channel transmission. However, these implementations are facing several limitations. First, they only transmit signals. This is incompatible with the operation of traditional wireless systems, where the base station or terminal device can both transmit and receive information. Second, the metasurface-based transmitter typically needs external RF sources and has a large profile, which is inconvenient for space-constrained applications such as wearable Internet of Things (IoT) devices, vehicles, and aircraft. Moreover, the modulation are only BPSK or QPSK, which suffer from a lack of high-order modulation and QAM schemes that carry more information capacities.

6. The proposed DIM scheme

From the perspective of the coding strategy, our proposed scheme can also be classified as

TSDCM. However, one significant difference is that our system is reciprocal, which means that it can serve as a low-profile transmitter and receiver simultaneously. The scheme supports the identical metasurface device to complete the signal radiating and receiving, thus making the system simpler and easier to maintain.

We note that the main limitation of the TSDCM scheme is the finite phase-shift states of each meta-atom. Therefore, an efficient optimization algorithm for spatial coding sequences is needed. We have developed the ADMM-based discrete optimization algorithm and experimentally validated the four-channel high-order modulation. We have also compared it with other methods reported in the field of DIM, such as the GA, modified GS, and squared-infinity norm Douglas-Rachford splitting. The simulations demonstrate that our method has faster convergence, better performance, and scalability. Although we conducted the experimental validation on a small-scale metasurface (8×8), our design can be easily extended to a larger scale, and the proposed algorithm is more competitive in this scenario.

Table R2.1. Comparison with reported schemes achieving physical-layer security

	Directional security	System Complexity	Performance	Realization strategy	Advantages	Disadvantages
Beamforming-based techniques	No	--	--	--	--	--
Metasurface-coated devices	No	--	--	--	--	--
Phased arrays	Yes	High: the utilization of expensive T/R components	Single-channel QPSK modulation ^[R8]	A four-element linear array; GA algorithm	The continuous phase shifts	Poor scalability due to expensive cost
Time-domain coding metasurfaces	No: Information leakage on harmonics	Low: optimization of periodic sequences	Single-channel 16QAM ^[R9]	A varactor-loaded metasurface with 8×16 elements; Optimization of periodic waves	The continuous phase shifts and magnitudes;	Unsecure on harmonic; Single-channel mode;
			Single-channel 256QAM ^[R10]	A 1-bit metasurface with 56×20 elements; Optimization of		

		periodic waves				
Time-space-domain coding metasurfaces	Yes	Low: optimization of coding sequences	Dual-channel QPSK modulation ^[R11]	A 1-bit metasurface with Modified GS algorithm	Secure against harmonic; Two-dimensional and multiple-channel mode.	Finite phase shifts; Low-order modulation;
			Three-channel BPSK modulation ^[R12]	A 1-bit metasurface with Modified GS algorithm		
Our work	Yes	Low: optimization of coding sequences	Four-channel 8PSK, 16QAM and 64QAM	A 2-bit metasurface with ADMM-based algorithm	Secure against harmonic; Two-dimensional and multiple-channel mode; Serve as transmitter or receiver with low profile; High-order modulation;	Finite phase shifts; Small-scale array

Changes:

We have modified the abstract and introduction to clarify the value and novelty of our work. We have added the following papers as references [11][12] in the revised manuscript.

[11] T. A. Tsiftsis, C. Valagiannopoulos, H. Liu, A. A. A. Boulogeorgos, N. I. Miridakis, *IEEE Vehicular Technology Magazine*. **2022**, 17, 27.

[12] D. Tulegenov, C. Valagiannopoulos, *Journal of Applied Physics*. **2022**, 131.

Furthermore, we have added the above reply to Supplementary Note 1.

[R3] Y. Liu, J. Li, A. P. Petropulu, *IEEE Transactions on Information Forensics and Security*. **2013**, 8, 682.

- [R4] Q. Wang, F. Zhou, R. Q. Hu, Y. Qian, *IEEE Transactions on Wireless Communications*. **2021**, 20, 2592.
- [R5] R. M. Yamada, A. O. Steinhardt, L. Mili, *IEEE Transactions on Wireless Communications*. **2017**, 16, 8026.
- [R6] T. A. Tsiftsis, C. Valagiannopoulos, H. Liu, A. A. A. Boulogeorgos, N. I. Miridakis, *IEEE Vehicular Technology Magazine*. **2022**, 17, 27.
- [R7] D. Tulegenov, C. Valagiannopoulos, *Journal of Applied Physics*. **2022**, 131.
- [R8] M. P. Daly, J. T. Bernhard, *IEEE Transactions on Antennas and Propagation*. **2009**, 57, 2633.
- [R9] J. Y. Dai, W. Tang, L. X. Yang, X. Li, M. Z. Chen, J. C. Ke, Q. Cheng, S. Jin, T. J. Cui, *IEEE Transactions on Antennas and Propagation*. **2020**, 68, 1618.
- [R10] M. Z. Chen, W. Tang, J. Y. Dai, J. C. Ke, L. Zhang, C. Zhang, J. Yang, L. Li, Q. Cheng, S. Jin, T. J. Cui, *National Science Review*. **2022**, 9, nwab134.
- [R11] H. Zhao, Y. Shuang, M. Wei, T. J. Cui, P. D. Hougne, L. Li, *Nature Communications*. **2020**, 11, 3926.
- [R12] X. Wan, C. Xiao, H. Huang, Q. Xiao, W. Xu, Y. Li, J. Eisenbeis, J. Wang, Z. Huang, Q. Cheng, S. Jin, T. Zwick, T. Cui, *Engineering*. **2022**, 8, 86.

REVIEWER COMMENTS

Reviewer #1 (Remarks to the Author):

1. In the Response letter, the proposed DIM system is claimed as reciprocal to simultaneously serve as transmitter and receiver. A reciprocal device means that the relationship between its input and output signal is linear, which enables forward and reverse signal conversion. However, the designed metasurface cannot obtain the input sine wave signal or recover the transmitted information by itself when it serves as a receiver. Therefore, it's not reasonable to call it a reciprocal device. Therefore, the unique features and distinguished advances related to its reciprocity are groundless. The authors should think about this more carefully and explain it in detail.

2. In the four-channel cases, the transmission effectiveness of different channels shows large difference, how to further improve this problem?

Reviewer #2 (Remarks to the Author):

Thanks for your detailed response.

The manuscript has improved, particularly the discussions on cross-talks add value. However, my main concern is still the novelty of this work. The authors claim that their main contribution over state of the art space-time-coding architectures are the following

1. "one significant difference is that our system is reciprocal, which means that it can serve as a low-profile transmitter and receiver simultaneously."

All metasurfaces are indeed reciprocal. this is not a significant contribution. Further, the metasurface used in this paper was already published in IEEE transactions on Antennas and Propagation. "Low-profile" is a qualitative term. If the authors are claiming novelty in the surface design (beyond the IEEE tran. paper) they have to specifically explain the hardware performance metrics.

2. "We note that the main limitation of the TSDCM scheme is the finite phase-shift states of each meta-atom. Therefore, an efficient optimization algorithm for spatial coding sequences is needed. We have developed the ADMM-based discrete optimization algorithm a..."

While this is true, designing this algorithm alone is not enough contribution for such a prestigious journal.

Reviewer #3 (Remarks to the Author):

The authors have made a decent effort to address the reviewers points. The paper can be published at Nature Communications.

Response Letter to Reviewers

We sincerely appreciate the further constructive comments from all reviewers, which help us improve the quality of the manuscript significantly. Below are our responses (in regular font) to each comment (*in italics and blue font*). We have revised the manuscript accordingly and **highlighted** all changes in the revised documents. We hope that the response letter has clarified the reviewers' concerns. We would be glad to respond to any further questions and comments. Thank you again for your consideration.

Specific comments from Reviewer #1

Reviewer #1 – Comments 1:

1. In the Response letter, the proposed DIM system is claimed as reciprocal to simultaneously serve as transmitter and receiver. A reciprocal device means that the relationship between its input and output signal is linear, which enables forward and reverse signal conversion. However, the designed metasurface cannot obtain the input sine wave signal or recover the transmitted information by itself when it serves as a receiver. Therefore, it's not reasonable to call it a reciprocal device. Therefore, the unique features and distinguished advances related to its reciprocity are groundless. The authors should think about this more carefully and explain it in detail.

Response:

We sincerely thank the reviewer for the insightful comments on the reciprocity of our DIM scheme. We are sorry that we focused on the analysis of measurements and did not discuss the reason for the reciprocity of the system in details in the previous response letter. We agree with the reviewer's comment on the reciprocal devices, which can transmit and receive signals. We will show that our system can realize such ability in the following discussion.

In our DIM scheme, digital information is directly encoded in electromagnetic (EM) waves and recovered by comparing the received fields with the corresponding constellation symbols. Reviewing the communication process, the input signal is represented by the constellation

symbol, and then the corresponding coding pattern is generated by our proposed discrete optimization algorithm. Finally, the programmable metasurface generates the radiation fields precisely in the desired directions, as indicated by the specified constellation diagram. Though the carrier of the signal is the sine wave, one can recover the transmitted information using a device that identifies the magnitude and phase of the received field.

Our hardware design integrates the advantages of programmable metasurface and phased array antennas. Therefore, it can transmit and receive the electromagnetic wave, which is guaranteed by the famous Rayleigh-Carson reciprocity theorem.^[R1] Specifically, when operating in the receiving mode, the main lobe beam of the metasurface is aligned with the transmitter by optimizing the coding pattern to ensure a high signal-to-noise ratio.

The above elaboration demonstrates that our DIM scheme can transmit and receive the digital information using an identical device, which is consistent with the reviewer's comment on the reciprocal devices. The main difference between the transmitting and receiving modes is the coding pattern. The former is updated in real time with the input constellation symbol, while the latter is fixed to ensure the high signal-to-noise ratio.

Changes:

We have added the following sentences to clarify the reciprocity of our DIM scheme.

- Lines 138~142, page 5: According to the principle of the DIM scheme, the hardware can work in the transmitting- or receiving-mode (see Supplementary Note 3 for more details). The main difference between the transmitting and receiving modes is the coding pattern. The former is updated in real time with the input constellation symbol, while the latter is fixed to ensure a high signal-to-noise ratio.

[R1] R.F. Harrington, *Time-Harmonic Electromagnetic Fields*. New York: McGraw-Hill, 1961.

Reviewer #1 – Comments 2:

2. *In the four-channel cases, the transmission effectiveness of different channels shows large*

difference, how to further improve this problem?

Response:

We sincerely thank the reviewer for the comments. We also notice that the performance of some channels shows a larger difference in the four-channel cases, especially for the 64QAM scheme. There are three possible reasons for this problem, followed by the corresponding solutions.

The first reason is the impact of phase accuracy. Due to the restriction of finite phase states, large quantization errors may occur in certain desired directions. Therefore, it is expected to increase the phase quantization level of the metasurface in our future work. We have conducted simulations with different quantization levels in the four-channel cases, as shown in Figure S8 in Supplementary Note 6. The results indicate significant improvements when the quantization level increases from 2 bits to 3 bits.

The second reason is the crosstalk between different channels. The measured cross-talk values in the four-channel 16QAM experiment are relatively high, indicating a serious interference between channels. To reduce the interference, one can increase the scale of metasurface, and thus effectively reduce the width of radiation beam. To prove this, we simulate the four-channel 16QAM scheme using a metasurface scale with 16×16 elements. Figure R1 shows that the cross-talk values between different users are lower (maximum of -19.2 dB) than those in Figure S21 in Supplementary Note 8 for the metasurface scale with 8×8 elements.

The third reason is the mutual coupling between adjacent elements, which deteriorates the radiation pattern of each element and causes the channel vector (Equation (2) in the main text) to be inconsistent with reality. Specifically, we use the ideal point-source model and ignore the influence of the mutual coupling. Recently, we have proposed a macroscopic model to take the mutual coupling between adjacent elements into account.^[R2] We will use this model to modify the channel vector calculation (Equation (2) in the main text) to reduce performance variations between different channels in our future work.

Figure R1. The cross-talk values of four users with the metasurface scale of 16×16 elements.

[R2] R. W. Shao *et al.*, *National Science Review*. **2023**, 11(3): nwa299.

General comments from Reviewer #2

The manuscript has improved, particularly the discussions on cross-talks add value. However, my main concern is still the novelty of this work.

Response:

We sincerely thank you for your positive recognition of our efforts. We hope that the following responses can highlight the novelty of our work.

Specific comments from Reviewer #2

Reviewer #2 – Comments 1:

"one significant difference is that our system is reciprocal, which means that it can serve as a low-profile transmitter and receiver simultaneously."

All metasurfaces are indeed reciprocal. this is not a significant contribution. Further, the metasurface used in this paper was already published in IEEE transactions on Antennas and Propagation. "Low-profile" is a qualitative term. If the authors are claiming novelty in the surface design (beyond the IEEE tran. paper) they have to specifically explain the hardware

performance metrics.

Response:

We sincerely thank the reviewer for the concerns on the novelty of our metasurface design. We agree with the reviewer's comment on the reciprocity of the metasurface. We should apologize for the misuse of the word "reciprocal", and it would be more accurate to claim our design versatile. Apart from realizing two-dimensional and multi-channel DIM, our system can also receive information using an identical device.

Most metasurface-based communication systems reported in the literature are only exploited for transmitters, while separate systems are needed for recovering the information. This is incompatible with the operation of traditional wireless systems, where the base station or terminal device can both transmit and receive information. It has been experimentally shown that the proposed system is capable of transmitting and receiving the information (please refer to Supplementary Note 3 for further details).

We admit that a programmable metasurface similar to the one used in this manuscript has been published in IEEE Transactions on Antennas and Propagation, but the receiving capability of the design in the current work was established by two important modifications to the programmable metasurface. Firstly, we have improved the feed network and used a Wilkinson power divider with high isolation. In the previous design, the feed network employed a T-type power divider, which made the received signals distorted due to serious interference between different channels. Secondly, we have replaced the varactor with PIN diodes at the feed network. This is necessary due to the limitations of the varactor, which requires high tunable voltages, complex control circuits, and large power supplies. Furthermore, the previous article is focused on hardware design, specifically targeting the basic features the antenna community is interested in, such as two-dimensional wide-angle scanning, high gain, and broad bandwidth. However, the metasurface exploited in this paper is a tool that we leverage to demonstrate secure wireless communications by using two-dimensional and multi-channel DIM. **This is the intriguing feature of the programmable metasurface: the same or similar hardware can produce numerous different functionalities.**

Although the “low profile” is an engineering indicator, it has two benefits. First, the low-profile designs are convenient for space-constrained applications such as wearable Internet of Things (IoT) devices, vehicles, and aircraft. Second, it eliminates the aperture occlusion effect caused by the extended feed horn, as used in the previous designs of the metasurface-based wireless communication systems in Refs. [R3] and [R4].

[R3] X. Wan *et al.*, *Engineering*. **2022**, 8, 86.

[R4] L. Zhang *et al.*, *Nature Electronics*. **2021**, 4, 218.

Figure R2. The representative metasurface-based communication systems with the extended feed horn. The works of (a) Ref. [R1], and (b) Ref. [R2].

Changes:

We have removed the claim of device reciprocity and added the following sentences to clarify the improvements in our hardware design.

- Lines 142-149, page 5: It is worth noting that the receiving capability of the design in the current work was established by two important modifications to the programmable metasurface. Firstly, we improved the feed network and used a Wilkinson power divider with high isolation. In the previous design, the feed network had employed a T-type power divider, which made the received signals distorted due to serious interference between different channels. Secondly, we replaced the varactor with the PIN diodes at the feed network. This is necessary due to the limitations of the varactor, which requires high

tunable voltages, complex control circuits, and large power supplies.

Reviewer #2 – Comments 2:

"We note that the main limitation of the TSDCM scheme is the finite phase-shift states of each meta-atom. Therefore, an efficient optimization algorithm for spatial coding sequences is needed. We have developed the ADMM-based discrete optimization algorithm a..."

While this is true, designing this algorithm alone is not enough contribution for such a prestigious journal.

Response:

We sincerely thank the reviewer for the concerns on the contribution of our work. It is the rigorous standards of reviewers that have made the journal prestigious since its inception, which encourages us a lot.

We admit that one of our contributions is the proposed ADMM-based discrete optimization algorithm. However, the focus of our work is to demonstrate secure wireless communication of **two-dimensional and multi-user DIM**. Our system supports multiple high-order modulation schemes and multi-channel transmissions, and exhibits better performance than the reported works (please see Table S1 in Supplementary Note 1 for more detailed comparisons). It is the good algorithm and improved hardware design that made it a reality.

It should be stressed that the optimization algorithm is the efficient and key tool that we leverage to obtain feasible coding sequences in DIM. It is also a trend that intelligent and efficient algorithms are combined with programmable metasurfaces with excellent performance to enable unprecedented and intriguing functions, such as intelligent imaging and sensing ^[R5,R6], automatic tracking ^[R7], and electromagnetic hacker ^[R8].

[R5] L. Li *et al.*, *Light: Science & Applications*. **2019**, 8, 97.

[R6] H. Y. Li *et al.*, *Patterns*. **2020**, 1, 100006.

[R7] W. Li *et al.*, *Nature Communications*. **2023**, 14, 989.

[R8] M. Wei *et al.*, *Nature Electronics*. **2023**, 6, 610.

General comments from Reviewer #3

The authors have made a decent effort to address the reviewers points. The paper can be published at Nature Communications.

Response:

We sincerely thank you for your positive recognition of our efforts.

REVIEWER COMMENTS

Reviewer #1 (Remarks to the Author):

The authors have responded to my comments properly in this round of revision. I think they need to make further minor revision to explain their response to my second comment somewhere in the main text of the manuscript.

Reviewer #2 (Remarks to the Author):

While I appreciate the authors' effort to improve their manuscript, I still have concerns regarding the novelty. The space-time coding (or directional modulation) and its application for secure connectivity have been published in several other works. The claimed delta seems to be the following: a programmable meta-surface (instead of a time-modulated phased array) can realize multi-user directional modulation, albeit with lower complexity. However, there have been previous demonstrations of metasurfaces for space-time coding and DM. If your contribution is on the reduced complexity, please provide a quantitative comparison against benchmarks (e.g., energy/per/bit, resulting, etc).

If the claimed contribution is on scaling one-user DM to multi-user DM, then the authors should justify why it is not straightforward to generalize and provide a more in-depth analysis of the design tradeoffs (i.e., maximum number of supported channels, minimum angular separation among users and Eve, etc).

Responses to Reviewers' Comments

We sincerely appreciate the new comments and suggestions from two reviewers, which further help us improve the quality of the manuscript. Below are our responses (in regular font) to each comment (*in italics and blue font*). We have revised the manuscript accordingly and highlighted all changes in the revised documents. We hope that the response letter has clarified the concerns of the reviewers.

Specific comments from Reviewer #1

Reviewer #1 – Comment 1:

The authors have responded to my comments properly in this round of revision. I think they need to make further minor revision to explain their response to my second comment somewhere in the main text of the manuscript.

Response:

We sincerely thank you for the recognition of our efforts. We have summarized the previous answers to your second question and added them to the main text of the manuscript:

- Lines 325~329, page 13: We remark that the transmission efficiency of different channels can be further improved by three methods: 1) increase the phase quantization level of the metasurface; 2) increase the scale of the metasurface to reduce the multi-user interference; and 3) employ an improved channel model to take the mutual coupling of the metasurface into account³⁸.

We also added the Ref. [R1] into the **References** section as the reference [38]:

[R1] R. W. Shao, J. W. Wu, Z. X. Wang, H. Xu, H. Q. Yang, Q. Cheng, T. J. Cui, *National Science Review*. **2024**, 11(3): nwa299.

General comments from Reviewer #2

While I appreciate the authors' effort to improve their manuscript, I still have concerns regarding the novelty. The space-time coding (or directional modulation) and its application for secure connectivity have been published in several other works. The claimed delta seems to be the following: a programmable meta-surface (instead of a time-modulated phased array) can realize multi-user directional modulation, albeit with lower complexity. However, there have been previous demonstrations of metasurfaces for space-time coding and DM.

Response:

We sincerely thank you for the insightful and professional comments. Following your comments, we have provided a quantitative comparison with the existing DIM systems and a more in-depth analysis of the design tradeoff. We hope that the following responses are sufficient for highlighting the novelty of the work.

Specific comments from Reviewer #2

Reviewer #2 – Comment 1:

If your contribution is on the reduced complexity, please provide a quantitative comparison against benchmarks (e.g., energy/per/bit, resulting, etc).

Response:

We have conducted a quantitative comparison with the existing categories that achieve DIM including the phased array, time-domain coding metasurface, and space-time-domain coding metasurface. The results are summarized in Table R1.

We here consider the following metrics to make a full comparison: the energy per bit, the performance (the number of supported channels and the supported modulation schemes), and the complexity of hardware.

The calculation of the energy per bit at unit time is

(R1)

where P , C , and M are the power consumption of the system, the information capacity of a single transmission, and the switching rate of the coding sequence, respectively. To facilitate a fair comparison, we set the switching rates M of different systems to the same value. In fact, the switching rate M depends on the performance of the active components and the control board, which renders an exact upper bound challenging to ascertain. Therefore, we quantify the energy per bit in terms of the (i.e.,) for simplification.

Table R1. Comparison with the reported schemes achieving DIM.

	Energy/bit	Performance		Hardware
		The number of channels	Supported modulation schemes	
Phased array	8 W	Single ^[R2]	QPSK	A four-element linear array
Time-domain coding metasurface	1.86 W	Single ^[R4]	256QAM	A 1-bit metasurface with 56×20 meta-atoms
Space-time-domain coding metasurface	1.33 W	Dual ^[R6]	QPSK	A 1-bit metasurface with 20×20 meta-atoms
	1.7 W	Three ^[R7]	QPSK	A 1-bit metasurface with 24×32 meta-atoms
Our work	0.106 W	Four	8PSK, 16QAM, and 64QAM	A 2-bit metasurface with 8×8 meta-atoms

1. Phased array

The conventional phased arrays, which require a T/R component for each array element, have the defects of high cost, high power consumption, and severe heat generation^[R1]. The power consumed by the T/R component is generally at the Watt level. For instance, the work in Ref.

[R2] reported the single-channel QPSK DIM based on a phased array. The system employs four antenna elements, each equipped with a digital phase shifter of MITEQ ^[R3]. The total consumption is about 16 W, so the energy per bit is 8 W/bit.

2. Time-domain coding metasurface

The work in Ref. [R4] reported a single-channel 256 QAM DIM based on the time-domain coding metasurface (TDCM). The metasurface consists of 56×20 meta-atoms, each equipped with a PIN diode (MADP-00097-14020) ^[R5]. The total consumption is about 14.9 W, so the energy per bit is about 1.86 W/bit.

Although the order of modulation is high, TDCM employs the harmonics of electromagnetic (EM) waves. The scheme causes two severe problems: 1) it leads to spectrum pollution and has low transmission efficiency; and 2) it fails to ensure directional secure transmissions due to the lack of space coding.

3. Space-time-domain coding metasurface

The work in Refs. [R6] and [R7] reported dual-channel and three-channel QPSK modulations based on the space-time-domain coding metasurfaces (STDCM). These metasurfaces consist of 20×20 and 24×32 meta-atoms, respectively, and each meta-atom is equipped with a PIN diode (MADP-00097-14020). The power consumption of the two metasurfaces are 5.32 W and 10.2 W, and the energies per bit are 1.33 W/bit and 1.7 W/bit, respectively.

The demonstrations based on STDCM have two defects: 1) the modulation scheme is mainly QPSK and the information capacity is low; and 2) STDCM needs an external feed horn, which causes several problems. Firstly, it has a low energy conversion efficiency. This is because the feed source is set at a certain distance from the metasurfaces, capturing and modulating only part of the waves launched by the source. The freedom of the meta-atoms at the edge of metasurfaces is wasted since the source only illuminates the central part of the metasurfaces to avoid the influence of diffraction. Secondly, the system has the occlusion effect caused by the feed source, which creates a partial blind spot. Finally, the high profile of the configuration

composed of STDCM and extended antennas is inconvenient for some space-constrained applications such as satellites, vehicles, and aircrafts.

4. Our work

The programmable metasurface (PM) consists of 8×8 meta-atoms, each equipped with four PIN diodes (MADP-000907-14020). The total power consumption of the metasurface is 2.55 W. The current system supports up to four channels of 64QAM transmission, which is equivalent to bits of information at a time. Therefore, the energy per bit of our design is about 0.106 W/bit, being the minimum among all above reported systems.

We have demonstrated the four-channel 8PSK, 16QAM, and 64 QAM DIMs based on the scheme of STDCM. We have designed and fabricated a low-profile PM, which features the abilities of simultaneous information modulation and wave radiation. Therefore, the hardware overcomes the problems of low energy conversion, blockage effects, and high profile that are typical and inherent to the conventional STDCM. Furthermore, the hardware has a smaller number of meta-atoms (64) and a higher phase quantization (2-bit), as compared with Refs. [R6] and [R7]. Although the scale is not as large as the previous work, the hardware has more degrees of freedom (DoF). We further developed a fast and feasible algorithm to efficiently exploit the massive number of DoF.

Changes:

We have added the analyses and comparison of different systems in Supplementary Note 1.

In the revised main text, we have also added the following sentences to clarify the low energy consumption of our design:

- Lines 93-94, page 3: Moreover, the required energy per bit is high due to the low information capacity of a single transmission (see Supplementary Note 1 for more details).

[R1] M. Robert, *Phased Array Antenna Handbook, Third Edition*. Artech, 2017.

- [R2] Digital Phase Shifters, MITEQ, Inc. [Online]. Available: <http://amps.miteq.com/datasheets/MITEQ-DPS.PDF>
- [R3] M. P. Daly, J. T. Bernhard, *IEEE Transactions on Antennas and Propagation*. **2009**, 57, 2633.
- [R4] M. Z. Chen, W. Tang, J. Y. Dai, J. C. Ke, L. Zhang, C. Zhang, J. Yang, L. Li, Q. Cheng, S. Jin, T. J. Cui, *National Science Review*. **2022**, 9, nwab134.
- [R5] MADP-000907-14020W PIN Diode, MACOM, Inc. [Online]. Available: <https://www.macom.com/products/product-detail/MADP-000907-14020W>.
- [R6] X. Wan, C. Xiao, H. Huang, Q. Xiao, W. Xu, Y. Li, J. Eisenbeis, J. Wang, Z. Huang, Q. Cheng, S. Jin, T. Zwick, T. Cui, *Engineering*. **2022**, 8, 86.
- [R7] H. Zhao, Y. Shuang, M. Wei, T. J. Cui, P. D. Hougne, L. Li, *Nature Communications*. **2020**, 11, 3926.

Specific comments from Reviewer #2

Reviewer #2 – Comment 2:

If the claimed contribution is on scaling one-user DM to multi-user DM, then the authors should justify why it is not straightforward to generalize and provide a more in-depth analysis of the design tradeoffs (i.e., maximum number of supported channels, minimum angular separation among users and Eve, etc).

Response:

We sincerely thank you for the suggestive comment, which inspires us to have a clear recognition of the challenges in our scheme.

For your first question, we claim that the multi-channel and high-order DIM is a challenging task for the metasurfaces with finite scale, especially with low-bit phase quantization. On the one hand, the finite scale of the metasurface restricts the DOF, which directly affects the number of supported channels. On the other hand, the error caused by the low-bit phase quantization hinders the realization of high-order modulation. To this end, it is desired to use large-scale metasurfaces with higher-bit phase quantization, but such a task is challenging in terms of

technique complexity and hardware cost. From the perspective of design tradeoff, we employ an 8×8 programmable metasurface with 2-bit phase quantization. Although the scale is not large enough, our system has massive degrees of freedom (DoF). Furthermore, we develop a fast and feasible algorithm to efficiently employ the massive number of DoF. **It is these two contributions that make our performance better than the previous work**, as summarized in Table R1.

Following your insightful comments, below we present a more in-depth analysis of design tradeoff. The initial step is to delineate the criteria for distinguishing users and eavesdroppers, which involves the determination of minimum angular width required to ensure directional security. Subsequently, the theoretical maximum number of supported channels is calculated according to the above angular width.

1. The minimum angular width to ensure directional security

As mentioned in the main text, the error vector magnitude (EVM) is a critical metric to assess the similarity between the received fields and reference constellation points. Particularly, the value of EVM is lower in the vicinity of desired users. In this context, we define the concept of the secure zone, where the value of EVM is smaller than a certain threshold ε to separate the desired users and eavesdroppers. As shown in Figure R1, EVM has a strong correlation with the radiation pattern of the metasurface, with lower value of EVM near the main lobe beam. Therefore, we utilize the main lobe beam as the secure zone and set the half-power beamwidth as the minimum angular width.

Figure R1. The relationship between EVM and the radiation pattern of the metasurface. The secure zone, marked by a transparent and green rectangle, is in the main lobe beam of the metasurface.

We firstly estimate the value of threshold ε when the secure zone is the main lobe beam. For simplicity, we consider the one-dimensional situation. The definition of EVM is

$$(R2)$$

where r , r_0 , and N are the received field, the reference constellation point, and the number of trials, respectively; r and r_0 are normalized to ensure $\int |r|^2 d\theta = \int |r_0|^2 d\theta$ and

when $\theta = \theta_0$. The received field r can be decomposed into

two parts, namely, $r = r_0 + \Delta r$, where Δr includes the error caused by the quantization effects of meta-atoms and the additive white Gaussian noise. Therefore, the ideal field is equal to r_0 in the desired direction. In other directions, r is smaller than r_0 (notice that the desired direction is in the main lobe beam). The value of EVM at the edge of main lobe beam is

(R3)

In Equation (R3), we suppose that the signal is statistically independent with noise. After ignoring the noise, the maximum value of EVM in the main lobe beam is about -10.7 dB. We should emphasize that the above value is calculated under the assumption of infinite resolution. For the case of 2-bit phase quantization, we will give the value soon.

To demonstrate the theoretical estimation, we conduct simulations to calculate the half-power beamwidth (i.e., 3-dB width) of the radiation pattern and the secure zone width (i.e., the region where $\text{EVM} < -10 \text{ dB}$). According to the above theory, the two widths are almost equal. We here consider the single-channel DIM that supports 16QAM scheme, and the desired directions are 0° , 25° , 40° , and 55° , respectively. We also change the number of meta-atoms with 4, 8, and 16 to validate the scalability. Figures R2(a) and R2(b) demonstrate how the half-power beamwidth and the width of the secure zone varies with the number of meta-atoms. As indicated, the widths decrease as the number of meta-atoms increases. Specifically, when the number of meta-atoms doubles, the widths decrease by half, which indicates that a larger array has superior characteristics of directional security. The difference between the two widths is small and decrease as the number of meta-atoms increases, as shown in Figure R2(c). Taking the case of 8 meta-atoms as a representative example, we extract the detailed values of the widths and the difference between them, as listed in Table R2. The difference between the two widths is smaller than 1° . It indicates that the two widths are almost equal, being consistent with the theoretical predication.

Figure R2. The simulation results to demonstrate the minimum angular width under the assumption of infinite resolution. (a) The half-power beamwidth of the radiation pattern. (b) The secure zone width (i.e., the region where $\text{EVM} < \epsilon$). (c) The difference between the two widths.

Table R2. The detailed 3-dB and EVM angular widths in different beam directions when the number of meta-atoms is 8 with the infinite resolution.

direction	width	3-dB width	EVM width	The difference
10°	13°	13°	13°	0°
25°	13.5°	13°	13.5°	0.5°
40°	16.5°	16°	16.5°	0.5°
55°	21°	20°	21°	1°

The above theory and simulation results demonstrate that the half-power beamwidth and the secure zone width (i.e., the region where $\text{EVM} < \epsilon$) is nearly the same under the assumption of infinite resolution. To find the secure zone width in the case of 2-bit phase quantization, we gradually increase the value of threshold ϵ until the difference between the two widths is minimal. As shown in Figure R3(a), the average difference between the two widths is smallest when the threshold value is $\epsilon = 0.01$. Figures R3(c) and R3(d) show the secure zone width (i.e., the region where $\text{EVM} < \epsilon$) and the difference between

the two widths, respectively. Taking the case of 8 meta-atoms as a representative example, we extract the detailed values of the widths and the difference between them, as listed in Table R3. The difference of the two widths is smaller than 1.5° , which is the smallest in many attempts about the value of threshold ε .

Furthermore, we quantify the noise according the optimal threshold value. On the one hand, the value of EVM at the edge of main lobe beam in the case of 2-bit phase quantization is

On the other hand, the value of EVM

in the case of infinite resolution is

The value of

the noise is about

. Hence we can claim that, no matter how the scale of

the metasurface increases, the minimum EVM in the desired direction is bigger than -27 dB under the constraint of 2-bit phase quantization. As shown in Figure R1, the value of EVM in the desired direction is about -24.8 dB, which is very close to the minimum and indicates the good performance of our discrete optimization algorithm.

If we utilize the main lobe beam of the metasurface as the secure zone and set the half-power beamwidth as the minimum angular width, the maximum value of EVM is -10.6 dB in the case of 2-bit phase quantization. The threshold value may be too large to be good enough to separate the desired users and eavesdroppers, especially in the case of high-order QAM scheme. It is obviously that the secure zone width decreases if one reduces the threshold value ε . However, there will be no clear relationship between the secure zone width and threshold value. More importantly, there will be also no clear relationship between the secure zone width and the scale of metasurface. Therefore, we employ the main lobe beam of metasurface as the secure zone.

Figure R3. The simulation results to demonstrate the minimum angular width in the case of 2-bit phase quantization. (a) The average difference between the two widths as the threshold value changes. (b) The half-power beamwidth of the radiation pattern. (c) The secure zone width (i.e., the region where). (d) The difference between the two widths when the threshold value is -10.6 dB.

Table R3. The detailed 3-dB and EVM angular widths at different beam directions when the number of meta-atoms is 8 with the 2-bit phase quantization.

direction \ width	3-dB width	EVM width	The difference
10°	13°	11.5°	-1.5°
25°	13°	14.5°	1.5°
40°	16°	16°	0°
55°	20°	19.5°	-0.5°

2. The maximal number of supported channels

If we utilize the main lobe beam of the metasurface as the secure zone, the minimum angular width has a closed form. For a metasurface with the aperture size A , the solid angle of the half-power beam is approximately given by ^[R8]

(R4)

where λ is the wavelength. The maximal number of the supported channels in the front-half plane is then given as

(R5)

where D is the directivity of the metasurface.

We should emphasize that Equations (R4) and (R5) ignore the issues of nonuniform surface current on the metasurface and the expansion of half-power beamwidth under large scanning angles. The actual number of supported channels will be smaller than the theoretical value .

Changes:

We have added the above analyses in a new Supplementary Note 13 in the revised version. We also added the following sentences in the main text to highlight our further modification:

- Lines 278-281, page 10: We further propose the concept of secure zone, defined by the value of EVM, to separate the desired users and eavesdroppers. The in-depth analyses are conducted on the minimum angular width and the maximum number of supported channels (see Supplementary Note 13 for more details).

[R8] J. D. Kraus and R. J. Marhefka, *Antennas for All Applications*. New-York: McGraw-Hill, 2002.

REVIEWERS' COMMENTS

Reviewer #2 (Remarks to the Author):

I don't have any further comments. Please include the comparison table you provided in the responses in the supplementary information as well, because this will be helpful for the readers to understand the delta over past work.

Responses to Reviewers' Comments

General comments from Reviewer #2

Reviewer #2 – Comment 1:

I don't have any further comments. Please include the comparison table you provided in the responses in the supplementary information as well, because this will be helpful for the readers to understand the delta over past work.

Response:

We sincerely appreciate your acceptance. Your insightful comments greatly helped us improve and strengthen this work. We are very pleased that you are happy with the changes we have made to the manuscript following your suggestions and questions. Following your suggestion, we have incorporated the comparative table, as shown in Table S1 of the supplementary information.